# Achieving Adaptivity and Optimality for Multi-armed Bandits using Exponential Kullback-Leibler Maillard Sampling

**Hao Qin**                                                           *hqin@arizona.edu*
*Department of Mathematics*
*University of Arizona*

**Kwang-Sung Jun**                                                    *kjun@cs.arizona.edu*
*Department of Computer Science*
*University of Arizona*

**Chicheng Zhang**                                                    *chichengz@cs.arizona.edu*
*Department of Computer Science*
*University of Arizona*

**Reviewed on OpenReview:** *https://openreview.net/forum?id=IuVkRmecVp*

## Abstract

We study the problem of $K$-armed bandits with reward distributions belonging to a one-parameter exponential distribution family. In the literature, several criteria have been proposed to evaluate the performance of such algorithms, including Asymptotic Optimality, Minimax Optimality, Sub-UCB, and variance-adaptive regret bound. Thompson Sampling-based and Upper Confidence Bound-based algorithms have been employed to achieve some of these criteria. However, none of these algorithms simultaneously satisfy all the aforementioned criteria.

In this paper, we design an algorithm, Exponential Kullback-LeiblerMaillard Sampling (abbrev. EXP-KL-MS), that achieves multiple optimality criteria simultaneously, including Asymptotic Optimality, minimax ratio with a $\sqrt{\ln(K)}$ factor, Sub-UCB, and a variance-adaptive regret bound. Our algorithm design follows the Minimum Empirical Divergence framework (Honda & Takemura, 2011; Maillard, 2011), with the exploration probability of arm $a$ proportional to $\exp\left(-L(N_{t-1,a})\mathrm{KL}(\hat{\mu}_{t-1,a}, \max_{a'} \hat{\mu}_{t-1,a'})\right)$, where $L(\cdot)$ is an inverse temperature function, $N_{t-1,a}$ is the number of times arm $a$ has been pulled before time $t$, $\hat{\mu}_{t-1,a}$ is the empirical mean of arm $a$ before time $t$, and $\mathrm{KL}(\cdot,\cdot)$ is the Kullback-Leibler divergence between two distributions in the one-parameter exponential distribution family. Our analysis allows different choices of inverse temperature function $L(k)$. We also provide numerical simulations demonstrating the effectiveness of our algorithms.

## 1 Introduction

The Multi-Armed Bandit (MAB) problem models sequential decision making in which an agent takes an action, receives a reward from the environment, and would like to learn to maximize its cumulative reward. It has attracted significant attention within the research community due to its foundational nature. Moreover, it has found practical applications in various domains, including online advertising (Geng et al., 2020) and clinical trials (Villar et al., 2015).

Formally, an MAB environment consists of $K$ arms (actions), each denoted by an integer $a \in [K] := \{1, \ldots, K\}$. Each arm is associated with a reward distribution $\nu_a$ with mean $\mu_a$. The learning agent selects an arm $I_t \in [K]$ at each time step $t$ and receives a reward $r_t \sim \nu_{I_t}$ from the environment.

One-Parameter Exponential Distribution (OPED) families are classes of probability distributions characterized by their ability to express the likelihood of a set of outcomes in terms of a natural parameter. The OPED family framework allows for flexible modeling of various types of data. In many applications, the reward comes from an OPED family. For example, the reward may follow a Bernoulli distribution (Bouneffouf et al., 2017; Shen et al., 2015) in the case of binary outcomes. Alves et al. (2021) models the user's click in the recommender system as a mixture Poisson distribution with a limited time window. Jia et al. (2021) models the pricing-based revenue management problem as an MAB, with different pricing being an arm and usage duration following an exponential distribution. An OPED family with identity sufficient statistic induced by base measure $m$ is defined as:

$$\mathcal{F}_m = \{p_\theta(dx) = m(dx)\exp\left(x\theta - b(\theta)\right) : \theta \in \Theta\}, \tag{1}$$

where $\theta$ is the natural parameter, $b(\theta) \coloneqq \ln\left(\int_{\mathbb{R}} \exp\left(x\theta\right) m(dx)\right)$ is the log-partition function, $\Theta \subset \mathbb{R}$ is the space of canonical parameters. Throughout the paper, we assume that the reward distributions of all arms belong to a common $\mathcal{F}_m$.

At each time step, the learning agent makes decisions based on the historical information it has gathered, balancing the exploration-exploitation trade-off. It may choose to explore by pulling arms that have not performed as well as expected to obtain better estimates. Alternatively, it might exploit the arm that has shown good performance, though this comes with the risk of relying on potentially inaccurate estimates. The agent's primary goal is to maximize its cumulative reward over time, which is equivalent to minimizing regret by choosing the optimal arm. We denote the maximum expected reward for the environment by $\mu_{\max} \coloneqq \max_{a \in [K]} \mu_a$ and define the pseudo-regret as

$$\text{Regret}(T) \coloneqq \sum_{t=1}^{T} \mu_{\max} - \mathbb{E}[r_t] = \sum_{t=1}^{T} \mu_{\max} - \mathbb{E}[\mu_{I_t}].$$

In the literature, several works have proposed MAB algorithms designed for rewards drawn from the exponential family of distributions (Korda et al., 2013; Cappé et al., 2013; Ménard & Garivier, 2017; Jin et al., 2022; 2023; Qin et al., 2023), achieving several optimality properties such as Minimax Optimality (Auer et al., 2003; Audibert et al., 2009a), Asymptotic Optimality (Lai et al., 1985; Korda et al., 2013; Jin et al., 2022; 2023), the Sub-UCB criterion (Lattimore, 2018), and Adaptive Variance Ratio (Qin et al., 2023) – see Section 3 (related work) for formal definitions. To date, no algorithm has been identified that simultaneously satisfies all of the above optimality and adaptivity criteria in the setting of OPED reward distributions (see Table 1 for a comparison and Section 3 for detailed discussions). ADA-UCB (Lattimore, 2018) satisfies multiple optimality criteria in the special case of Gaussian rewards. The most recent example is ExpTS (Jin et al., 2022), which achieves a logarithmic minimax ratio of $\sqrt{\ln(K)}$, along with asymptotic optimality and Sub-UCB. However, several studies (Ménard & Garivier, 2017; Jin et al., 2022; 2023) focus their analysis on the maximum variance assumption (Assumption 2), which results in their regret bounds not enjoy an adaptive variance ratio guarantee.

**Our contributions.** In recent years, the research community has shown increasing interest in Maillard Sampling-style algorithms (Honda & Takemura, 2011; Maillard, 2011), a family of randomized MAB algorithms with several desirable features, including both instance-dependent and instance-independent guarantees, as well as a closed-form exploration probability suitable for offline evaluation. However, to date, such algorithms and analyses have been designed and conducted only under restrictive assumptions on the reward distributions, such as finite supports (Honda & Takemura, 2011), sub-Gaussian distributions (Bian & Jun, 2022), and Bernoulli distributions (Qin et al., 2023).

In this paper, we propose a Maillard Sampling-style algorithm for a range of OPED reward distributions, which we refer to as Exponential Kullback-Leibler Maillard Sampling (Exp-KL-MS). We demonstrate that Exp-KL-MS possesses several key properties, including asymptotic optimality, a logarithmic minimax ratio of $\sqrt{\ln(K)}$, Sub-UCB, and an adaptive variance ratio, as established through both asymptotic and finite-time analyses. The Exp-KL-MS algorithm selects an arm according to the following distribution:

$$\mathbb{P}(I_t = a) \propto \exp\left(-L(N_{t-1,a})\mathcal{D}(\hat{\nu}_{t,a}, \hat{\nu}_{t,\max})\right), \tag{2}$$

where $\mathcal{D}(\nu, \nu')$ is the Kullback-Leibler (KL) divergence between two distributions $\nu$ and $\nu'$. $N_{t-1,a}$ is the number of times arm $a$ has been pulled up to time $t-1$, $\hat{\nu}_{t,a}$ is the maximum likelihood estimate (MLE) of the reward distribution $\nu_a$ based on the rewards from arm $a$ up to time step $t-1$ (inclusively). $\hat{\nu}_{t,\max}$ is the MLE of the distribution with the highest empirical mean at time step $t$. The function $L(\cdot)$ is an inverse temperature function, with $L(k) = k - 1$ for $k \geq 1$.

A large value of $N_{t-1,a}$ indicates that arm $a$ has been pulled frequently in the past, leading to a smaller value for $\mathbb{P}(I_t = a)$ and making arm $a$ less likely to be selected. Similarly, a large value of $\mathcal{D}(\hat{\nu}_{t,a}, \hat{\nu}_{t,\max})$ suggests that the estimated reward distribution of arm $a$ deviates significantly from that of the empirical best arm, resulting in a smaller $\mathbb{P}(I_t = a)$. Additionally, we can interpret EXP-KL-MS as applying the principle of Minimum Empirical Divergence (MED) (Honda & Takemura, 2011) to OPEDs. For detailed comparison, see Section 3.

We also analyze other choices of $L(k)$, such as $L(k) = k/d$, where $d \geq 1$, and $L(k) = k$, and present their guarantees in Appendix appendix A. Generally, as long as $L(k)$ is monotonically increasing with $k$ and remains no greater than $k$, the algorithm, which we refer to as GENERAL-EXP-KL-MS, will exhibit multiple optimality properties, including a logarithmic minimax ratio of $\sqrt{\ln(K)}$ and Sub-UCB.

We show that EXP-KL-MS achieves a finite-time regret guarantee (Theorem 1) which can be simultaneously converted into:

- Minimax optimality guarantee (Corollary 2), up to a logarithmic factor. EXP-KL-MS's regret is at most a suboptimal logarithmic factor $\sqrt{\ln(K)}$ compared to the minimax optimal regret of $\Theta(\sqrt{KT})$ (Audibert et al., 2009a; Auer et al., 2003).

- An asymptotic regret upper bound (Corollary 3) that matches the lower bound established by Lai et al. (1985), thus showing EXP-KL-MS satisfies asymptotic optimality.

- A Sub-UCB regret guarantee (Corollary 4) that ensures that the algorithm's performance is at least as good as the UCB algorithm in the finite-time regime.

- Adaptive variance ratio (Corollary 5). EXP-KL-MS achieves a regret bound of $\widetilde{O}\big(\sqrt{V(\mu_{\max})KT}\big)$, which has an instance-specific parameter $V(\mu_{\max})$ that adapts to the variance of the optimal arm's reward distribution. Before our work, such results were only established in the $[0,1]$-bounded reward setting (Qin et al., 2023).

**Our Techniques** A natural way to extend Maillard Sampling (MS) (Maillard, 2011) to the OPED reward setting is to analyze the sampling rule (Equation (2)) with $L(k) = k$, which was analyzed in prior works in finite-sized support (Honda & Takemura, 2010) or specific reward distribution (e.g. Gaussian (Bian & Jun, 2022), Bernoulli (Qin et al., 2023)) settings. However, in the asymptotic optimality analysis of this generalized algorithm, a naive generalization of the analysis in prior works (Honda & Takemura, 2010; Bian & Jun, 2022; Qin et al., 2023) can no longer bound the number of suboptimal arm pulls in time steps when the optimal arm performs poorly. Concretely, the bound on the "optimal arm underestimated" term in the Bernoulli bandits analysis is obtained via an integral that is finite on the singular points; for other OPED families (e.g. exponential), the same integral may diverge on the singular points (Eq. (22)), so a naive extension of the Bernoulli analysis cannot deliver asymptotic optimality. We defer the detailed explanation to Appendix D.4.

To overcome this challenge, we take inspiration from Jin et al. (2022)'s Thompson sampling-style algorithm and modify our algorithm to $L(k) = k - 1$. As we prove in our analysis, this slight change helps the proposed algorithm to have good properties shared by many MS-based algorithms and allows us to show new variance-adaptive regret guarantees.

## 2 Preliminaries

We consider a standard $K$-armed bandit problem, where arms are indexed by $\{1, 2, \ldots, K\} =: [K]$. For an arm $a$, it is associated with a reward distribution $\nu_a$ over $[R_{\min}, R_{\max}]$ with mean $\mu_a$, where $\mu_a, R_{\min}, R_{\max} \in$

Table 1: Comparison of different MAB algorithms for reward distributions belonging to an OPED family in the form of Equation (1). All entries report each algorithm's published guarantees in the general OPED reward setting defined by Assumption 1. "N/A" indicates that it is unknown whether the algorithm satisfies the given guarantee, while "No" explicitly indicates that the algorithm does not satisfy it. No* means the published analysis does not yield the guarantee in general OPED, although the corresponding result has been established for the Bernoulli reward setting by Qin et al. (2023). ExpTS/ExpTS$^+$ uses the maximum variance assumption, thus it is unclear whether their regret bounds satisfy the adaptive variance ratio guarantee.

| Algorithm & Analysis | Instance-Dependent | | Instance-Adaptive | Instance-Independent |
|---|---|---|---|---|
| | Asymptotic Optimality | Sub- UCB | Variance Ratio | Minimax Ratio |
| TS  (1933; 2013) | Yes | N/A | N/A | N/A |
| ExpTS  (2022) | Yes | Yes | N/A | $\sqrt{\ln(K)}$ |
| ExpTS$^+$  (2022) | Yes | No | No | 1 |
| $\varepsilon$-Exploring TS$^\star$  (2023) | Yes | No | No | $\sqrt{\ln(K)}$ |
| kl-UCB  (2013; 2023) | Yes | Yes | N/A | $\sqrt{\ln(T)}$ |
| kl-UCB++  (2017; 2023) | Yes | N/A | No* | 1 |
| Exp-KL-MS | Yes | Yes | Yes | $\sqrt{\ln(K)}$ |

$\mathbb{R} \cup \{-\infty, +\infty\}$ and $R_{\min} \leq \mu_a \leq R_{\max}$. The distribution associated with the optimal arm is $\nu_{\max}$, and its mean is $\mu_{\max}$.

The agent interacts with the bandit environment $T$ times. At each time step $t$, the agent pulls an arm $I_t$ from $[K]$ and receives a reward $r_t$. Before time $t$, the number of times arm $a$ has been pulled is $N_{t-1,a} := \sum_{i=1}^{t-1} \mathbf{1}_{I_i=a}$, where $\mathbf{1}$. is an indicator function. The empirical mean of arm $a$ is $\hat{\mu}_{t,a} := 1/N_{t,a} \sum_{i=1}^{t} r_i \mathbf{1}_{I_i=a}$. We also denote the estimated reward distributions as $\{\hat{\nu}_{t,1}, \hat{\nu}_{t,2}, \ldots, \hat{\nu}_{t,K}\}$, which are distributions in $\mathcal{F}_m$ with means $\hat{\mu}_{t,1}, \hat{\mu}_{t,2}, \ldots, \hat{\mu}_{t,K}$. The best empirical reward is $\hat{\mu}_{t,\max} := \max_{a \in [K]} \hat{\mu}_{t,a}$, which is associated with the distribution $\hat{\nu}_{t,\max}$.

Additionally, we denote the arm sampling distribution at time step $t$ by $(p_{t,a})_{a \in [K]}$ where each $p_{t,a}$ represents the probability of pulling arm $a$ at time step $t$.

## 2.1 OPED family and variance function

We introduce several assumptions to characterize the behavior of the reward distributions and to facilitate our analysis. First, we assume that the log-partition function is sufficiently simple to permit tractable analysis.

**Assumption 1.** $b(\theta)$ *is twice differentiable with a continuous second derivative* $b''(\theta) > 0$, $\forall \theta \in \Theta$.

**Assumption 2.** *For any distribution $p_\theta$ in $\mathcal{F}_m$, its variance is bounded above by $\bar{V} > 0$.*

It can be verified that many widely used distributions, such as Bernoulli, Poisson, and Gaussian distributions, satisfy these two assumptions. Based on Assumptions 1 and 2, a reward distribution $\nu \in \mathcal{F}_m$ has the following properties, $\mu := b'(\theta) = \mathbb{E}_{x \sim \nu}[x], b''(\theta) = \mathrm{Var}_{x \sim \nu}[x] \leq \bar{V}$.

Assumption 2 imposes a maximum variance constraint on the distribution family, whereas in many distribution families, the variance can be unbounded. For example, as shown in Table 2, the Gamma and Inverse Gaussian distribution families exhibit unbounded variance without restricting the maximum mean value. Additionally, we require that all arm distributions within a single bandit instance belong to the same distribution family:

**Assumption 3.** *There exists a known OPED family $\mathcal{F}_m$ s.t. $\forall a \in [K], \nu_a \in \mathcal{F}_m$.*

Assumption 3 states that for any measure $m(\cdot)$ and function $b(\cdot)$, one can define a distribution family $\mathcal{F}_m$ under the OPED framework, from which all reward distributions $\nu_{a a \in \mathcal{K}}$ are drawn. For example, by choosing $m(\cdot)$ as the counting measure on the set $\{0, 1\}$, we recover the family of all Bernoulli distributions. By

Table 2: Some examples of OPED family in the form of Equation (1) along with their key parameters. $\mathcal{B}$ denotes the Bernoulli distribution family. $\mathcal{P}$ represents the Poisson distribution family. $\mathcal{N}_\sigma$ includes all Normal distributions with fixed variance $\sigma^2$. $\Gamma_k$ denotes the Gamma distribution family with fixed shape parameter $k$. We use a non-standard parameterization of the Gamma distribution based on the mean $\mu$ and shape $k$ intentionally to unify notation across different distribution families. $\mathcal{IG}_\lambda$ is the inverse Gaussian distribution family with fixed $\lambda$. The variance function maps mean to the variance, and all these families satisfy Assumption 4. For example, the mean parameter in $\Gamma_k$ and $\mathcal{IG}_\lambda$ is bounded by $M$. The variance function for $\Gamma_k$ is $V(x) = \frac{x^2}{k}$ with a Lipschitz constant of $\frac{2M}{k}$, while for $\mathcal{IG}_\lambda$, the variance function is $V(x) = \frac{x^3}{\lambda}$ with a Lipschitz constant of $\frac{3M^2}{\lambda}$.

| Distribution | Mean | Variance |
|---|---|---|
| $\mathcal{B} = \left\{ p(x) = \mu^x (1-\mu)^{1-x}, \mu \in (0,1) \right\}$ | $\mu$ | $\mu(1-\mu)$ |
| $\mathcal{P}(M) = \left\{ p(x) = \frac{\mu^x e^{-\mu}}{x!}, \mu \in (0,M) \right\}$ | $\mu$ | $\mu$ |
| $\mathcal{N}_\sigma = \left\{ p(x) = \frac{1}{\sigma\sqrt{2\pi}} \exp\left( -\frac{1}{2}\left(\frac{x-\mu}{\sigma}\right)^2 \right), \mu \in \mathbb{R} \right\}$ | $\mu$ | $\sigma^2$ |
| $\Gamma_k(M) = \left\{ p(x) = \frac{1}{\Gamma(k)(\mu/k)^k} x^{k-1} e^{-xk/\mu}, \mu \in (0,M) \right\}$ | $\mu$ | $\mu^2/k$ |
| $\mathcal{IG}_\lambda(M) = \left\{ p(x) = \sqrt{\frac{\lambda}{2\pi x^3}} \exp\left( -\frac{\lambda(x-\mu)^2}{2\mu^2 x} \right), \mu \in (0,M) \right\}$ | $\mu$ | $\mu^3/\lambda$ |

defining $m(\cdot) = \sum_{i=0}^{\infty} \frac{1}{i!} \delta_i(\cdot)$, where $\delta_x$ denotes the Dirac measure at $x$, we obtain the family of all Poisson distributions. [1]

We define the variance function as $V(\mu) = b''((b')^{-1}(\mu))$, which maps a mean $\mu(\theta)$ to its corresponding variance, i.e., $V : \mu(\theta) \mapsto b''(\theta)$. The KL divergence between two distributions $\nu$ and $\nu'$ is defined as $\mathcal{D}(\nu, \nu') := \mathbb{E}_{X \sim \nu}\left[ \ln\left(\frac{d\nu}{d\nu'}(X)\right) \right]$ if $\nu$ is absolutely continuous w.r.t. $\nu'$ and $+\infty$ otherwise. To emphasize the mean parameters in our analysis, we denote the KL divergence between two distributions $\nu_i, \nu_j$ with means $\mu_i, \mu_j$ as: $D_{\mathrm{KL}}(\mu_i, \mu_j) = \mathcal{D}(\nu_i, \nu_j)$. According to Lehmann & Casella (2006), if the distributions $\nu_i$ and $\nu_j$ have natural parameters $\theta_i, \theta_j$, respectively, their KL divergence is given by

$$D_{\mathrm{KL}}(\mu_i, \mu_j) = b(\theta_j) - b(\theta_i) - b'(\theta_i)(\theta_j - \theta_i) \tag{3}$$

## 3 Related Work

We now review the common types of regret guarantees in the literature.

### 3.1 Fully instance-independent regret guarantees

Fully instance-independent regret guarantees provide uniform performance bounds across all bandit instances. Auer et al. (2003) shows that, in the Bernoulli reward setting, the regret lower bound for any bandit algorithm is at least $\Omega(\sqrt{KT})$. Conversely, Audibert et al. (2009a) shows that the MOSS algorithm achieves a regret upper bound of at most $O(\sqrt{KT})$ for reward distributions supported on $[0, 1]$. Motivated by this, an algorithm is said to be **minimax optimal** if its regret satisfies $\mathrm{Regret}(T) = \Theta(\sqrt{KT})$. More generally, an algorithm is said to have a minimax ratio if there exists a function $f(K, T)$ such that $\mathrm{Regret}(T) = O(\sqrt{KT}f(K,T))$. Certain bandit algorithms employing carefully designed sampling distributions achieve a minimax ratio of $\sqrt{\ln(K)}$ (Jin et al., 2022; 2023). Numerous upper confidence-bound strategies, but not all, achieve a minimax ratio of $\sqrt{\ln(T)}$ (Auer et al., 2002; Cappé et al., 2013). Agrawal & Goyal (2017) shows that Thompson Sampling (TS) with a Beta prior can reach a $\sqrt{\ln(T)}$ minimax ratio, and when the reward distributions are Gaussian, the minimax ratio improves to $\sqrt{\ln(K)}$. Furthermore, kl-UCB++ (Ménard & Garivier, 2017) achieves a minimax ratio of $O(1)$ assuming OPED reward distributions.

---

[1]$\mathbb{N}_0$ represents all natural numbers starting from 0. Some distribution families, such as the Gamma distributions with a fixed scale, varying shape parameter $\alpha$, are characterized by a single parameter. However, they do not have the form of Equation (1) due to the sufficient statistic being nonlinear.

### 3.2 Fully instance-dependent regret guarantees

Instance-dependent regret guarantees provide bounds that adapt to the difficulty of each problem instance. These guarantees encompass two primary criteria: **asymptotic optimality** and **Sub-UCB**.

Lai et al. (1985) shows that for every instance under a consistent algorithm, the regret is lower-bounded: $\limsup_{T\to\infty} \frac{\text{Regret}(T)}{\ln(T)} \geq \sum_{a\in[K]:\Delta_a>0} \frac{\Delta_a}{\mathcal{D}(\nu_a,\nu_{\max})}$, where $\Delta_a := \mu_{\max} - \mu_a$ is the suboptimality gap of arm $a$. Several studies proposed algorithms with asymptotic optimality guarantees, including TS with conjugate priors (Korda et al., 2013), ExpTS (Jin et al., 2022), which uses non-conjugate priors, and KL-UCB (Cappé et al., 2013). Algorithms such as AOUCB (Lattimore & Szepesvári, 2020) and MS/MS$^+$ (Bian & Jun, 2022) have also demonstrated asymptotic optimality guarantees under the assumption of sub-Gaussian reward distributions.

Before Lattimore (2018), the literature primarily focused on asymptotic optimality and minimax optimality. While MOSS (Audibert et al., 2009a) is both asymptotic optimal and minimax optimal, Lattimore (2018) demonstrated that, in certain bandit instances, MOSS falls short in comparison with the simpler UCB algorithm in the finite time regime. This observation suggests that traditional measures of optimality, such as asymptotic optimality and minimax optimality, do not fully capture the complete performance spectrum of a bandit algorithm. To address this gap, Lattimore (2018) introduced the Sub-UCB criterion as a complement to asymptotic and minimax optimality. The Sub-UCB criterion aims to evaluate whether an algorithm can match the performance of the UCB algorithm in finite-time regimes. An algorithm is said to satisfy the Sub-UCB criterion if there exist two constants $C_1$ and $C_2$ such that $\text{Regret}(T) \leq C_1 \sum_{a\in[K]} \Delta_a + C_2 \sum_{a\in[K]:\Delta_a>0} \frac{\ln(T)}{\Delta_a}$.

This criterion is called "Sub-UC" because it represents a standard form of gap-dependent regret guarantee associated with UCB (Auer et al., 2002; Lattimore, 2018). Algorithms such as MOSS (Audibert et al., 2009a), MOSS-Anytime (Degenne & Perchet, 2016), and kl-UCB++ (Ménard & Garivier, 2017) fail to satisfy the Sub-UCB criterion, despite achieving minimax and asymptotic optimality. More recently, MS$^+$ (Bian & Jun, 2022) and KL-MS (Qin et al., 2023) have shown that it is possible to achieve both the Sub-UCB criterion and minimax ratio with a $\sqrt{\ln(K)}$ ratio simultaneously, provided that all arms' reward distributions are either sub-Gaussian or supported on $[0,1]$. To make the necessity of this finite-time criterion concrete, Lattimore (2018) exhibits a $K$-armed Gaussian bandit instance (with gaps $\Delta_2 = 1/K$, $\Delta_a = 1$ for $a > 2$, and horizon $T = K^3$) on which MOSS, despite being asymptotically and minimax optimal, incurs $\Omega(\sqrt{KT}) = \Omega(K^2)$ regret, while the simpler UCB incurs only $O(K\ln K)$ regret.

### 3.3 Partially instance-independent guarantees

Partially instance-independent guarantees lie between fully instance-independent and instance-dependent regret guarantees. One such guarantee studied in prior works is the adaptive variance ratio (Qin et al., 2023). An algorithm $\mathcal{A}$ is said to achieve an adaptive variance ratio if the regret of the algorithm can be bounded by $\text{Regret}(T) \leq \widetilde{O}\left(\sqrt{V(\mu_{\max})KT}\right)$ where $V(\mu)$ is the variance of the reward distribution in $\mathcal{F}_m$ with mean parameter $\mu$; in this notation, $V(\mu_{\max}) = \text{Var}_{r\sim\nu_{\max}}[r]$ represents the variance of the reward distribution of the optimal arm. [2] Algorithms that achieve an adaptive variance ratio incorporate environment-specific parameters, enabling them to achieve tighter regret bounds tailored to different instances. For instance, in a Bernoulli environment, a regret upper bound of $\sqrt{V(\mu_{\max})KT}$ would be much smaller for MAB instances with favorable $\mu_{\max}$ values. Since $V(\mu_{\max}) = \mu_{\max}(1-\mu_{\max})$, this can be $\ll 1$ when $\mu_{\max}$ is close to 0 or 1. In this case, this regret bound can be significantly better than the usual $O(\sqrt{KT})$ regret bound.

In the literature, the maximum variance assumption has been used in some works (Jin et al., 2022; 2023; Ménard & Garivier, 2017) to derive finite-time instance-independent regret bounds, resulting in a nonadaptive variance ratio of $\bar{V}$. Qin et al. (2023) proves that KL-MS satisfies an adaptive variance ratio which utilizes the instance-specific parameter $V(\mu_{\max})$, which can be much better than $\bar{V}$. Qin et al. (2023) shows that kl-UCB (Cappé et al., 2013), kl-UCB++ (Ménard & Garivier, 2017) and UCB-V (Audibert et al., 2009b) can also achieve an adaptive variance ratio of $V(\mu_{\max})$ in the Bernoulli reward setting; in the general OPED

---

[2]$\widetilde{O}(\cdot)$ is a variant of the standard Big-O notation $O(\cdot)$ that hides logarithmic factors.

setting, the corresponding guarantee for kl-UCB++ is not established by the published analysis (the No* entry in Table 1), and the case of kl-UCB remains open.

### 3.4 Exponential family bandit algorithms with simultaneous adaptivity and optimality guarantees

Several bandit algorithms in the literature work in the setting of OPED reward distributions (Korda et al., 2013; Cappé et al., 2013; Ménard & Garivier, 2017). Thompson Sampling (TS) (Thompson, 1933) and kl-UCB (Cappé et al., 2013) were among the first to use posterior sampling and the optimism strategy, respectively. Both algorithms have been shown to satisfy asymptotic optimality in their original analyses.

More recently, Ménard & Garivier (2017) proposes kl-UCB++ and demonstrates that it satisfies asymptotic and minimax optimality simultaneously. Qin et al. (2023, Appendix F.1 and F.2) give refined analyses of kl-UCB and kl-UCB++, and show that they have variance-adaptive regret bounds of $O(\sqrt{KV(\mu_{\max})T\ln T})$ and $O(\sqrt{K^3 V(\mu_{\max})T\ln T})$, respectively; however, their results are restricted to the Bernoulli reward setting. Jin et al. (2022) proposes ExpTS and shows that it also achieves a logarithmic minimax ratio of $\sqrt{\ln(K)}$, and ExpTS$^+$ satisfies minimax optimality. However, both algorithms lack an analysis of the adaptive variance ratio. A detailed comparison with the works most related to ours is presented in Table 1.

## 4 Algorithms

We present our main algorithmic framework, GENERAL-EXP-KL-MS, in Algorithm 1. For the first $K$ time steps, the agent pulls each arm once. After the first $K$ time steps, the agent selects an arm $a$ based on the arm sampling distribution $(p_{t,a})_{a\in[K]}$, which is proportional to $\exp\left(-L(N_{t-1,a})\cdot D_{\mathrm{KL}}\left(\hat{\mu}_{t-1,a},\hat{\mu}_{t-1,\max}\right)\right)$, where $L(\cdot)$ is an inverse temperature function that satisfies $0 < L(k) \le k$ and increases monotonically with $k$. Every time the agent receives a reward $r_t$ from arm $a$, it updates the arm pulls count $N_{t,a}$ and the empirical mean estimate $\hat{\mu}_{t,a}$. The choice of the inverse temperature function $L(\cdot)$ affects the balance between exploration and

---

**Algorithm 1** GENERAL-EXP-KL-MS

> **Input:** $K \ge 2$
> **for** $t = 1, 2, \cdots, T$ **do**
>   **if** $t \le K$ **then**
>     Pull arm $I_t = t$ and observe reward $r_t \sim \nu_{I_t}$.
>   **else**
>     $p_{t,a} = \exp\left(-L(N_{t-1,a})D_{\mathrm{KL}}\left(\hat{\mu}_{t-1,a},\hat{\mu}_{t-1,\max}\right)\right)/M_t$, where
>     $M_t$ is a normalization factor, $M_t := \sum_{a=1}^{K}\exp\left(-L(N_{t-1,a})D_{\mathrm{KL}}\left(\hat{\mu}_{t-1,a},\hat{\mu}_{t-1,\max}\right)\right)$
>     Pull arm $I_t \sim p_t$ and observe reward $r_t \sim \nu_{I_t}$.
>   **end if**
> **end for**

---

exploitation. In our main algorithm (EXP-KL-MS), which is a specific instance of GENERAL-EXP-KL-MS, we choose $L(k) = k - 1$. In Appendix A, we also analyze variants of the algorithm where $L(k) = k$ and $L(k) = k/d$ for a constant $d > 1$. In an idealized setting where the estimation of the KL-divergence is accurate, the probability of pulling an arm $a$, denoted by $p_{t,a}$, is proportional to $\exp\left(-N_{t-1,a}\mathcal{D}(\nu_a, \nu_{\max})\right)$. Consequently, we expect the number of times arm $a$ is pulled to be approximately $\frac{\ln(T)}{\mathcal{D}(\nu_a,\nu_{\max})}$ over a total of $T$ time steps. This expectation aligns with the asymptotically optimal number of arm pulls for any consistent algorithm (Lai et al., 1985).

Algorithm 1 generalizes algorithms from prior works in the following way: if the reward distributions are Bernoulli, we set $L(k) = k$, and GENERAL-EXP-KL-MS becomes KL-MS (Qin et al., 2023); if the reward distributions are Gaussian with fixed variance, we also set $L(k) = k$, and GENERAL-EXP-KL-MS becomes MS (Bian & Jun, 2022; Maillard, 2011). In comparison to the MED algorithm (Honda & Takemura, 2011), GENERAL-EXP-KL-MS works in the OPED reward setting, while MED assumes that the reward distributions have finite supports.

## 5 Performance Guarantees

We focus on Algorithm 1 with $L(k) = k - 1$, which we abbreviate as EXP-KL-MS. Based on the main conclusion in Theorem 1, we conclude that Algorithm 1, EXP-KL-MS satisfies several key properties: a logarithmic minimax ratio of $\sqrt{\ln(K)}$ (Corollary 2), asymptotic optimality (Corollary 3), the Sub-UCB criterion (Corollary 4) and adaptive variance ratio (Corollary 5).

**Theorem 1.** *For any $K$-arm bandit problem with Assumptions 1 and 3, EXP-KL-MS (Algorithm 1) with $L(k) = k - 1$ has regret bounded as follows. For any $\Delta > 0$ and $c \in (0, \frac{1}{4}]$:*

$$
\begin{aligned}
\text{Regret}(T) \leq{} & T\Delta + \sum_{a \in [K]: \Delta_a > \Delta} \Delta_a \left( \frac{\ln\left(T D_{\text{KL}}\left(\mu_a + c\Delta_a, \mu_{\max} - c\Delta_a\right) \vee e\right)}{D_{\text{KL}}\left(\mu_a + c\Delta_a, \mu_{\max} - c\Delta_a\right)} \right) + \Delta_a \\
& + \sum_{a \in [K]: \Delta_a > \Delta} \Delta_a \left( \frac{1}{D_{\text{KL}}\left(\mu_a + c\Delta_a, \mu_{\max} - c\Delta_a\right)} + \frac{1}{D_{\text{KL}}\left(\mu_a + c\Delta_a, \mu_a\right)} \right) \\
& + \left( \sum_{a \in [K]: \Delta_a > \Delta} \Delta_a \left( \frac{1}{D_{\text{KL}}\left(\mu_{\max} - c\Delta_a, \mu_{\max}\right)} + \frac{1}{\left(D_{\text{KL}}\left(\mu_{\max} - c\Delta_a, \mu_{\max}\right)\right)^2} \right) \right) \wedge \\
& \left( \sum_{a \in [K]: \Delta_a > \Delta} \Delta_a \left( \frac{6 + 5e \ln(T D_{\text{KL}}\left(\mu_{\max} - c\Delta_a, \mu_{\max}\right) \vee e)}{D_{\text{KL}}\left(\mu_{\max} - c\Delta_a, \mu_{\max}\right)} \right) \right)
\end{aligned}
\tag{4}
$$

The first term, $T\Delta$, accounts for cases when arms $a$ with $\Delta_a \leq \Delta$ are pulled. The remaining terms account for arms with larger $\Delta_a$. The second term is the most significant, playing a key role in both the asymptotic and finite-time analyses. As $T \to \infty$ and $\Delta \to 0$, dividing both sides of the regret bound by $\ln(T)$ reveals that only the second term contributes to the regret upper bound. The third and fourth terms are lower-order compared to the second term. The last term notably consists of the two parts. The first one is used to prove asymptotic optimality because it is independent of $T$. However, relying on this part would result in a suboptimal minimax regret of $\widetilde{O}(T^{3/4})$. The second term is included to address this issue and is of the order $\widetilde{O}\left(\sum_{a \in [K]: \Delta_a > \Delta} \frac{\bar{V}}{\Delta_a}\right)$, enabling a sharp minimax optimality and Sub-UCB analysis. Combining Theorem 1 with Lemma 25, a generalized Pinsker Inequality that lower bounds the KL-divergence between two distributions in $\mathcal{F}_m$, we have:

**Corollary 2** (Logarithmic Minimax Ratio). *For any $K$-arm bandit problem with Assumptions 1 to 3, EXP-KL-MS has:*

$$
\text{Regret}(T) \leq O\left(\sqrt{\bar{V} K T \ln(K)} + \sum_{a \in [K]} \Delta_a\right).
$$

**Corollary 3** (Asymptotic Optimality). *For any $K$-arm bandit problem with Assumptions 1 to 3, EXP-KL-MS satisfies that:*

$$
\limsup_{T \to \infty} \frac{\text{Regret}(T)}{\ln(T)} = \sum_{a \in [K]: \Delta_a > 0} \frac{\Delta_a}{D_{\text{KL}}\left(\mu_a, \mu_{\max}\right)}.
$$

**Corollary 4** (Sub-UCB). *For any $K$-arm bandit problem with Assumptions 1 to 3, EXP-KL-MS satisfies that*

$$
\text{Regret}(T) \leq O\left( \sum_{a: \Delta_a > 0} \frac{\bar{V} \ln(T)}{\Delta_a} + \Delta_a \right).
$$

Corollary 2 establishes a $\sqrt{\ln(K)}$ factor in the minimax ratio. Corollary 3 shows that EXP-KL-MS satisfies asymptotic optimality, ensuring that the long-term performance of EXP-KL-MS is guaranteed to be optimal. Corollary 4 demonstrates that EXP-KL-MS satisfies the Sub-UCB criterion, ensuring that it will never perform worse than the regret bound achieved by UCB algorithms in the finite-time regime. The choice that $L(k) = k - 1$ in EXP-KL-MS is crucial, as other choices of $L(\cdot)$ may cause parts of the regret bound given by Equation (4) to no longer hold, resulting in failing to achieve asymptotic optimality or a sharp $\sqrt{\ln(K)}$

minimax ratio. In particular, picking the slightly larger $L(k) = k$ would already break our current proof of asymptotic optimality; we defer the detailed discussion of the technical reason to Appendix D.4, where we show that a naive extension of the analysis of Qin et al. (2023) in the Bernoulli setting yields a vacuous bound on the "optimal arm underestimated" term for $L(k) = k$ in general OPED settings. For comparison, we present results for other choices of $L(k)$ in Appendix A.

**On achieving constant minimax ratio and Sub-UCB simultaneously.** To the best of our knowledge, the only algorithm that simultaneously achieves a constant minimax ratio and Sub-UCB is ADA-UCB (Lattimore, 2018), for Gaussian rewards, which uses a meticulously crafted confidence bound. We believe that the $\sqrt{\ln(K)}$ minimax ratio is tight for our algorithm, Exp-KL-MS. Modifying our algorithm to achieve both guarantees simultaneously remains an interesting open question. Specifically, the data-dependent confidence-level adjustment of ADA-UCB (Lattimore, 2018) constructs its confidence width at arm $a$ based on a data-dependent quantity $H_a(t-1)$ that couples across arms, and no other route to a constant minimax ratio with Sub-UCB is currently known. Translating this to our Maillard-sampling framework, the term that enters $\log(\cdot)$ in a UCB index corresponds to a multiplicative factor in front of $\exp(\cdot)$ in our sampling probability. We therefore speculate that multiplying $1/H$ (with $H$ as defined in Lattimore (2018)) in front of the $\exp(\cdot)$ in our sampling rule may enable us to remove the $\sqrt{\ln K}$ factor while preserving Sub-UCB. The corresponding analysis, however, is substantially more involved, and we leave it as future work. Next, we introduce a key assumption that allows us to derive the adaptive variance ratio.

**Assumption 4.** *For any bandit instance from a known distribution family $\mathcal{F}_m(M)$ with a Variance function $V(\cdot)$, there exists $C_L > 0$ such that $\forall \mu, \mu' \in (0, M], |V(\mu) - V(\mu')| \leq C_L |\mu - \mu'|$.*

Assumption 4 also covers a large set of OPED families as we mentioned in Table 2, including Bernoulli, Poisson, Gaussian with fixed variance, Gamma with fixed shape parameter $k$, and Inverse Gaussian distributions. For instance, the Gamma distribution with fixed shape parameter $k$ has mean $k\theta$ and variance $k\theta^2$, so its variance function is $V(\mu) = \mu^2/k$. Within the interval $[0, M]$, $V(x)$ satisfies Assumption 4 with Lipschitz constant $2M/k$. In real world applications, it is reasonable to expect that the mean number of user clicks (Alves et al., 2021) and mean revenue (Jia et al., 2021) to be within a reasonable range, and thus we believe that Assumption 4 is practical.

In the refined version of Theorem 1, we can replace $\bar{V}$ by $V(\mu_{\max})$ and show Exp-KL-MS has adaptive variance ratio in Corollary 5.

**Corollary 5** (Adaptive Variance Ratio). *For any $K$-arm bandit problem with Assumptions 1, 3 and 4, Exp-KL-MS has:*
$$\text{Regret}(T) \leq O\big(\sqrt{V(\mu_{\max})KT\ln(K)} + K\ln(T)\big).$$

Corollary 5 shows that Exp-KL-MS satisfies both a logarithmic minimax ratio of $\sqrt{\ln(K)}$ and an adaptive variance ratio. Such an adaptive regret bound is no worse than Corollary 2 and can be much better when $V(\mu_{\max}) \ll \bar{V}$.

**Numerical Experiments.** We compare the regret of our algorithm Exp-KL-MS against the state-of-the-art algorithms, including kl-UCB (Cappé et al., 2013), ExpTS (Jin et al., 2022) and kl-UCB++ (Ménard & Garivier, 2017). The reward environments are borrowed from Kaufmann et al. (2012). The original experiment has two environments, one has a mean vector $[0.20, 0.25]$, and the other has a mean vector $[0.80, 0.90]$ and both have Bernoulli reward distributions. We replace the Bernoulli distributions with Gaussian, Gamma, and Poisson distributions with the same mean vectors. We plot the average pseudo reward of all algorithms against time steps in Figure 3 and the average pseudo regret in Figure 4 in Appendix F. We also include another experiment with different $L(\cdot)$ choices in Appendix F and result can be found in Figures 5 and 6. In Appendix F.3 and Appendix F.4, we additionally report two scaling experiments that verify the predictions of Corollary 2 and Corollary 5 directly: a $(K, T)$-varying study that evaluates Exp-KL-MS on instances with suboptimal arms having gaps $\Delta = \sqrt{K/T}$ that exhibits the minimax regret bound (results in Figures 7 to 9), and a $V(\mu_1)$-varying study that evaluates Exp-KL-MS on instances with suboptimal arms having gaps $\Delta = \sqrt{V(\mu_1)K/T}$ (results in Figure 10), across five OPED reward families. The empirical slopes match the predicted $\sqrt{KT\ln K}$ and $\sqrt{V(\mu_{\max})KT\ln K}$ rates, respectively.

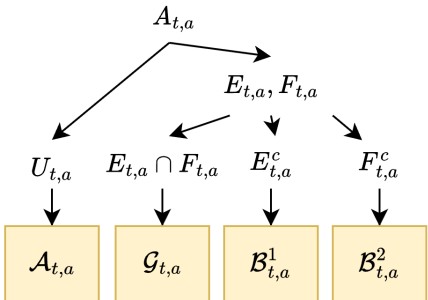

Figure 1: Case splitting of our regret analysis.

## 6 Proof Sketch

This section outlines the proof of Theorem 1. The regret of a bandit algorithm can be rewritten as the product between the reward gap of arm $a$ and the expected number of times arm $a$ is pulled over the arm space: $\text{Regret}(T) = \mathbb{E}\left[\sum_{t=1}^{T} \Delta_{I_t}\right] = \sum_{a \in [K]} \Delta_a \mathbb{E}[N_{T,a}]$. Since we know $\mathbb{E}[N_{T,a}] = \sum_{t=1}^{T} \mathbb{E}[\mathbf{1}_{I_t=a}]$, it suffices to upper bound $\mathbb{E}[N_{T,a}]$ for each suboptimal arm $a$. At each time step $t$, the probability of pulling arm $a$ can be decomposed into four disjoint cases, which we will introduce shortly. We first define $\varepsilon_{1,a}, \varepsilon_{2,a} > 0$ such that $\forall a \in [K], \varepsilon_{1,a} + \varepsilon_{2,a} < \Delta_a$ and the four disjoint cases are defined based on these two parameters. We give a brief explanation of the four basic cases below.

- $A_{t,a} \coloneqq \{I_t = a\}$ represents the base event that arm $a$ is pulled at time step $t$.

- $U_{t,a} \coloneqq \{N_{t,a} < u_a + 1\}$ represents the event that the number of samples of arm $a$ is less than a threshold $u_a \coloneqq \frac{\ln(T D_{\text{KL}}(\mu_a + \varepsilon_{1,a}, \mu_{\max} - \varepsilon_{2,a}) \vee e)}{D_{\text{KL}}(\mu_a + \varepsilon_{1,a}, \mu_{\max} - \varepsilon_{2,a})}$

- $E_{t,a} \coloneqq \{\hat{\mu}_{t,a} \leq \mu_a + \varepsilon_{1,a}\}$ represents the event that the expected reward of the suboptimal arm $a$ is not overestimated by more than $\varepsilon_{1,a}$.

- $F_{t,a} \coloneqq \{\hat{\mu}_{t,\max} \geq \mu_{\max} - \varepsilon_{2,a}\}$ represents the event that the best arm's expected reward is not underestimated by more than $\varepsilon_{2,a}$.

We combine the above four basic events to form four different cases that can capture all possible scenarios of pulling arm $a$ at time step $t$, and give a graphical illustration in Figure 1.

- $\mathcal{A}_{t,a} \coloneqq A_{t,a} \cap U_{t,a}$: arm $a$ is pulled and the number of samples of arm $a$ is below $u_a$.

- $\mathcal{G}_{t,a} \coloneqq A_{t,a} \cap U_{t,a}^c \cap E_{t,a} \cap F_{t,a}$: arm $a$ is pulled, the number of samples of arm $a$ is above $u_a$, and both the expected reward of arm $a$ and the optimal arm are accurately estimated.

- $\mathcal{B}_{t,a}^1 \coloneqq A_{t,a} \cap E_{t,a}^c$: arm $a$ is pulled and the expected reward of arm $a$ is overestimated.

- $\mathcal{B}_{t,a}^2 \coloneqq A_{t,a} \cap F_{t,a}^c$: arm $a$ is pulled and the expected reward of the optimal arm is underestimated.

After defining the above events, we can express the probability of pulling arm $a$ at time step $t$ as:

$$\mathbb{E}[\mathbf{1}_{I_t=a}] \leq \mathbb{E}[\mathbf{1}_{\mathcal{A}_{t,a}}] + \mathbb{E}[\mathbf{1}_{\mathcal{G}_{t,a}}] + \mathbb{E}[\mathbf{1}_{\mathcal{B}_{t,a}^1}] + \mathbb{E}[\mathbf{1}_{\mathcal{B}_{t,a}^2}]$$

By summing over $t = 1, \ldots, T$ and grouping the four cases accordingly, we can decompose the $\mathbb{E}[N_{T,a}]$ into four terms. Formally, we bound $\mathbb{E}[N_{T,a}]$ as follows:

$$\mathbb{E}[N_{T,a}] \leq \underbrace{\sum_{t=1}^{T} \mathbb{E}[\mathbf{1}_{\mathcal{A}_{t,a}}]}_{=: \mathcal{A}_a} + \underbrace{\sum_{t=1}^{T} \mathbb{E}[\mathbf{1}_{\mathcal{G}_{t,a}}]}_{=: \mathcal{G}_a} + \underbrace{\sum_{t=1}^{T} \mathbb{E}[\mathbf{1}_{\mathcal{B}_{t,a}^1}]}_{=: \mathcal{B}_a^1} + \underbrace{\sum_{t=1}^{T} \mathbb{E}[\mathbf{1}_{\mathcal{B}_{t,a}^2}]}_{=: \mathcal{B}_a^2}$$

$\mathcal{A}_a$ handles situations where the number of pulls of arm $a$ is below a threshold $u_a$, which can be easily controlled by the threshold $u_a$. The remaining three terms focus on scenarios where there are enough pulls to accurately estimate arm $a$'s expected reward with high probability. $\mathcal{G}_a$ addresses the case when the reward estimations of the suboptimal arm $a$ and the optimal arm are accurate. Since the estimation of arm $a$, $\hat{\mu}_{t,a}$, is at least $\Delta_a - \varepsilon_{1,a} - \varepsilon_{2,a}$ away from $\hat{\mu}_{t,\max}$, the expected number of pulls to arm $a$ is upper bounded by $\frac{1}{D_{\mathrm{KL}}(\mu_a + \varepsilon_{1,a}, \mu_{\max} - \varepsilon_{2,a})}$, as shown by a straightforward calculation (See Appendix D.1, Proof of Proposition 15). $\mathcal{B}_a^1$ represents the case where the expected reward of arm $a$ is overestimated. Using the well-known Chernoff tail bound (Lemma 26), we can control the probability of pulling arm $a$ by $\exp\left(-kD_{\mathrm{KL}}\left(\mu_a + \varepsilon_{1,a}, \mu_a\right)\right)$ for each count of pulls to arm $a$, where $k = 1, \ldots, T$. The sum of these probabilities is bounded by $\frac{1}{D_{\mathrm{KL}}(\mu_a + \varepsilon_{1,a}, \mu_a)}$ (See Appendix D.2, Proof of Proposition 16). $\mathcal{B}_a^2$ represents the case when the expected reward of the optimal arm is underestimated, and arm $a$ is selected. One straightforward way to bound it is by applying the same Chernoff bound used for bounding $\mathcal{B}_a^1$, utilizing $L(k) = k - 1$, which leads to the following bound in Equation (5):

$$\mathcal{B}_a^2 \leq \frac{1}{D_{\mathrm{KL}}\left(\mu_{\max} - \varepsilon_{2,a}, \mu_{\max}\right)} + \frac{1}{\left(D_{\mathrm{KL}}\left(\mu_{\max} - \varepsilon_{2,a}, \mu_{\max}\right)\right)^2} \tag{5}$$

The second term, and an inverse-KL-square term arises due to the differing conditions between $\mathcal{B}_a^1$ and $\mathcal{B}_a^2$: while both bound the estimated mean away from the ground truth, $\mathcal{B}_a^1$ pertains to the same arm being pulled, whereas $\mathcal{B}_a^2$ involves a different arm, introducing an additional squared KL term. This bound is useful to establish asymptotic optimality since when $T \to \infty$, with proper settings of $\varepsilon_{2,a}$ the RHS of the Equation (5) becomes $o(\ln T)$. However, relying on bound (5) alone would result in an undesirable $O(T^{3/4})$ minimax regret bound. To prove that Exp-KL-MS satisfies $\sqrt{\ln(K)}$ minimax ratio and Sub-UCB, we derive another upper bound for $\mathcal{B}_a^2$:

$$\mathcal{B}_a^2 \leq \frac{\ln(TD_{\mathrm{KL}}\left(\mu_{\max} - \varepsilon_{2,a}, \mu_{\max}\right) \vee e)}{D_{\mathrm{KL}}\left(\mu_{\max} - \varepsilon_{2,a}, \mu_{\max}\right)}. \tag{6}$$

Equation (6) contains only $D_{\mathrm{KL}}\left(\mu_{\max} - \varepsilon_{2,a}, \mu_{\max}\right)$ in the denominator, which guarantees a $\sqrt{\ln(K)}$ minimax ratio and Sub-UCB.

Without loss of generality, we assume arm 1 is the optimal arm in the following proof sketch, which means $\mu_1 = \mu_{\max}$. Let $\hat{\mu}_{(k),1}$ denote the empirical mean estimator of $\mu_1$ after $k$ observations. Notice that we change the subscript of $\hat{\mu}$ from $t$ to $(k)$, representing the empirical mean reward from the optimal arm's first $k$ pulls at time step $t$. We notice that event $\mathcal{B}_{t,a}^2$ happening implies that the optimal arm 1 is also underestimated at time step $t$, and $\hat{\mu}_{t,1} \leq \mu_{\max} - \varepsilon_{2,a}$. Then we construct a series of high-probability "clean" events using a sequence of thresholds $\boldsymbol{\alpha} := \{\alpha_k\}_{k=1}^T$ to lower bound $\hat{\mu}_{(k),1}$ for each $k = 1, \ldots, T$. The "clean" event is defined as $\mathcal{E}_k(\alpha_k) := \{\alpha_k \leq \hat{\mu}_{(k),1}\}$ and $\mathcal{E}(\boldsymbol{\alpha}) := \cap_{1 \leq k \leq T} \mathcal{E}_k(\alpha_k)$, the event that all $\hat{\mu}_{(k),1}$ from $k = 1$ to $T$ are lower bounded by $\alpha_k$. By selecting $\alpha_k$ to satisfy the condition $D_{\mathrm{KL}}\left(\hat{\mu}_{(k),1}, \mu_{\max} - \varepsilon_2\right) \leq \frac{2\ln(T/k)}{k}$ and plugging it into Lemma 19, we derive Equation (6), which is useful for proving the $\sqrt{\ln(K)}$ minimax ratio and Sub-UCB. Such condition on $\alpha_k$ ensures that $\hat{\mu}_{(k),1}$ is not too far away from $\mu_{\max} - \varepsilon_{2,a}$ measured by KL divergence.

**On the challenge of establishing asymptotic optimality for** $L(k) = k$**.** As mentioned in the Introduction, a natural idea is to analyze the properties of Algorithm 1 with $L(k) = k$. While we show in Appendix A that such a choice satisfies adaptive minimax ratio and Sub-UCB, we did not succeed in showing its asymptotic optimality. Towards proving asymptotic optimality, we aim to bound $\mathcal{B}_a^2$ by $\frac{1}{D_{\mathrm{KL}}(\mu_{\max} - \varepsilon_{2,a}, \mu_{\max})}$ plus some lower order terms. However, a naive extension of the previous proof techniques (Qin et al., 2023) bounds $\mathcal{B}_a^2$ by the integral $\mathbb{E}\left[\mathbf{1}_{\hat{\mu}_{(k),1} \leq \mu_{\max} - \varepsilon_{2,a}} \cdot \exp\left(D_{\mathrm{KL}}\left(\hat{\mu}_{(k),1}, \mu_{\max}\right)\right)\right]$, which diverges when the reward distribution is e.g., exponential. We conjecture that this may not be an artifact of our analysis, but rather a fundamental limitation of Exp-KL-MS with $L(k) = k$.

# 7 Conclusions

In this paper, we introduce the General-Exp-KL-MS algorithm, which works for OPED families of the form Equation (1) and we prove that when the inverse temperature function is set to $L(k) = k - 1$, Exp-KL-MS

achieves a minimax ratio of $\sqrt{\ln(K)}$, asymptotic optimality, adaptive variance, and the Sub-UCB criterion at the same time. Our algorithm, EXP-KL-MS, is the first to simultaneously achieve all four guarantees in the general exponential family reward setting. Moreover, to the best of our knowledge, it is the only algorithm that maintains a closed-form expression of the arm sampling distribution at each round for OPED reward distributions, which is amenable to offline evaluation (see e.g., Qin et al., 2023, Figure 1). Additionally, we identify a new and natural Lipschitz assumption on the variance function, and give refined bounds on the number of arm pulls using KL divergences. Prior results on adaptive variance ratio bounds were limited to the Bernoulli reward setting (Qin et al., 2023).

An interesting direction for future work is to generalize the result to OPED families with general sufficient statistics $T$ beyond the identity function $T(x) = x$, such as the Beta distribution, and to demonstrate that GENERAL-EXP-KL-MS can still satisfy all the criteria. One immediate idea is to apply transformation $T$ to the rewards to reduce the problem to OPED bandits with identity sufficient statistics. Although Baudry et al. (2023) presents a more general assumption on the reward distribution, a finite-time regret bound has not been established in their work.

Second, we believe that EXP-KL-MS has significant potential in the contextual bandit problem. We hope that the techniques developed in EXP-KL-MS can be applied to design adaptive and optimal algorithms for generalized linear bandit problems with reward distributions belonging to exponential families (Filippi et al., 2010). Progress in this direction has been madefor the bounded reward (Lee et al., 2021) and the Gaussian reward setting (Balagopalan & Jun, 2024). Finally, we believe that Maillard sampling style algorithms can be designed for pure exploration problems, such as best-arm identification, either with a prescribed confidence using as few samples as possible (fixed-confidence) (Jourdan et al., 2022) or within a given sample budget (fixed-budget). We leave the design of such an algorithm as an open direction. One plausible construction, loosely inspired by top-two sampling (Jourdan et al., 2022), is to assign probability $1/2$ to the empirical best arm and distribute the remaining $1/2$ over the other arms proportionally to $\exp\left(-L(N_{t-1,a})D_{\mathrm{KL}}\left(\hat{\mu}_{t-1,a}, \hat{\mu}_{t-1,\max}\right)\right)$. This preserves a closed-form sampling distribution while sampling the empirical best arm at a constant rate.

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

## Contents

## A  Extensions

In this section, we present the results from other choices of function $L(k)$ and demonstrate that they also satisfy various desirable properties. Here, considering the overall restriction on $L(k)$ is $0 < L(k) \le k$, we pick two examples, $L(k) = k/d$ where $d > 1$ and $L(k) = k$.

### A.1  Extension 1: $L(k) = k/d$, $d > 1$

In this case, the inverse temperature function imposed by the number of arm pulls is attenuated by a constant factor $d$.

**Corollary 6.** (Logarithmic Minimax Ratio and Adaptive Variance Ratio) *For any $K$-arm bandit problem with Assumptions 1, 3 and 4, when $d > 1$ GENERAL-EXP-KL-MS with $L(k) = k/d$ has regret:*

$$\text{Regret}(T) \le O\big(\sqrt{V(\mu_{\max})KT\ln(K)}\big) + O\big(K\ln(T)\big).$$

**Corollary 7.** (Sub-UCB criterion) *For any $K$-arm bandit problem with Assumptions 1, 3 and 4, when $d > 1$ GENERAL-EXP-KL-MS with $L(k) = k/d$ satisfies Sub-UCB criterion, which means that its regret is bounded by*

$$\text{Regret}(T) \le O\Big( \sum_{a:\Delta_a > 0} \frac{\ln(T)}{\Delta_a} + \Delta_a \Big).$$

The above Corollaries show that GENERAL-EXP-KL-MS with $L(k) = k/d$ can have the same minimax ratio as EXP-KL-MS, adaptive variance ratio, and Sub-UCB. However, since the newly introduced additional factor

$d$, it violates the asymptotic optimality, resulting in a constant factor difference compared to EXP-KL-MS in the asymptotic performance.

### A.2 Extension 2: $L(k) = k$

GENERAL-EXP-KL-MS is the same as KL-MS when $L(k) = k$. Based on the current proof framework, we can only show that GENERAL-EXP-KL-MS with $L(k) = k$ satisfies an adaptive variance ratio and has a minimax ratio as $\ln(T)$.

**Corollary 8.** (Logarithmic Minimax Ratio and Adaptive Variance Ratio) *For any $K$-arm bandit problem with Assumptions 1, 3 and 4, GENERAL-EXP-KL-MS with $L(k) = k$ has regret bounded as:*

$$\text{Regret}(T) \leq O\big(\sqrt{V(\mu_{\max})KT\ln(T)}\big) + O\big(K\ln(T)\big).$$

**Corollary 9.** (Sub-UCB criterion) *For any $K$-arm bandit problem with Assumptions 1, 3 and 4, GENERAL-EXP-KL-MS with $L(k) = k$ satisfies the Sub-UCB criterion and has regret bounded as*

$$\text{Regret}(T) \leq O\big( \sum_{a:\Delta_a>0} \frac{\ln(T)}{\Delta_a} + \Delta_a\big).$$

## B Proof of Main Theorem (Theorem 1)

Before presenting the details, we outline our proof roadmap in Figure 2. We divide the proof into three phases, moving from left to right. Appendix B consists of our main conclusion when $L(k) = k - 1$, Theorem 1 and its direct consequence, Corollary 10 (without Lipschitzness Assumption 4) and Corollary 11 (with Lipschitzness Assumption 4). Appendix C contains all results from other choices of inverse temperature function $L(k)$. We include the case where $L(k) = k/d$ in Theorem 12 and the case where $L(k) = k$ in Theorem 13. Appendix D includes all propositions which are used to prove Theorem 1 (when $L(k) = k - 1$), Theorem 12 (when $L(k) = k/d$), and Theorem 13 (when $L(k) = k$). All proofs of the proposition are also provided in Appendix D. Appendix E includes all auxiliary lemmas used to prove propositions in our analysis, as well as a KL-lower bound lemma (Lemma 25).

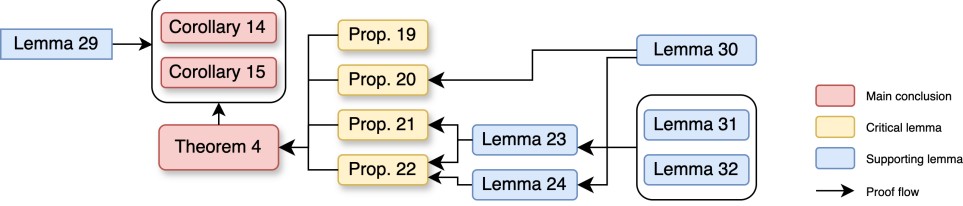

Figure 2: Roadmap of the proof of Theorem 1

### B.1 Proof of Theorem 1

In this section, we focus on the left half of the proof and show the proof of Theorem 1. Recall that we have decomposed the regret into four terms:

$$\text{Regret} \leq \sum_{a\in[K]:\Delta_a>0} \Delta_a \left(\mathcal{A}_a + \mathcal{G}_a + \mathcal{B}_a^1 + \mathcal{B}_a^2\right)$$

The proof of Theorem 1 follows straightforwardly from applying Propositions 15 to 18. $\mathcal{A}_a$ is bounded by the threshold through a trivial analysis. $\mathcal{G}_a$ is bounded by using Proposition 15. $\mathcal{B}_a^1$ is bounded by Proposition 16 and $\mathcal{B}_a^2$ is bounded by the minimum among results from Propositions 17 and 18.

Here, to remind us, we restate our main conclusion (Theorem 1):

**Theorem 1.** *For any K-arm bandit problem with Assumptions 1 and 3, EXP-KL-MS (Algorithm 1) with* $L(k) = k - 1$ *has regret bounded as follows. For any* $\Delta > 0$ *and* $c \in (0, \frac{1}{4}]$:

$$\text{Regret}(T) \leq T\Delta + \sum_{a \in [K]:\Delta_a > \Delta} \Delta_a \left( \frac{\ln \left( T D_{\text{KL}} \left( \mu_a + c\Delta_a, \mu_{\max} - c\Delta_a \right) \vee e \right)}{D_{\text{KL}} \left( \mu_a + c\Delta_a, \mu_{\max} - c\Delta_a \right)} \right) + \Delta_a$$

$$+ \sum_{a \in [K]:\Delta_a > \Delta} \Delta_a \left( \frac{1}{D_{\text{KL}} \left( \mu_a + c\Delta_a, \mu_{\max} - c\Delta_a \right)} + \frac{1}{D_{\text{KL}} \left( \mu_a + c\Delta_a, \mu_a \right)} \right)$$

$$+ \left( \sum_{a \in [K]:\Delta_a > \Delta} \Delta_a \left( \frac{1}{D_{\text{KL}} \left( \mu_{\max} - c\Delta_a, \mu_{\max} \right)} + \frac{1}{\left( D_{\text{KL}} \left( \mu_{\max} - c\Delta_a, \mu_{\max} \right) \right)^2} \right) \right) \wedge$$

$$\left( \sum_{a \in [K]:\Delta_a > \Delta} \Delta_a \left( \frac{6 + 5e \ln(T D_{\text{KL}} \left( \mu_{\max} - c\Delta_a, \mu_{\max} \right) \vee e)}{D_{\text{KL}} \left( \mu_{\max} - c\Delta_a, \mu_{\max} \right)} \right) \right) \tag{4}$$

*Proof of Theorem 1.* Recall the proof sketch we mentioned in Section 6, for each arm $a$ such that $\Delta_a > \Delta$, we divide the event of pulling a suboptimal arm into four subevents at each time step. Recall the definition of terms $\mathcal{A}_{t,a}, \mathcal{G}_{t,a}, \mathcal{B}^1_{t,a}$ and $\mathcal{B}^2_{t,a}$:

$$\mathcal{A}_a = \sum_{t=1}^{T} \mathbb{E}\left[ \mathbf{1}_{\mathcal{A}_{t,a}} \right] = \sum_{t=1}^{T} \mathbb{E}\left[ \mathbf{1}_{A_{t,a} \cap U_{t,a}} \right]$$

$$\mathcal{G}_a = \sum_{t=1}^{T} \mathbb{E}\left[ \mathbf{1}_{\mathcal{G}_{t,a}} \right] = \sum_{t=1}^{T} \mathbb{E}\left[ \mathbf{1}_{A_{t,a} \cap U^c_{t,a} \cap E_{t,a} \cap F_{t,a}} \right]$$

$$\mathcal{B}^1_a = \sum_{t=1}^{T} \mathbb{E}\left[ \mathbf{1}_{\mathcal{B}^1_{t,a}} \right] = \sum_{t=1}^{T} \mathbb{E}\left[ \mathbf{1}_{A_{t,a} \cap E^c_{t,a}} \right]$$

$$\mathcal{B}^2_a = \sum_{t=1}^{T} \mathbb{E}\left[ \mathbf{1}_{\mathcal{B}^2_{t,a}} \right] = \sum_{t=1}^{T} \mathbb{E}\left[ \mathbf{1}_{A_{t,a} \cap F^c_{t,a}} \right]$$

Based on the above split cases, we can decompose the regret as follows:

$$\text{Regret}(T) = \sum_{a \in [K]:\Delta_a \leq \Delta} \Delta_a \mathbb{E}[N_{T,a}] + \sum_{a \in [K]:\Delta_a > \Delta} \Delta_a \mathbb{E}[N_{T,a}]$$

$$\leq T\Delta + \sum_{a \in [K]:\Delta_a > \Delta} \Delta_a \mathbb{E}[N_{T,a}]$$

$$\leq T\Delta + \sum_{a \in [K]:\Delta_a > \Delta} \Delta_a \left( \mathcal{A}_a + \mathcal{G}_a + \mathcal{B}^1_a + \mathcal{B}^2_a \right)$$

$$\leq T\Delta + \sum_{a \in [K]:\Delta_a > \Delta} \Delta_a \left( u_a + \mathcal{G}_a + \mathcal{B}^1_a + \mathcal{B}^2_a \right)$$

In the first inequality, we bound the regret incurred by steps where an arm with $\Delta_a$ smaller than $\Delta$ is pulled by $\Delta$, noting that there are at most $T$ steps in total. In the second inequality, we bound the first summation term by $T\Delta$. In the third inequality, we decompose the regret from pulling arm $a$ into $\mathcal{A}_a, \mathcal{G}_a, \mathcal{B}^1_a$, and $\mathcal{B}^2_a$. In the last inequality, we bound $\mathcal{A}_a$ by $u_a$, since the event $U_{t,a}$ restricts the number of times arm $a$ can be pulled to at most $u_a$ times.

Therefore, for each arm $a$, we need to apply Propositions 15 to 18 to bound $\mathcal{G}_{t,a}, \mathcal{B}^1_{t,a}$ and $\mathcal{B}^2_{t,a}$, respectively. Recall the inverse temperature function $L(k) = k - 1$ and the choice of $u_a = \frac{\ln(T D_{\text{KL}}(\mu_a + \varepsilon_{1,a}, \mu_{\max} - \varepsilon_{2,a}) \vee e)}{D_{\text{KL}}(\mu_a + \varepsilon_{1,a}, \mu_{\max} - \varepsilon_{2,a})}$, we have

- 

$$\mathcal{G}_a \leq T \exp\left(-L(u_a)D_{\mathrm{KL}}\left(\mu_a + \varepsilon_{1,a}, \mu_{\max} - \varepsilon_{2,a}\right)\right) \leq \frac{1}{D_{\mathrm{KL}}\left(\mu_a + \varepsilon_{1,a}, \mu_{\max} - \varepsilon_{2,a}\right)} \quad \text{(Proposition 15)}$$

- 

$$\mathcal{B}_a^1 \leq \frac{1}{D_{\mathrm{KL}}\left(\mu_a + \varepsilon_{1,a}, \mu_a\right)} \quad \text{(Proposition 16)}$$

- 

$$\mathcal{B}_a^2 \leq \frac{1}{D_{\mathrm{KL}}\left(\mu_{\max} - \varepsilon_{2,a}, \mu_{\max}\right)} + \sum_{k=1}^{T} \frac{L(k)}{k - L(k)} e^{-kD_{\mathrm{KL}}(\mu_{\max} - \varepsilon_{2,a}, \mu_{\max})} \quad \text{(Proposition 17)}$$

$$= \frac{1}{D_{\mathrm{KL}}\left(\mu_{\max} - \varepsilon_{2,a}, \mu_{\max}\right)} + \sum_{k=1}^{T} (k-1)\exp\left(-kD_{\mathrm{KL}}\left(\mu_{\max} - \varepsilon_{2,a}, \mu_{\max}\right)\right) \quad (L(k) = k - 1)$$

$$\leq \frac{1}{D_{\mathrm{KL}}\left(\mu_{\max} - \varepsilon_{2,a}, \mu_{\max}\right)} + \frac{1}{\left(\exp\left(D_{\mathrm{KL}}\left(\mu_{\max} - \varepsilon_{2,a}, \mu_{\max}\right)\right) - 1\right)^2}$$

$$\text{(Sum the second term and let } T \to \infty)$$

$$\leq \frac{1}{D_{\mathrm{KL}}\left(\mu_{\max} - \varepsilon_{2,a}, \mu_{\max}\right)} + \frac{1}{\left(D_{\mathrm{KL}}\left(\mu_{\max} - \varepsilon_{2,a}, \mu_{\max}\right)\right)^2}$$

The second inequality holds because the sum of the first $T$ terms of the series $\sum_{k=1}^{T} (k - 1)\exp\left(-kD_{\mathrm{KL}}\left(\mu_{\max} - \varepsilon_{2,a}, \mu_{\max}\right)\right)$ is less than $\sum_{k=1}^{\infty} (k-1)\exp\left(-kD_{\mathrm{KL}}\left(\mu_{\max} - \varepsilon_{2,a}, \mu_{\max}\right)\right)$, which converges to $\frac{1}{(\exp(D_{\mathrm{KL}}(\mu_{\max} - \varepsilon_{2,a}, \mu_{\max})) - 1)^2}$ since $D_{\mathrm{KL}}\left(\mu_{\max} - \varepsilon_{2,a}, \mu_{\max}\right)$ is nonnegative.

- 

$$\mathcal{B}_a^2 \leq \frac{6}{D_{\mathrm{KL}}\left(\mu_{\max} - \varepsilon_{2,a}, \mu_{\max}\right)} + \sum_{k=1}^{T} \frac{e\,L(k)}{k} \exp(-kD_{\mathrm{KL}}\left(\mu_{\max} - \varepsilon_{2,a}, \mu_{\max}\right)) \cdot \ln(T/k)$$

$$\text{(Proposition 18)}$$

$$\leq \frac{6}{D_{\mathrm{KL}}\left(\mu_{\max} - \varepsilon_{2,a}, \mu_{\max}\right)} + e \sum_{k=1}^{T} \exp\left(-kD_{\mathrm{KL}}\left(\mu_{\max} - \varepsilon_{2,a}, \mu_{\max}\right)\right) \cdot \ln(T/k) \quad (L(k) = k - 1)$$

$$\leq \frac{6}{D_{\mathrm{KL}}\left(\mu_{\max} - \varepsilon_{2,a}, \mu_{\max}\right)} + \frac{5e \ln(TD_{\mathrm{KL}}\left(\mu_{\max} - \varepsilon_{2,a}, \mu_{\max}\right) \vee e)}{D_{\mathrm{KL}}\left(\mu_{\max} - \varepsilon_{2,a}, \mu_{\max}\right)} \quad \text{(Lemma 27)}$$

$$\leq \frac{(6 + 5e) \ln(TD_{\mathrm{KL}}\left(\mu_{\max} - \varepsilon_{2,a}, \mu_{\max}\right) \vee e)}{D_{\mathrm{KL}}\left(\mu_{\max} - \varepsilon_{2,a}, \mu_{\max}\right)}$$

For each arm, we let $\varepsilon_{1,a} = \varepsilon_{2,a} = c\Delta_a$ and combine these inequalities, we will obtain the upper bound shown in the Equation (4). $\qquad\square$

## B.2 Proofs of Immediate Corollaries of Theorem 1

Starting from Theorem 1, we utilize a lower bound lemma for the KL divergence (Lemma 25) to derive two immediate results (Corollaries 10 and 11) under different assumptions. Corollary 10 relies on the maximum variance assumption (Assumption 2), while Corollary 11 relies on the Lipschitz continuity of the variance function (Assumption 4). From Corollary 10, by choosing appropriate values for $\Delta$ and $c$, we derive results such as the logarithmic minimax ratio (Corollary 2), asymptotic optimality (Corollary 3), and Sub-UCB criterion (Corollary 4). From the immediate result Corollary 11, we derive the adaptive variance ratio (Corollary 5).

Corollaries 10 and 11 are auxiliary corollaries introduced to simplify the analysis and provide additional immediate results. For simplicity, we prove only the stronger version of the auxiliary corollary among those two, Corollary 11, under Assumption 4. Corollary 10 can be proven using the same procedure as Corollary 11, but by substituting a different result from Lemma 25.

**Corollary 10** (Regret upper bound corollary). *For any K-arm bandit problem with Assumptions 1 to 3, EXP-KL-MS (Algorithm 1) has regret bounded as follows. For any $\Delta > 0$ and $c \in (0, \frac{1}{4}]$:*

$$\text{Regret}(T) \leq T\Delta + \sum_{a \in [K]:\Delta_a > \Delta} \Delta_a \left( \frac{\ln\left(T D_{\text{KL}}\left(\mu_a + c\Delta_a, \mu_{\max} - c\Delta_a\right) \vee e\right)}{D_{\text{KL}}\left(\mu_a + c\Delta_a, \mu_{\max} - c\Delta_a\right)} \right)$$

$$+ \sum_{a \in [K]:\Delta_a > \Delta} \left( \frac{2}{(1-2c)^2} + \frac{2}{c^2} \right) \frac{\bar{V}}{\Delta_a} + \left( \left( \frac{2\bar{V}}{c^2\Delta_a} + \frac{4\bar{V}^2}{c^4\Delta_a^3} \right) \wedge \left( \frac{2(6+5e)\bar{V}}{c^2\Delta_a} \ln\left( \frac{T\Delta_a^2}{\bar{V}} \vee e \right) \right) \right) \quad (7)$$

**Corollary 11** (Regret upper bound corollary). *For any K-arm bandit problem with Assumptions 1, 3 and 4, EXP-KL-MS (Algorithm 1) has regret bounded as follows. For any $\Delta > 0$ and $c \in (0, \frac{1}{4}]$:*

$$\text{Regret}(T) \leq T\Delta + \sum_{a \in [K]:\Delta_a > \Delta} \Delta_a \left( \frac{\ln\left(T D_{\text{KL}}\left(\mu_a + c\Delta_a, \mu_{\max} - c\Delta_a\right) \vee e\right)}{D_{\text{KL}}\left(\mu_a + c\Delta_a, \mu_{\max} - c\Delta_a\right)} \right)$$

$$+ \sum_{a \in [K]:\Delta_a > \Delta} \left( \frac{2}{(1-2c)^2} + \frac{2}{c^2} \right) \left( \frac{V(\mu_{\max})}{\Delta_a} + C_L \right)$$

$$+ \sum_{a \in [K]:\Delta_a > \Delta} \left( \frac{8}{c^4} \left( \frac{V(\mu_{\max})^2}{\Delta_a^3} + \frac{C_L^2}{\Delta_a} \right) \right) \wedge \left( \frac{2(6+5e)}{c^2} \left( \frac{V(\mu_{\max})}{\Delta_a} + C_L \right) \ln\left( \frac{T\Delta_a^2}{V(\mu_{\max})} \vee e \right) + O\left( \Delta_a \right) \right) \quad (8)$$

*Proof of Corollary 11.* Based on the main equation (Equation (4)) in Theorem 1, it suffices to show that the summation term except $u_a$ on the right side of the main equation is bounded by $\left( \frac{V(\mu_{\max})}{\Delta_a^2} + \frac{C_L}{\Delta_a} \right)$ except $\frac{1}{(D_{\text{KL}}(\mu_{\max} - \varepsilon_{2,a}, \mu_{\max}))^2}$ which is dominated by $\left( \frac{V(\mu_{\max})^2}{\Delta_a^4} + \frac{C_L^2}{\Delta_a^2} \right)$ ignoring constant factor. With Assumption 4, KL divergences between two distributions in $\mathcal{F}_m(M)$ can be lower bounded using Lemma 25 and we will apply such lower bound to upper bound the RHS of the main equation. Let $\varepsilon_{1,a} = \varepsilon_{2,a} = c\Delta_a, c \in (0, \frac{1}{4}]$, each term in the RHS of the main equation can be upper bounded as follows:

- 

$$T\exp\left(-L(u_a)D_{\text{KL}}\left(\mu_a + \varepsilon_{1,a}, \mu_{\max} - \varepsilon_{2,a}\right)\right) \leq 2 \cdot \left( \frac{V(\mu_{\max} - \varepsilon_{2,a}) + C_L(\Delta_a - \varepsilon_{1,a} - \varepsilon_{2,a})}{(\Delta_a - \varepsilon_{1,a} - \varepsilon_{2,a})^2} \right)$$
(Lemma 25)

$$\leq 2 \cdot \left( \frac{V(\mu_{\max}) + C_L(\Delta_a - \varepsilon_{1,a})}{(\Delta_a - \varepsilon_{1,a} - \varepsilon_{2,a})^2} \right)$$
(Lipschitz property of $V(\cdot)$)

$$\leq \frac{2}{(1-2c)^2} \left( \frac{V(\mu_{\max})}{\Delta_a^2} + \frac{C_L}{\Delta_a} \right)$$
($\varepsilon_{1,a} = \varepsilon_{2,a} = c\Delta_a$)

- 

$$\frac{1}{D_{\text{KL}}\left(\mu_a + \varepsilon_{1,a}, \mu_a\right)} \leq 2 \cdot \left( \frac{V(\mu_a + \varepsilon_{1,a}) + C_L\varepsilon_{1,a}}{\varepsilon_{1,a}^2} \right)$$
(Lemma 25)

$$\leq 2 \cdot \left( \frac{V(\mu_{\max}) + C_L\Delta_a}{c^2\Delta_a^2} \right)$$
($\varepsilon_{1,a} = c\Delta_a$)

$$= \frac{2}{c^2} \left( \frac{V(\mu_{\max})}{\Delta_a^2} + \frac{C_L}{\Delta_a} \right)$$

- 

$$\frac{1}{\left(D_{\text{KL}}\left(\mu_{\max} - \varepsilon_{2,a}, \mu_{\max}\right)\right)^2} \leq 4 \cdot \left( \frac{V(\mu_{\max}) + C_L\varepsilon_{2,a}}{\varepsilon_{2,a}^2} \right)^2$$
(Lemma 25)

$$\leq \frac{8}{c^4} \left( \frac{V(\mu_{\max})^2}{\Delta_a^4} + \frac{C_L^2}{\Delta_a^2} \right)$$
($\varepsilon_{2,a} = c\Delta_a$ and AM-QM)

- 

$$\frac{\ln(TD_{\mathrm{KL}}\left(\mu_{\max}-\varepsilon_{2,a},\mu_{\max}\right)\vee e)}{D_{\mathrm{KL}}\left(\mu_{\max}-\varepsilon_{2,a},\mu_{\max}\right)}\leq\frac{2(V(\mu_{\max})+C_L\varepsilon_{2,a})}{\varepsilon_{2,a}^2}\ln\left(\frac{T\varepsilon_{2,a}^2}{2(V(\mu_{\max})+C_L\varepsilon_{2,a})}\vee e\right)$$

(Monotonicity of $\ln(x\vee e)/x$ and Lemma 25)

$$\leq\frac{2}{c^2}\left(\frac{V(\mu_{\max})}{\Delta_a^2}+\frac{C_L}{\Delta_a}\right)\ln\left(\frac{T\Delta_a^2}{V(\mu_{\max})}\vee e\right)\qquad(\varepsilon_{2,a}=c\Delta_a)$$

Using the above inequalities, substituting them into the main equation (Equation (4)) in Theorem 1 and upper bounding the regret,

$$\mathrm{Regret}(T)$$

$$\leq T\Delta+\sum_{a\in[K]:\Delta_a>\Delta}\Delta_a\left(\frac{\ln\left(TD_{\mathrm{KL}}\left(\mu_a+c\Delta_a,\mu_{\max}-c\Delta_a\right)\vee e\right)}{D_{\mathrm{KL}}\left(\mu_a+c\Delta_a,\mu_{\max}-c\Delta_a\right)}\right)+\Delta_a$$

$$+\sum_{a\in[K]:\Delta_a>\Delta}\left(\frac{2}{(1-2c)^2}+\frac{2}{c^2}\right)\left(\frac{V(\mu_{\max})}{\Delta_a}+C_L\right)$$

$$+\sum_{a\in[K]:\Delta_a>\Delta}\left(\frac{8}{c^4}\left(\frac{V(\mu_{\max})^2}{\Delta_a^3}+\frac{C_L^2}{\Delta_a}\right)\right)\wedge\left(\frac{2(6+5e)}{c^2}\left(\frac{V(\mu_{\max})}{\Delta_a}+C_L\right)\ln\left(\frac{T\Delta_a^2}{V(\mu_{\max})}\vee e\right)\right)$$

$\square$

### B.3 Showing That Exp-KL-MS Satisfies Multiple Optimality and Adaptivity Criterion

Notice that $\sqrt{\ln(K)}$ minimax ratio (Corollary 2), Asymptotic Optimality (Corollary 3) and Sub-UCB criterion (Corollary 4) rely on the maximum variance assumption (Assumption 2) instead of Lipschitzness variance assumption (Assumption 4). Therefore, we use Corollary 10 to prove the above three corollaries and use Corollary 11 to prove Corollary 5.

#### B.3.1 Proof of the Logarithmic Minimax Ratio

*Proof of Corollary 2.* We start from Corollary 10. First, we can upper bound the first summation term in the conclusion (Equation (7)) of Corollary 10. Based on the monotonicity of the function $f(x)=\frac{\ln(cx\vee e)}{x}$ for $c>0$, and using the result from KL lower bound Lemma (Lemma 25), which states that

$$D_{\mathrm{KL}}\left(\mu_a+c\Delta_a,\mu_{\max}-c\Delta_a\right)\geq\frac{(1-2c)^2\Delta_a^2}{2\bar{V}},$$

we can show that such a summation term is bounded by

$$\frac{\Delta_a\ln\left(TD_{\mathrm{KL}}\left(\mu_a+c\Delta_a,\mu_{\max}-c\Delta_a\right)\vee e\right)}{D_{\mathrm{KL}}\left(\mu_a+c\Delta_a,\mu_{\max}-c\Delta_a\right)}$$

$$\leq\frac{\Delta_a\ln\left(\frac{T(1-2c)^2\Delta_a^2}{2\bar{V}}\vee e\right)}{\frac{(1-2c)^2\Delta_a^2}{2\bar{V}}}\leq\frac{2\bar{V}}{(1-2c)^2\Delta_a}\ln\left(\frac{T\Delta_a^2}{\bar{V}}\vee e\right)$$

$$=O\left(\frac{\bar{V}}{\Delta_a}\ln\left(\frac{T\Delta_a^2}{\bar{V}}\vee e\right)\right)\tag{9}$$

Then we will apply the above inequality to the main Corollary 10. By letting $c=\frac{1}{4}$ and $\Delta=\sqrt{\frac{\bar{V}K\ln(K)}{T}}$, we can upper bound the regret by

$$\text{Regret}(T) \leq T\Delta + \sum_{a \in [K]:\Delta_a > \Delta} O\left(\frac{\bar{V}}{\Delta_a} \ln\left(\frac{T\Delta_a^2}{\bar{V}} \vee e\right) + \Delta_a\right)$$

$$\leq \sqrt{\bar{V}KT\ln(K)} + O\left(\sum_{a:\Delta_a > \Delta} \Delta_a\right)$$

The last term $O\left(\sum_{a:\Delta_a > \Delta} \Delta_a\right)$ has lower order than $\sqrt{KT}$ when $T$ is sufficiently large and we can conclude that EXP-KL-MS enjoys a minimax ratio of $\sqrt{\ln(K)}$. $\square$

Notice that if the variance function $V(x)$ is also always a constant, such as when the rewards of all arms follow a Gaussian distribution with fixed variance $\sigma^2$, $C_L = 0$ and the lower order term will only include $O(\Delta_a)$.

### B.3.2 Proof of the Asymptotic Optimality

*Proof of Corollary 3.* Recall that the KL-divergence has the following expression according to Equation (3),

$$D_{\text{KL}}(\mu_i, \mu_j) = b(\theta_j) - b(\theta_i) - b'(\theta_j)(\theta_j - \theta_i),$$

and according to Assumption 1 we know that $b''(\cdot)$ is continuous and always positive in the parameter space $\Theta$. Therefore, $D_{\text{KL}}(\mu_i, \mu_j)$ is continuous in terms of $\mu_i$ and $\mu_j$. From Theorem 1, we only need to find two sequences $\{\varepsilon_{1,a}^t\}_{t=1}^T$ and $\{\varepsilon_{2,a}^t\}_{t=1}^T$ such that they satisfy the following

$$\varepsilon_{1,a}^T \to 0, \varepsilon_{2,a}^T \to 0 \text{ as } T \to \infty,$$
$$D_{\text{KL}}\left(\mu_{\max} - \varepsilon_{2,a}^T, \mu_{\max}\right) \to (\ln(T))^{-1/3}, \quad D_{\text{KL}}\left(\mu_a + \varepsilon_{1,a}^T, \mu_a\right) \to (\ln(T))^{-1/2}$$

Since $\varepsilon_{1,a}^T \to 0, \varepsilon_{2,a}^T \to 0$ as $T \to \infty$, by the continuity of $D_{\text{KL}}(\cdot, \cdot)$, we can also conclude that $D_{\text{KL}}\left(\mu_a + \varepsilon_{1,a}^T, \mu_{\max} - \varepsilon_{2,a}^T\right)$ converges to $D_{\text{KL}}(\mu_a, \mu_{\max})$. Based on the above observations, starting from Theorem 1 and letting $\Delta = 0$ we have the following

$$\limsup_{T \to \infty} \frac{\text{Regret}(T)}{\ln(T)}$$

$$\leq \lim_{T \to \infty} \sum_{a \in [K]:\Delta_a > 0} \frac{\Delta_a}{\ln(T)} \left(\frac{\ln\left(TD_{\text{KL}}\left(\mu_a + \varepsilon_{1,a}, \mu_{\max} - \varepsilon_{2,a}\right) \vee e\right)}{D_{\text{KL}}\left(\mu_a + \varepsilon_{1,a}, \mu_{\max} - \varepsilon_{2,a}\right)}\right)$$

$$+ \sum_{a \in [K]:\Delta_a > 0} \frac{\Delta_a}{\ln(T)} \left(\frac{1}{D_{\text{KL}}\left(\mu_a + \varepsilon_{1,a}, \mu_{\max} - \varepsilon_{2,a}\right)} + \frac{1}{D_{\text{KL}}\left(\mu_a + \varepsilon_{1,a}, \mu_a\right)}\right)$$

$$+ \sum_{a \in [K]:\Delta_a > 0} \frac{\Delta_a}{\ln(T)} \left(\frac{1}{D_{\text{KL}}\left(\mu_{\max} - \varepsilon_{2,a}, \mu_{\max}\right)} + \frac{1}{\left(D_{\text{KL}}\left(\mu_{\max} - \varepsilon_{2,a}, \mu_{\max}\right)\right)^2}\right)$$

$$= \sum_{a \in [K]:\Delta_a > 0} \frac{\Delta_a}{D_{\text{KL}}(\mu_a, \mu_{\max})} + \lim_{T \to \infty} \sum_{a \in [K]:\Delta_a > 0} \frac{\Delta_a}{\ln(T)} \left(\frac{\ln(D_{\text{KL}}(\mu_a, \mu_{\max}) \vee e)}{D_{\text{KL}}(\mu_a, \mu_{\max})}\right)$$

$$+ \sum_{a \in [K]:\Delta_a > 0} \frac{\Delta_a}{\ln(T)} \left(\frac{1}{D_{\text{KL}}(\mu_a, \mu_{\max})} + \ln(T)^{1/3}\right)$$

$$+ \sum_{a \in [K]:\Delta_a > 0} \frac{\Delta_a}{\ln(T)} \left((\ln(T))^{1/3} + (\ln(T))^{2/3}\right)$$

$$= \sum_{a \in [K]:\Delta_a > 0} \frac{\Delta_a}{D_{\text{KL}}(\mu_a, \mu_{\max})}$$

Notice that in the above equations, to avoid notation clutter, we use $\varepsilon_{1,a}$ and $\varepsilon_{2,a}$ to denote the $\varepsilon_{1,a}^T$ and $\varepsilon_{2,a}^T$, respectively.

Then from Lai et al. (1985), we know that the lower bound of the regret is $\sum_{a \in [K]:\Delta_a > 0} \frac{\Delta_a}{D_{\mathrm{KL}}(\mu_a, \mu_{\max})}$, which matches the upper bound we derived above. Therefore, we can conclude that EXP-KL-MS is asymptotically optimal. $\qquad\square$

### B.3.3   Proof of Sub-UCB Criterion

*Proof of Corollary 4.* We start from Corollary 10 and set $\Delta = 0$. For the second term on the RHS of the conclusion (Equation (7)) of the corollary, we know it can be upper bounded by

$$\frac{\Delta_a \ln(T D_{\mathrm{KL}}(\mu_a + c\Delta_a, \mu_{\max} - c\Delta_a))}{D_{\mathrm{KL}}(\mu_a + c\Delta_a, \mu_{\max} - c\Delta_a)} \le O\left(\frac{\bar{V}}{\Delta_a} \ln\left(\frac{T\Delta_a^2}{\bar{V}} \vee e\right)\right),$$

as shown in the intermediate step (Equation (9)) in the proof of Corollary 2. For the other terms on the RHS of Equation (7), they can all be upper bounded by $O\left(\frac{\bar{V}}{\Delta_a} \ln\left(\frac{T\Delta_a^2}{\bar{V}} \vee e\right)\right)$ and we can conclude that

$$\mathrm{Regret}(T) \le O\left(\sum_{a \in [K]:\Delta_a > 0} \frac{\bar{V}}{\Delta_a} \ln\left(\frac{T\Delta_a^2}{\bar{V}} \vee e\right) + \Delta_a\right) = O\left(\sum_{a \in [K]:\Delta_a > 0} \frac{\bar{V} \ln(T)}{\Delta_a} + \Delta_a\right)$$

$\qquad\square$

### B.3.4   Proof of Adaptive Variance Ratio

*Proof of Corollary 5.* Notice that the adaptive variance ratio requires Assumption 4, so we start from Corollary 11. First, we can upper bound the first summation term in the conclusion (Equation (8)) of Corollary 11. Based on the monotonicity of the function $f(x) = \frac{\ln(ax \vee e)}{x}$ for $a > 0$, and using the result from Lemma for a lower bound to the KL divergence (Lemma 25), which states that

$$D_{\mathrm{KL}}(\mu_a + c\Delta_a, \mu_{\max} - c\Delta_a) \ge \frac{1}{2}\frac{(1-2c)^2\Delta_a^2}{V(\mu_{\max} - c\Delta_a) + C_L(1-2c)\Delta_a} \ge \frac{1}{2}\frac{(1-2c)^2\Delta_a^2}{V(\mu_{\max}) + C_L(1-c)\Delta_a},$$

we can show that the second term on the RHS of Equation (8) is bounded by

$$\frac{\Delta_a \ln\left(T D_{\mathrm{KL}}(\mu_a + c\Delta_a, \mu_{\max} - c\Delta_a) \vee e\right)}{D_{\mathrm{KL}}(\mu_a + c\Delta_a, \mu_{\max} - c\Delta_a)}$$

$$\le \frac{\Delta_a \ln\left(\frac{1}{2}\left(\frac{T(1-2c)^2\Delta_a^2}{V(\mu_{\max}) + C_L(1-c)\Delta_a}\right) \vee e\right)}{\frac{(1-2c)^2\Delta_a^2}{V(\mu_{\max}) + C_L(1-c)\Delta_a}} \le \frac{V(\mu_{\max}) + C_L(1-c)\Delta_a}{(1-2c)^2\Delta_a} \ln\left(\frac{T\Delta_a}{V(\mu_{\max})} \vee e\right)$$

$$= O\left(\left(\frac{V(\mu_{\max})}{\Delta_a} + C_L\right) \ln\left(\frac{T\Delta_a^2}{V(\mu_{\max})} \vee e\right)\right) \tag{10}$$

Then we will apply the above inequality to the main Theorem 1. By letting $c = \frac{1}{4}$ and $\Delta = \sqrt{\frac{V(\mu_{\max})K\ln(K)}{T}}$, we can upper bound the regret by

$$\mathrm{Regret}(T) \le T\Delta + \sum_{a \in [K]:\Delta_a > \Delta} O\left(\left(\frac{V(\mu_{\max})}{\Delta_a} + C_L\right) \ln\left(\frac{T\Delta_a^2}{V(\mu_{\max})} \vee e\right) + \Delta_a\right)$$

$$\le \sqrt{V(\mu_{\max})KT\ln(K)} + O\left(\sum_{a:\Delta_a > \Delta} C_L \ln\left(\frac{T\Delta_a^2}{V(\mu_{\max})} \vee e\right) + \Delta_a\right)$$

The last term $O\left(\sum_{a:\Delta_a > \Delta} C_L \ln\left(\frac{T\Delta_a^2}{V(\mu_{\max})} \vee e\right)\right)$ has lower order than $\sqrt{KT}$ when $T$ is sufficiently large. We can conclude that EXP-KL-MS enjoys an adaptive minimax ratio as $\sqrt{V(\mu_{\max})}$.

Notice that in the last step of the above inequality, we assume $V(\mu_{\max})$ is not too small. If $V(\mu_{\max}) \to 0$, we can let $\Delta \to 0$ and the above inequality implies that

$$\text{Regret}(T) \le T\Delta + \sum_{a \in [K]:\Delta_a > \Delta} O\left(C_L \ln\left(\frac{T\Delta_a^2}{C_L} \vee e\right) + \Delta_a\right) \le \sum_{a \in [K]:\Delta_a > \Delta} O\left(C_L \ln\left(\frac{T\Delta_a^2}{C_L} \vee e\right) + \Delta_a\right)$$

The above regret bound is much smaller than the minimax bound when $T$ is large. $\qquad\square$

## C  Results of Extensions

In this section, we will present results from different choices of inverse temperature function $L(\cdot)$. We demonstrate that GENERAL-EXP-KL-MS with $L(k) = k/d$ (where $d > 1$) and $L(k) = k$ are two examples that exhibit some desirable properties, but not all of them. This shows that GENERAL-EXP-KL-MS is flexible, allowing for a variety of choices for $L(\cdot)$.

### C.1  $L(k) = k/d$

In this subsection, we present a theorem statement with its proof that serves a similar role to Theorem 1, providing a regret upper bound for GENERAL-EXP-KL-MS with $L(k) = k/d$. Additionally, GENERAL-EXP-KL-MS with $L(k) = k/d$ can achieve a logarithmic minimax ratio and satisfy the Sub-UCB criterion. We summarize these results in Corollary 6 and Corollary 7, with their proofs provided afterwards.

**Theorem 12.** *For any $K$-arm bandit problem with Assumptions 1, 3 and 4 and $L(k) = k/d$, $d > 1$ GENERAL-EXP-KL-MS (Algorithm 1) has regret bounded as follows. For any $\Delta \ge 0$ and $c \in (0, \frac{1}{4}]$:*

$$\begin{aligned}
\text{Regret}(T) \le &T\Delta + \sum_{a \in [K]:\Delta_a > \Delta} \Delta_a \left(\frac{d\ln\left(TD_{\text{KL}}\left(\mu_a + c\Delta_a, \mu_{\max} - c\Delta_a\right) \vee e\right)}{D_{\text{KL}}\left(\mu_a + c\Delta_a, \mu_{\max} - c\Delta_a\right)}\right) + \Delta_a \\
&+ \left(\frac{2}{(1 - 2c)^2} + \frac{2(2d - 1)}{c^2(d - 1)}\right)\left(\frac{V(\mu_{\max})}{\Delta_a} + C_L\right)
\end{aligned} \tag{11}$$

Notice that the RHS of Equation (11) cannot guarantee that GENERAL-EXP-KL-MS with $L(k) = k/d$ achieves asymptotic optimality due to the presence of an additional constant factor $d$ in the second term. However, since the RHS does not include $\ln(T)$ beyond the leading term (the second term), we can demonstrate that GENERAL-EXP-KL-MS with $L(k) = k/d$ achieves minimax optimality, has an adaptive variance ratio of $\sqrt{V(\mu_{\max})}$ and satisfies Sub-UCB criterion.

*Proof of Theorem 12.* We follow the proof strategy used in proving Theorem 1 and Corollary 11 but change the definition of $u$ from $\frac{\ln(TD_{\text{KL}}(\mu_a + \varepsilon_{1,a}, \mu_{\max} - \varepsilon_{2,a}) \vee e)}{D_{\text{KL}}(\mu_a + \varepsilon_{1,a}, \mu_{\max} - \varepsilon_{2,a})} + 1$ to $\frac{d\ln(TD_{\text{KL}}(\mu_a + \varepsilon_{1,a}, \mu_{\max} - \varepsilon_{2,a}) \vee e)}{D_{\text{KL}}(\mu_{\max} + \varepsilon_{1,a}, \mu_a - \varepsilon_{2,a})} + 1$. The reason we make such a change is that we need $\mathcal{G}_a$ to be controlled by $1/D_{\text{KL}}\left(\mu_{\max} + \varepsilon_{1,a}, \mu_a - \varepsilon_{2,a}\right)$. Following the same case splitting we did in the proof of Theorem 1, we have

$$\text{Regret}(T) \le T\Delta + \sum_{a \in [K]:\Delta_a > \Delta} \Delta_a \left(u + \mathcal{G}_a + \mathcal{B}_a^1 + \mathcal{B}_a^2\right)$$

and we bound each term on the RHS by

- 
$$\begin{aligned}
\mathcal{G}_a \le &T\exp\left(-L(u_a)D_{\text{KL}}\left(\mu_a + \varepsilon_{1,a}, \mu_{\max} - \varepsilon_{2,a}\right)\right) \le \frac{1}{D_{\text{KL}}\left(\mu_a + \varepsilon_{1,a}, \mu_{\max} - \varepsilon_{2,a}\right)} \quad &\text{(Proposition 15)} \\
\le &\frac{2}{(1 - 2c)^2}\left(\frac{V(\mu_{\max})}{\Delta_a^2} + \frac{C_L}{\Delta_a}\right) \quad &\text{(Lemma 25)}
\end{aligned}$$

- 

$$\mathcal{B}_a^1 \leq \frac{1}{D_{\mathrm{KL}}\left(\mu_a + \varepsilon_{1,a}, \mu_a\right)} \qquad \text{(Proposition 16)}$$

$$\leq 2 \cdot \left(\frac{V(\mu_{\max}) + C_L \Delta_a}{\varepsilon_{1,a}^2}\right) \leq \frac{2}{c^2}\left(\frac{V(\mu_{\max})}{\Delta_a^2} + \frac{C_L}{\Delta_a}\right) \qquad \text{(Lemma 25)}$$

- 

$$\mathcal{B}_a^2 \leq \frac{1}{D_{\mathrm{KL}}\left(\mu_{\max} - \varepsilon_{2,a}, \mu_{\max}\right)} + \sum_{k=1}^T \frac{L(k)}{k - L(k)} e^{-k D_{\mathrm{KL}}(\mu_{\max} - \varepsilon_{2,a}, \mu_{\max})} \qquad \text{(Proposition 17)}$$

$$= \frac{1}{D_{\mathrm{KL}}\left(\mu_{\max} - \varepsilon_{2,a}, \mu_{\max}\right)} + \frac{1}{d-1} \sum_{k=1}^T \exp\left(-k D_{\mathrm{KL}}\left(\mu_{\max} - \varepsilon_{2,a}, \mu_{\max}\right)\right)$$

$$\leq \frac{1}{D_{\mathrm{KL}}\left(\mu_{\max} - \varepsilon_{2,a}, \mu_{\max}\right)} + \frac{1}{d-1} \cdot \frac{1}{D_{\mathrm{KL}}\left(\mu_{\max} - \varepsilon_{2,a}, \mu_{\max}\right)}$$
$$\text{(for } a > 0,\ \sum_{k=1}^\infty e^{-ak} = \frac{1}{e^a - 1} \leq \frac{1}{a}\text{)}$$

$$\leq \frac{d}{(d-1) D_{\mathrm{KL}}\left(\mu_{\max} - \varepsilon_{2,a}, \mu_{\max}\right)}$$

$$\leq \frac{2d}{c^2(d-1)}\left(\frac{V(\mu_{\max})}{\Delta_a^2} + \frac{C_L}{\Delta_a}\right)$$

Therefore, we can bound the regret by

$$\mathrm{Regret}(T)$$
$$\leq T\Delta + \sum_{a \in [K]:\Delta_a > \Delta} \Delta_a \left(\frac{d \ln\left(T D_{\mathrm{KL}}\left(\mu_a + c\Delta_a, \mu_{\max} - c\Delta_a\right) \vee e\right)}{D_{\mathrm{KL}}\left(\mu_a + c\Delta_a, \mu_{\max} - c\Delta_a\right)} + 1\right)$$
$$+ \frac{2\Delta_a}{(1-2c)^2}\left(\frac{V(\mu_{\max})}{\Delta_a^2} + \frac{C_L}{\Delta_a}\right) + \frac{2\Delta_a}{c^2}\left(\frac{V(\mu_{\max})}{\Delta_a^2} + \frac{C_L}{\Delta_a}\right) + \left(\frac{2d\Delta_a}{c^2(d-1)}\left(\frac{V(\mu_{\max})}{\Delta_a^2} + \frac{C_L}{\Delta_a}\right)\right)$$
$$\leq T\Delta + \sum_{a \in [K]:\Delta_a > \Delta} \Delta_a \left(\frac{d \ln\left(T D_{\mathrm{KL}}\left(\mu_a + c\Delta_a, \mu_{\max} - c\Delta_a\right) \vee e\right)}{D_{\mathrm{KL}}\left(\mu_a + c\Delta_a, \mu_{\max} - c\Delta_a\right)}\right) + \Delta_a$$
$$+ \sum_{a \in [K]:\Delta_a > \Delta} \left(\frac{2}{(1-2c)^2} + \frac{2(2d-1)}{c^2(d-1)}\right)\left(\frac{V(\mu_{\max})}{\Delta_a} + C_L\right)$$

$$\square$$

### C.1.1 Proof of the Logarithmic Minimax Ratio and Adaptive Variance Ratio

Based on the above Theorem 12, we can show that GENERAL-EXP-KL-MS with $L(k) = k/d$ has a logarithmic minimax ratio and adaptive variance ratio in Corollary 6.

**Corollary 6.** (Logarithmic Minimax Ratio and Adaptive Variance Ratio) *For any $K$-arm bandit problem with Assumptions 1, 3 and 4, when $d > 1$ GENERAL-EXP-KL-MS with $L(k) = k/d$ has regret:*

$$\mathrm{Regret}(T) \leq O\left(\sqrt{V(\mu_{\max}) K T \ln(K)}\right) + O\left(K \ln(T)\right).$$

*Proof of Corollary 6.* We follow the proof of Corollary 2 to bound the RHS of Equation (11) in Theorem 12. By choosing $\Delta = \sqrt{V(\mu_{\max}) K / T}$, it suffices to show each term in the RHS of Equation (11) is upper bounded by $\sqrt{KT}$ or $\sum_{a \in [K]} \Delta_a$. The first term $T\Delta = O\left(\sqrt{KT}\right)$ because of the choice of $\Delta$. In the second term, we

can utilize Equation (10) from the proof of Corollary 5

$$\sum_{a\in[K]:\Delta_a>\Delta}\Delta_a\left(\frac{d\ln\left(TD_{\mathrm{KL}}\left(\mu_a+c\Delta_a,\mu_{\max}-c\Delta_a\right)\vee e\right)}{D_{\mathrm{KL}}\left(\mu_a+c\Delta_a,\mu_{\max}-c\Delta_a\right)}\right)$$

$$\leq O\left(\sum_{a\in[K]:\Delta_a>\Delta}\left(\frac{V(\mu_{\max})}{\Delta_a}+C_L\right)\ln\left(\frac{T\Delta_a^2}{V(\mu_{\max})}\vee e\right)\right)\qquad\text{(Equation (10))}$$

$$\leq O\left(\sum_{a\in[K]:\Delta_a>\Delta}\left(\frac{V(\mu_{\max})}{\Delta}+C_L\right)\ln\left(\frac{T\Delta^2}{V(\mu_{\max})}\vee e\right)\right)\qquad\text{(Monotonicity property)}$$

$$\leq O\left(\sqrt{KT\ln(K)}\right)+O\left(K\ln(T)\right)\qquad(\Delta=\sqrt{V(\mu_{\max})K/T})$$

For the fourth term, we have the following, ignoring the constant factors,

$$\sum_{a\in[K]:\Delta_a>0}\frac{V(\mu_{\max})}{\Delta_a}+C_L\leq\sum_{a\in[K]:\Delta_a>0}\frac{V(\mu_{\max})}{\Delta}+C_L=\sqrt{V(\mu_{\max})KT\ln(K)}+C_LK$$

Overall, by combining the above bounds, it holds that

$$\mathrm{Regret}(T)=O\left(\sqrt{V(\mu_{\max})KT\ln(K)}\right)+O\left(K\ln(T)\right)$$

$\square$

### C.1.2 Proof of Sub-UCB Criterion

Based on the above Theorem 12, we can show that GENERAL-EXP-KL-MS with $L(k)=k/d$ satisfies Sub-UCB criterion in Corollary 7.

**Corollary 7.** (Sub-UCB criterion) *For any $K$-arm bandit problem with Assumptions 1, 3 and 4, when $d>1$ GENERAL-EXP-KL-MS with $L(k)=k/d$ satisfies Sub-UCB criterion, which means that its regret is bounded by*

$$\mathrm{Regret}(T)\leq O\Big(\sum_{a:\Delta_a>0}\frac{\ln(T)}{\Delta_a}+\Delta_a\Big).$$

*Proof of Corollary 7.* We follow the proof of Corollary 4 to bound the RHS of Equation (11) in Theorem 12 and choose $\Delta=0$. Then for each term in the RHS of Equation (11), we can show that,

- For the second term,

$$\sum_{a\in[K]:\Delta_a>\Delta}\Delta_a\left(\frac{d\ln\left(TD_{\mathrm{KL}}\left(\mu_a+c\Delta_a,\mu_{\max}-c\Delta_a\right)\vee e\right)}{D_{\mathrm{KL}}\left(\mu_a+c\Delta_a,\mu_{\max}-c\Delta_a\right)}\right)$$

$$\leq O\left(\sum_{a\in[K]:\Delta_a>\Delta}\left(\frac{V(\mu_{\max})}{\Delta_a}+C_L\right)\ln\left(\frac{T\Delta_a^2}{V(\mu_{\max})}\vee e\right)\right)\qquad\text{(Equation (10))}$$

$$=O\left(\sum_{a\in[K]:\Delta_a>\Delta}\frac{V(\mu_{\max})\ln(T)}{\Delta_a}+\Delta_a\right)$$

- For the fourth term,

$$\sum_{a\in[K]:\Delta_a>\Delta}\frac{V(\mu_{\max})}{\Delta_a}+C_L=O\left(\sum_{a\in[K]:\Delta_a>\Delta}\frac{V(\mu_{\max})\ln(T)}{\Delta_a}+\Delta_a\right)$$

Combining the above analysis, we can conclude that

$$\text{Regret}(T) = O\left(\sum_{a \in [K]:\Delta_a > \Delta} \frac{\ln(T)}{\Delta_a} + \Delta_a\right)$$

$\square$

## C.2  $L(k) = k$

In this subsection, we present Theorem 13 that provides a regret upper bound to GENERAL-EXP-KL-MS with $L(k) = k$ and shows that it has an adaptive variance ratio in Corollary 8 and Sub-UCB criterion in Corollary 9.

**Theorem 13.** *For any $K$-arm bandit problem with Assumptions 1, 3 and 4 and $L(k) = k$, GENERAL-EXP-KL-MS (Algorithm 1) has regret bounded as follows. For any $\Delta \geq 0$ and $c \in (0, \frac{1}{4}]$:*

$$\text{Regret}(T) \leq T\Delta + \sum_{a \in [K]:\Delta_a > \Delta} \Delta_a \left(\frac{\ln\left(TD_{\text{KL}}\left(\mu_a + c\Delta_a, \mu_{\max} - c\Delta_a\right) \vee e\right)}{D_{\text{KL}}\left(\mu_a + c\Delta_a, \mu_{\max} - c\Delta_a\right)}\right) + \Delta_a$$
$$+ \left(\frac{2}{(1 - 2c)^2} + \frac{14}{c^2} + \frac{20\ln(T)}{c^2}\right)\left(\frac{V(\mu_{\max})}{\Delta_a} + C_L\right) \tag{12}$$

**Remark 14.** *The reason we obtain a $\ln(T)$ term for the third one (as opposed to $O(1) \wedge O(\ln(T))$) for $L(k) = k - 1$ and $L(k) = k/d$ is that when $L(k) = k$, Proposition 17 is not applicable, leaving us to only be able to bound $\mathcal{B}_a^2$ using Proposition 18 (Equation (19)).*

*Proof of Theorem 13.* We follow the same proof procedure as in Theorem 1 and Theorem 12. This time, we define $u_a$ as $\frac{\ln(TD_{\text{KL}}(\mu_a + \varepsilon_{1,a}, \mu_{\max} - \varepsilon_{2,a}) \vee e)}{D_{\text{KL}}(\mu_a + \varepsilon_{1,a}, \mu_{\max} - \varepsilon_{2,a})} + 1$ and decompose the regret as follows:

$$\text{Regret}(T) \leq T\Delta + \sum_{a \in [K]:\Delta_a > \Delta} \Delta_a u + \mathcal{G}_a + \mathcal{B}_a^1 + \mathcal{B}_a^2$$

and for each term on the RHS they are bounded by

$$\mathcal{G}_a \leq T\exp\left(-L(u_a)D_{\text{KL}}\left(\mu_a + \varepsilon_{1,a}, \mu_{\max} - \varepsilon_{2,a}\right)\right) \leq \frac{1}{D_{\text{KL}}\left(\mu_a + \varepsilon_{1,a}, \mu_{\max} - \varepsilon_{2,a}\right)} \quad \text{(Proposition 15)}$$
$$\leq \frac{2}{(1 - 2c)^2}\left(\frac{V(\mu_{\max})}{\Delta_a^2} + \frac{C_L}{\Delta_a}\right)$$
$$\mathcal{B}_a^1 \leq \frac{1}{D_{\text{KL}}\left(\mu_a + \varepsilon_{1,a}, \mu_a\right)} \quad \text{(Proposition 16)}$$
$$\leq 2 \cdot \left(\frac{V(\mu_{\max}) + C_L\Delta_a}{\varepsilon_{1,a}^2}\right) \leq \frac{2}{c^2}\left(\frac{V(\mu_{\max})}{\Delta_a^2} + \frac{C_L}{\Delta_a}\right)$$
$$\mathcal{B}_a^2 \leq \frac{6 + 5e\ln(TD_{\text{KL}}\left(\mu_{\max} - \varepsilon_2, \mu_{\max}\right) \vee e)}{D_{\text{KL}}\left(\mu_{\max} - \varepsilon_2, \mu_{\max}\right)} \quad \text{(Proposition 18)}$$
$$\leq \frac{2}{c^2}\left(\frac{V(\mu_{\max})}{\Delta_a^2} + \frac{C_L}{\Delta_a}\right)(6 + 5e\ln(T))$$

In the second inequality of bounding $\mathcal{B}_a^2$, we bound the sum by $\sum_{k=1}^{T} \frac{e \ln(T/k)}{k} \leq e \ln(T) \sum_{k=1}^{T} \frac{1}{k} \leq e \ln(T) (\ln(T) + 1)$. Therefore, we can bound the regret by

$$
\begin{aligned}
\text{Regret}(T) \leq & T\Delta + \sum_{a \in [K]: \Delta_a > \Delta} \Delta_a \left( \frac{\ln \left( T D_{\text{KL}} \left( \mu_a + c\Delta_a, \mu_{\max} - c\Delta_a \right) \vee e \right)}{D_{\text{KL}} \left( \mu_a + c\Delta_a, \mu_{\max} - c\Delta_a \right)} + 1 \right) \\
& + \sum_{a \in [K]: \Delta_a > \Delta} \frac{2\Delta_a}{(1-2c)^2} \left( \frac{V(\mu_{\max})}{\Delta_a^2} + \frac{C_L}{\Delta_a} \right) + \frac{2\Delta_a}{c^2} \left( \frac{V(\mu_{\max})}{\Delta_a^2} + \frac{C_L}{\Delta_a} \right) \\
& + \sum_{a \in [K]: \Delta_a > \Delta} \Delta_a \left( \frac{2}{c^2} \left( \frac{V(\mu_{\max})}{\Delta_a^2} + \frac{C_L}{\Delta_a} \right) (6 + 10 \ln(T)) \right) \\
= & T\Delta + \sum_{a \in [K]: \Delta_a > \Delta} \Delta_a \left( \frac{\ln \left( T D_{\text{KL}} \left( \mu_a + c\Delta_a, \mu_{\max} - c\Delta_a \right) \vee e \right)}{D_{\text{KL}} \left( \mu_a + c\Delta_a, \mu_{\max} - c\Delta_a \right)} \right) + \Delta_a \\
& + \left( \frac{2}{(1-2c)^2} + \frac{14}{c^2} + \frac{20 \ln(T)}{c^2} \right) \left( \frac{V(\mu_{\max})}{\Delta_a} + C_L \right)
\end{aligned}
$$

$\square$

### C.2.1 Proof of the Logarithmic Minimax Ratio and Adaptive Variance Ratio

Based on the above Theorem 13, we can show that GENERAL-EXP-KL-MS with $L(k) = k$ has a logarithmic minimax ratio and adaptive variance ratio in Corollary 8.

**Corollary 8.** (Logarithmic Minimax Ratio and Adaptive Variance Ratio) *For any $K$-arm bandit problem with Assumptions 1, 3 and 4, GENERAL-EXP-KL-MS with $L(k) = k$ has regret bounded as:*

$$
\text{Regret}(T) \leq O\big( \sqrt{V(\mu_{\max}) K T \ln(T)} \big) + O\big( K \ln(T) \big).
$$

*Proof of Corollary 8.* We follow the proof of Corollary 2 and choose $\Delta = \sqrt{V(\mu_{\max}) K \ln(T) / T}$. Then it suffices to show each term in the RHS of Equation (12) is upper bounded by $\sqrt{V(\mu_{\max}) K T \ln(T)}$ or $K \ln(T)$, ignoring the constant. The first term $T\Delta = O\big( \sqrt{K T \ln(T)} \big)$ because the value of $\Delta$. In the second term, we can utilize result of Equation (10) from the proof of Corollary 5

$$
\sum_{a \in [K]: \Delta_a > \Delta} \Delta_a \left( \frac{\ln \left( T D_{\text{KL}} \left( \mu_a + c\Delta_a, \mu_{\max} - c\Delta_a \right) \vee e \right)}{D_{\text{KL}} \left( \mu_a + c\Delta_a, \mu_{\max} - c\Delta_a \right)} \right) \leq O \left( \sqrt{K T \ln(T)} \right) + O \left( K \ln(T) \right)
$$

For the fourth term, we have the following equations, ignoring constant factor,

$$
\sum_{a \in [K]: \Delta_a > 0} \left( \frac{2}{(1-2c)^2} + \frac{14}{c^2} + \frac{20 \ln(T)}{c^2} \right) \left( \frac{V(\mu_{\max})}{\Delta_a} + C_L \right)
$$

$$
\leq O \left( \sum_{a \in [K]: \Delta_a > 0} \ln(T) \left( \frac{V(\mu_{\max})}{\Delta} + C_L \Delta_a \right) \right) = O \left( \sqrt{V(\mu_{\max}) K T \ln(T)} \right) + O \left( K \ln(T) \right)
$$

Overall, by combining the above analysis, it suffices to show that

$$
\text{Regret}(T) = O \left( \sqrt{V(\mu_{\max}) K T \ln(T)} \right) + O \left( K \ln(T) \right)
$$

$\square$

### C.2.2 Proof of Sub-UCB Criterion

Based on the above Theorem 13, we can show that GENERAL-EXP-KL-MS with $L(k) = k$ satisfies Sub-UCB criterion in Corollary 9.

**Corollary 9.** (Sub-UCB criterion) *For any $K$-arm bandit problem with Assumptions 1, 3 and 4, GENERAL-EXP-KL-MS with $L(k) = k$ satisfies the Sub-UCB criterion and has regret bounded as*

$$\text{Regret}(T) \leq O\Big( \sum_{a:\Delta_a > 0} \frac{\ln(T)}{\Delta_a} + \Delta_a \Big).$$

*Proof of Corollary 9.* We follow the proof of Corollary 4 to bound the RHS of Equation (12) in Theorem 13 and choose $\Delta = 0$. Then for each term in the RHS of Equation (12), we can show that,

- For the second term,

$$\sum_{a \in [K]:\Delta_a > \Delta} \Delta_a \left( \frac{d \ln \left( T D_{\text{KL}} \left( \mu_a + c\Delta_a, \mu_{\max} - c\Delta_a \right) \vee e \right)}{D_{\text{KL}} \left( \mu_a + c\Delta_a, \mu_{\max} - c\Delta_a \right)} \right)$$

$$\leq O \left( \sum_{a \in [K]:\Delta_a > \Delta} \left( \frac{V(\mu_{\max})}{\Delta_a} + C_L \right) \ln \left( \frac{T\Delta_a^2}{V(\mu_{\max})} \vee e \right) \right) \qquad \text{(Equation (10))}$$

$$= O \left( \sum_{a \in [K]:\Delta_a > \Delta} \frac{V(\mu_{\max}) \ln(T)}{\Delta_a} + \Delta_a \right)$$

- For the fourth term,

$$\sum_{a \in [K]:\Delta_a > \Delta} \frac{V(\mu_{\max})}{\Delta_a} + C_L = O \left( \sum_{a \in [K]:\Delta_a > \Delta} \frac{V(\mu_{\max}) \ln(T)}{\Delta_a} + \Delta_a \right)$$

Combining the above analysis, we can conclude that

$$\text{Regret}(T) = O \left( \sum_{a \in [K]:\Delta_a > \Delta} \frac{\ln(T)}{\Delta_a} + \Delta_a \right)$$

$\square$

# D  Proof of Propositions

In this section, we focus on proving the propositions in the middle of Figure 2. All propositions hold for general choices of the inverse temperature function $L$ that satisfy $0 < L(k) \leq k$ and increase monotonically with $k$.

Proposition 15 follows directly from the definition of $\mathcal{G}_a$. Proposition 16 leverages the Chernoff's tail bound (Lemma 26) for general exponential family random variables. For Propositions 17 and 18, as mentioned in the Proof Sketch (Section 6), the proof involves constructing a series of clean events to form intervals that lower bound the random variation of $\hat{\mu}_{t,1}$. Proposition 17 relies on Lemma 19 and Proposition 18 relies on Lemmas 19 and 20 and we defer the proof of these two lemmas to Supporting Lemmas (Appendix E).

To simplify the notation, we omit the arm index and use $\varepsilon_1$ to represent $\varepsilon_{1,a}$ and $\varepsilon_2$ to represent $\varepsilon_{2,a}$, as the same analysis applies to every arm.

## D.1  Favorable Term $\mathcal{G}_a$

Recall that the definition of the good estimation event $\mathcal{G}_a$ is

$$\mathcal{G}_a := \mathbb{E} \left[ \sum_{t=K+1}^{T} \mathbf{1} \left\{ A_{t,a} \cap U_{t-1,a}^c \cap E_{t-1,a} \right\} \right],$$

which is the expected number of times arm $a$ is pulled in the case when the agent has collected enough samples, and the estimation of $\mu_{\max}$ is well bounded from below, and $\mu_a$ is well bounded from above. According to the design of the algorithm, the probability of pulling a suboptimal arm $a$ should be small and decrease as $N_{t-1,a}$ increases.

**Proposition 15.** *When $0 < L(k) \leq x$,*

$$\mathcal{G}_a \leq T \exp\left(-L(u) D_{\mathrm{KL}}\left(\mu_a + \varepsilon_1, \mu_{\max} - \varepsilon_2\right)\right) \tag{13}$$

*Proof of Proposition 15.* Recall the notations that $A_{t,a} = \{I_t = a\}$, $U_{t-1,a}^c = \{N_{t-1,a} \geq u\}$, $E_{t-1,a} = \{\hat{\mu}_{t-1,a} \leq \mu_a + \varepsilon_1\}$, $F_{t-1,a} = \{\hat{\mu}_{t-1,\max} \leq \mu_{\max} - \varepsilon_2\}$.

$$
\begin{aligned}
\mathcal{G}_a &= \mathbb{E}\left[\sum_{t=K+1}^{T} \mathbf{1}_{A_{t,a} \cap U_{t-1,a}^c \cap E_{t-1,a} \cap F_{t-1,a}} \mid \mathcal{H}_{t-1}\right] \\
&= \mathbb{E}\left[\sum_{t=K+1}^{T} \mathbf{1}_{U_{t-1,a}^c \cap E_{t-1,a} \cap F_{t-1,a}} \mathbb{E}\left[\mathbf{1}_{A_{t,a}} \mid \mathcal{H}_{t-1}\right]\right] && \text{(Law of the total expectation)} \\
&\leq \sum_{t=K+1}^{T} \mathbb{E}\left[\mathbf{1}\left\{U_{t-1,a}^c \cap E_{t-1,a} \cap F_{t-1,a}\right\} \exp\left(-L(N_{t-1,a}) D_{\mathrm{KL}}\left(\hat{\mu}_{t-1,a}, \hat{\mu}_{t-1,\max}\right)\right)\right] \\
&&& \text{(By Algorithm 1 and } M_t \geq 1, \forall t \in [T]) \\
&\leq \sum_{t=1}^{T} \exp\left(-L(u) \cdot D_{\mathrm{KL}}\left(\mu_a + \varepsilon_1, \mu_{\max} - \varepsilon_2\right)\right) && \text{(When } U_{t-1,a}^c \text{ is true, } N_{t-1,a} \geq u \Rightarrow L(N_{t-1,a}) \geq L(u)) \\
&\leq T \exp\left(-L(u) D_{\mathrm{KL}}\left(\mu_a + \varepsilon_1, \mu_{\max} - \varepsilon_2\right)\right) \tag{14}
\end{aligned}
$$

$\square$

### D.2 Unfavorable Term of Arm $a$, $\mathcal{B}_a^1$

The definition of $\mathcal{B}_a^1$ is

$$\mathcal{B}_a^1 := \mathbb{E}\left[\sum_{t=K+1}^{T} \mathbf{1}\left\{A_{t,a} \cap U_{t-1,a}^c \cap E_{t-1,a}^c \cap F_{t-1,a}\right\}\right],$$

$\mathcal{B}_a^1$ is the expected number of times arm $a$ is pulled in the case where the agent has collected enough samples, but the estimation of arm $a$ deviates from the true mean $\mu_a$ by at least $\varepsilon_1$. The probability of pulling suboptimal arm $a$ is negligible as the number of samples of arm $a$ increases, thus this unfavorable term is relatively small.

**Proposition 16.** *When $0 < L(k) \leq k$,*

$$\mathcal{B}_a^1 \leq \frac{1}{D_{\mathrm{KL}}\left(\mu_a + \varepsilon_1, \mu_a\right)} \tag{15}$$

*Proof of Proposition 16.* Recall the notations that $A_{t,a} = \{I_t = a\}$, $E_{t-1,a}^c = \{\hat{\mu}_{t-1,a} > \mu_a + \varepsilon_1\}$. We have:

$$\mathcal{B}_a^1 = \mathbb{E}\left[\sum_{t=K+1}^T \mathbf{1}\left\{A_{t,a} \cap E_{t-1,a}^c\right\}\right] \leq \mathbb{E}\left[\sum_{k=2}^\infty \mathbf{1}\left\{E_{\tau_a(k)-1,a}^c\right\}\right]$$

( only if $t = \tau_a(k)$ for some $k \geq 2$ the inner indicator is non-zero)

$$= \mathbb{E}\left[\sum_{k=1}^\infty \mathbf{1}\left\{E_{\tau_a(k)}^c\right\}\right] \qquad (E_{\tau_a(k)-1} = E_{\tau_a(k-1)})$$

$$\leq \sum_{k=1}^\infty \exp\left(-k \cdot D_{\mathrm{KL}}\left(\mu_a + \varepsilon_1, \mu_a\right)\right) \qquad \text{(Applying Lemma 26)}$$

$$\leq \frac{\exp\left(-D_{\mathrm{KL}}\left(\mu_a + \varepsilon_1, \mu_a\right)\right)}{1 - \exp\left(-D_{\mathrm{KL}}\left(\mu_a + \varepsilon_1, \mu_a\right)\right)} \qquad \text{(Geometric sum)}$$

$$\leq \frac{1}{D_{\mathrm{KL}}\left(\mu_a + \varepsilon_1, \mu_a\right)} \qquad (16)$$

$\square$

## D.3  Unfavorable Term of the Optimal Arm, $\mathcal{B}_a^2$

Now we need to bound the last case $\mathcal{B}_a^2$. The definition of $\mathcal{B}_a^2$ is

$$\mathcal{B}_a^2 := \mathbb{E}\left[\sum_{t=K+1}^T \mathbf{1}\left\{A_{t,a} \cap U_{t-1,a}^c \cap F_{t-1,a}^c\right\}\right],$$

which represents the expected number of times arm $a$ is pulled when the agent has collected enough samples, and the empirical best mean is underestimated. To achieve asymptotic optimality and minimax optimality with a logarithmic factor, we present two propositions, each of which will be used to establish the respective property. Specifically, Proposition 17 is used to prove asymptotic optimality EXP-KL-MS, while Proposition 18 is used to prove minimax optimality with a $O\big(\sqrt{\ln(K)}\big)$ factor.

**Proposition 17.** *If* $0 < L(k) < k$,

$$\mathcal{B}_a^2 \leq \frac{1}{D_{\mathrm{KL}}\left(\mu_{\max} - \varepsilon_2, \mu_{\max}\right)} + \sum_{k=1}^T \frac{L(k)}{k - L(k)} e^{-k D_{\mathrm{KL}}(\mu_{\max} - \varepsilon_2, \mu_{\max})} \qquad (17)$$

**Proposition 18.** *If* $0 < L(k) < k$,

$$\mathcal{B}_a^2 \leq \frac{6}{D_{\mathrm{KL}}\left(\mu_{\max} - \varepsilon_2, \mu_{\max}\right)} + \sum_{k=1}^T \frac{e\,L(k)}{k} \exp(-k D_{\mathrm{KL}}\left(\mu_{\max} - \varepsilon_2, \mu_{\max}\right)) \cdot \ln(T/k) \qquad (18)$$

*If* $L(k) = k$,

$$\mathcal{B}_a^2 \leq \frac{6 + 5e \ln(T D_{\mathrm{KL}}\left(\mu_{\max} - \varepsilon_2, \mu_{\max}\right) \vee e)}{D_{\mathrm{KL}}\left(\mu_{\max} - \varepsilon_2, \mu_{\max}\right)} \qquad (19)$$

Before delving into the details of the proof, we first recall the key idea outlined in Section 6. Consider the definition of $\mathcal{B}_a^2$, which represents the case when the optimal arm's empirical mean is underestimated, when this happens, by definition of $F_{t,a}^c$ (Recall $F_{t,a} := \{\hat{\mu}_{t,\max} \leq \mu_{\max} - \varepsilon_2\}$), $\hat{\mu}_{t,a}$ will be upper bounded by $\mu_{\max} - \varepsilon_2$ for $t = 1$ to $T$. Next, we construct a series of events $\{\mathcal{E}_k(\alpha_k)\}_{k=1}^\infty$. For each event $\mathcal{E}_k(\alpha_k)$ it is defined as $\{\alpha_k \leq \hat{\mu}_{(k),1}\}$ and is determined by $\alpha_k$. We also define the intersections of all events $\{\mathcal{E}_k(\alpha_k)\}_{k=1}^\infty$ as $\mathcal{E}(\boldsymbol{\alpha}) = \cap_{k=1}^\infty \mathcal{E}_k(\alpha_k)$, where $\boldsymbol{\alpha} = \{\alpha_1, \alpha_2, \dots\}$ are lower bounds of $\hat{\mu}_{(k),1}$, ensuring that the value of $\hat{\mu}_{(k),1}$ remains within a reasonable range as measured in terms of KL distance. Recall that $\hat{\mu}_{(k),1}$ is the empirical mean from the first $k$ times arm pulls of the optimal arm. Specifically, $\hat{\mu}_{(k),1} = \frac{1}{k}\sum_{t=1}^T r_t \mathbf{1}_{N_{t,1} < k, I_t = 1}$.

We use Lemma 19 to handle the case where all $\hat{\mu}_{(k),1}$ are restricted to this reasonable range ($\mathcal{E}(\boldsymbol{\alpha})$ is true), and Lemma 20 to address the other case ($\mathcal{E}(\boldsymbol{\alpha})$ is false). Thus, by selecting different $\boldsymbol{\alpha}$ we can use Lemma 19 to prove Propositions 17 and 18.

*Proof of Proposition 17.* Recall the definition of $\mathcal{B}_a^2$. We let all $\alpha_k, 1 \leq k \leq T$ to be $R_{\min}$, then $\mathcal{E}(\boldsymbol{\alpha})$ will not impose any additional constraints on $\hat{\mu}_{(k),1}$ except $F_{t-1,a}^c$. Therefore, we only need to apply Lemma 19 directly. When $0 < L(k) < k$,

$$
\begin{aligned}
\mathcal{B}_a^2 =& \mathbb{E}\left[\sum_{t=K+1}^{T} \mathbf{1}\left\{A_{t,a} \cap F_{t-1,a}^c\right\}\right] = \mathbb{E}\left[\sum_{t=K+1}^{T} \mathbf{1}\left\{A_{t,a} \cap F_{t-1,a}^c \cap \mathcal{E}(\boldsymbol{\alpha})\right\}\right] \\
\leq& \frac{1}{D_{\mathrm{KL}}\left(\mu_{\max} - \varepsilon_2, \mu_{\max}\right)} + \sum_{k=1}^{T} \frac{L(k)e^{-kD_{\mathrm{KL}}(\mu_{\max}-\varepsilon_2,\mu_{\max})}\left(1 - e^{-(k-L(k))D_{\mathrm{KL}}(\alpha_k,\mu_{\max}-\varepsilon_2)}\right)}{k - L(k)} \\
\leq& \frac{1}{D_{\mathrm{KL}}\left(\mu_{\max} - \varepsilon_2, \mu_{\max}\right)} + \sum_{k=1}^{T} \frac{L(k)}{k - L(k)} e^{-kD_{\mathrm{KL}}(\mu_{\max}-\varepsilon_2,\mu_{\max})} \quad \left(1 - \exp\left(-D_{\mathrm{KL}}\left(\alpha_k, \mu_{\max} - \varepsilon_2\right)\right) \leq 1\right)
\end{aligned}
$$

$\square$

*Proof of Proposition 18.* We choose $\boldsymbol{\alpha}$ such that $\forall k \geq 1, D_{\mathrm{KL}}\left(\alpha_k, x\right) \leq \frac{e\ln(T/k)}{k}$ when $\alpha_k \leq x \leq \mu_{\max} - \varepsilon_2$ and when $\mathcal{E}(\boldsymbol{\alpha})$ happens

$$
D_{\mathrm{KL}}\left(\hat{\mu}_{(k),1}, \mu_{\max} - \varepsilon_2\right) \leq \frac{e\ln(T/k)}{k}, \forall 1 \leq k \leq T.
$$

Based on whether $\mathcal{E}(\boldsymbol{\alpha})$ holds or not we can split $\mathcal{B}_a^2$ into two terms, $\mathcal{B}_a^{2,1}$ and $\mathcal{B}_a^{2,2}$, and bound them by Lemma 19 and Lemma 20, respectively,

$$
\begin{aligned}
\mathcal{B}_a^2 \leq& \mathbb{E}\left[\sum_{t=K+1}^{T} \mathbf{1}\left\{A_{t,a} \cap F_{t-1,a}^c \cap \mathcal{E}(\boldsymbol{\alpha})\right\}\right] + \mathbb{E}\left[\sum_{t=K+1}^{T} \mathbf{1}\left\{\mathcal{E}(\boldsymbol{\alpha})^c\right\}\right] \\
\leq& \underbrace{\mathbb{E}\left[\sum_{t=K+1}^{T} \mathbf{1}\left\{A_{t,a} \cap F_{t-1,a}^c \cap \mathcal{E}(\boldsymbol{\alpha})\right\}\right]}_{\mathcal{B}_a^{2,1}} + \underbrace{T\mathbb{E}\left[\mathbf{1}\left\{\mathcal{E}(\boldsymbol{\alpha})^c\right\}\right]}_{\mathcal{B}_a^{2,2}}
\end{aligned}
$$

Next, we will split the case into two cases: $0 < L(k) < k$ and $L(k) = k$ and prove each of them separately.

**When $0 < L(k) < k$:** To acquire an ideal upper bound, it suffices for us to accomplish the following two aims:

$$
\mathcal{B}_a^{2,1} \leq \frac{1}{D_{\mathrm{KL}}\left(\mu_{\max} - \varepsilon_2, \mu_{\max}\right)} + \sum_{k=1}^{T} \frac{e\,L(k)\,e^{-kD_{\mathrm{KL}}(\mu_{\max}-\varepsilon_2,\mu_{\max})}\ln(T/k)}{k}
$$

Then we apply Equation (20) from Lemma 19 to bound the above equation,

$$
\begin{aligned}
\mathcal{B}_a^{2,1} \leq & \frac{1}{D_{\mathrm{KL}}\left(\mu_{\max} - \varepsilon_2, \mu_{\max}\right)} + \sum_{k=1}^{T} \frac{L(k) e^{-k D_{\mathrm{KL}}(\mu_{\max} - \varepsilon_2, \mu_{\max})} \left(1 - e^{-(k-L(k)) D_{\mathrm{KL}}(\alpha_k, \mu_{\max} - \varepsilon_2)}\right)}{k - L(k)} \\
\leq & \frac{1}{D_{\mathrm{KL}}\left(\mu_{\max} - \varepsilon_2, \mu_{\max}\right)} + \sum_{k=1}^{T} L(k) \exp(-k D_{\mathrm{KL}}\left(\mu_{\max} - \varepsilon_2, \mu_{\max}\right)) \cdot D_{\mathrm{KL}}\left(\alpha_k, \mu_{\max} - \varepsilon_2\right) \\
& \hspace{8cm} (1 - e^{-x} \leq x \text{ when } x \geq 0) \\
\leq & \frac{1}{D_{\mathrm{KL}}\left(\mu_{\max} - \varepsilon_2, \mu_{\max}\right)} + \sum_{k=1}^{T} L(k) \exp(-k D_{\mathrm{KL}}\left(\mu_{\max} - \varepsilon_2, \mu_{\max}\right)) \cdot \frac{e \ln(T/k)}{k} \\
& \hspace{8cm} (\text{Recall the definition of } \alpha_k) \\
\leq & \frac{1}{D_{\mathrm{KL}}\left(\mu_{\max} - \varepsilon_2, \mu_{\max}\right)} + \sum_{k=1}^{T} \frac{e\, L(k)\, e^{-k D_{\mathrm{KL}}(\mu_{\max} - \varepsilon_2, \mu_{\max})} \ln(T/k)}{k}
\end{aligned}
$$

For $\mathcal{B}_a^{2,2}$, we apply Lemma 20 for each summand by setting $\varepsilon = \varepsilon_2$,

$$
\mathcal{B}_a^{2,2} \leq \frac{5}{D_{\mathrm{KL}}\left(\mu_{\max} - \varepsilon_2, \mu_{\max}\right)}
$$

Then we combine the upper bounds of $\mathcal{B}_a^{2,1}$ and $\mathcal{B}_a^{2,2}$, and get the following bound on $\mathcal{B}_a^2$

$$
\begin{aligned}
\mathcal{B}_a^2 \leq & \mathcal{B}_a^{2,1} + \mathcal{B}_a^{2,2} \\
\leq & \frac{1}{D_{\mathrm{KL}}\left(\mu_{\max} - \varepsilon_2, \mu_{\max}\right)} + \sum_{k=1}^{T} \frac{e\, L(k)\, e^{-k D_{\mathrm{KL}}(\mu_{\max} - \varepsilon_2, \mu_{\max})} \ln(T/k)}{k} + \frac{5}{D_{\mathrm{KL}}\left(\mu_{\max} - \varepsilon_2, \mu_{\max}\right)} \\
\leq & \frac{6}{D_{\mathrm{KL}}\left(\mu_{\max} - \varepsilon_2, \mu_{\max}\right)} + \sum_{k=1}^{T} \frac{e\, L(k)}{k} \exp(-k D_{\mathrm{KL}}\left(\mu_{\max} - \varepsilon_2, \mu_{\max}\right)) \cdot \ln(T/k)
\end{aligned}
$$

**When $L(k) = k$:** We still do the same splitting, and use Equation (21) from Lemma 19 to bound $\mathcal{B}_a^{2,1}$ as

$$
\begin{aligned}
\mathcal{B}_a^{2,1} \leq & \frac{1}{D_{\mathrm{KL}}\left(\mu_{\max} - \varepsilon_2, \mu_{\max}\right)} + \sum_{k=1}^{T} \frac{k D_{\mathrm{KL}}\left(\alpha_k, \mu_{\max} - \varepsilon_2\right)}{\exp\left(-k D_{\mathrm{KL}}\left(\mu_{\max} - \varepsilon_2, \mu_{\max}\right)\right)} \\
\leq & \frac{1}{D_{\mathrm{KL}}\left(\mu_{\max} - \varepsilon_2, \mu_{\max}\right)} + \sum_{k=1}^{T} \frac{e \ln(T/k)}{D_{\mathrm{KL}}\left(\mu_{\max} - \varepsilon_2, \mu_{\max}\right)} & (\text{By the definition of } \alpha_k) \\
\leq & \frac{1}{D_{\mathrm{KL}}\left(\mu_{\max} - \varepsilon_2, \mu_{\max}\right)} + \frac{5e \ln(T D_{\mathrm{KL}}\left(\mu_{\max} - \varepsilon_2, \mu_{\max}\right) \vee e)}{D_{\mathrm{KL}}\left(\mu_{\max} - \varepsilon_2, \mu_{\max}\right)} & (\text{Lemma 27})
\end{aligned}
$$

We can derive the final result by

$$
\begin{aligned}
\mathcal{B}_a^2 \leq & \mathcal{B}_a^{2,1} + \mathcal{B}_a^{2,2} \\
\leq & \frac{1}{D_{\mathrm{KL}}\left(\mu_{\max} - \varepsilon_2, \mu_{\max}\right)} + \frac{5e \ln(T D_{\mathrm{KL}}\left(\mu_{\max} - \varepsilon_2, \mu_{\max}\right) \vee e)}{D_{\mathrm{KL}}\left(\mu_{\max} - \varepsilon_2, \mu_{\max}\right)} + \frac{5}{D_{\mathrm{KL}}\left(\mu_{\max} - \varepsilon_2, \mu_{\max}\right)} \\
= & \frac{6 + 5e \ln(T D_{\mathrm{KL}}\left(\mu_{\max} - \varepsilon_2, \mu_{\max}\right) \vee e)}{D_{\mathrm{KL}}\left(\mu_{\max} - \varepsilon_2, \mu_{\max}\right)}
\end{aligned}
$$

$\square$

**D.4 On the difficulty of showing asymptotic optimality of $L(k) = k$.**

Readers may have noticed that the both Propositions 17 and 18 bound the same term $\mathcal{B}_a^2$, but they serve different purposes: the former is for showing asymptotic optimality and the latter is for showing minimax optimality. In order to show asymptotic optimality, we need to bound $\mathcal{B}_a^2$ by a quantity independent of $T$, which Proposition 18 fails to show and we need Proposition 17 instead. However, in proving Proposition 17 we need to take $L(k) < k$ rather than $L(k) = k$ because the integrand after applying the probability transfer upper bound can diverge at $L(k) = k$ for some OPED families, leading to a vacuous bound on $\mathcal{B}_a^2$. Specifically, after step 1 (applying the probability transferring lemma), we bound each summand of $\mathcal{B}_a^2$ by the summand of the RHS of Eq. (22):

$$\mathbb{E}\left[\mathbf{1}\left\{R_{\min} \leq \hat{\mu}_{(k),1} \leq \mu_{\max} - \varepsilon_2\right\} \cdot \exp\left(L(k)D_{\mathrm{KL}}\left(\hat{\mu}_{(k),1}, \mu_{\max} - \varepsilon_2\right)\right)\right]$$
$$= \int_{R_{\min}}^{\mu_{\max}-\varepsilon_2} p_k(x) \cdot \exp\left(L(k)D_{\mathrm{KL}}\left(x, \mu_{\max} - \varepsilon_2\right)\right) dx.$$

where $p_k(x)$ is the probability density function of $\hat{\mu}_{(k),1}$. In the Bernoulli setting, the integrand stays bounded when $L(k) = k$. For general OPED families, the integrand can blow up as $x$ approaches $R_{\min}$ when $L(k) = k$. For example, if rewards are from exponential distributions, the support set of the integrand is $[0, \mu_{\max} - \varepsilon_2)$ and KL-divergence between two exponential distributions with means $\mu < \nu$ is $D_{\mathrm{KL}}(\mu, \nu) = \frac{\mu}{\nu} - 1 - \log(\frac{\mu}{\nu})$, which behaves like $\log \mu$ as $\mu \to 0^+$. The random variable $k\hat{\mu}_{(k),1}$ is Gamma distributed with shape parameter $k$ and scale parameter $\mu_{\max}$. Plugging the density function and KL-divergence into the integrand shows that the integrand behaves like

$$I_k = \int_0^{\mu_{\max}-\varepsilon_2} \frac{x^{k-1}\exp\left(-\frac{kx}{\mu_{\max}}\right)}{\Gamma(k)\mu_{\max}^k} \cdot \exp\left(L(k)\left(\frac{x}{\mu_{\max}-\varepsilon_2} - 1 - \log\frac{x}{\mu_{\max}-\varepsilon_2}\right)\right) dx$$
$$\text{(Ignore the constant from density)}$$
$$= \int_0^{\mu_{\max}-\varepsilon_2} x^{k-1} \cdot \left(\frac{\mu_{\max}-\varepsilon_2}{x}\right)^{L(k)} \cdot \frac{1}{\Gamma(k)\mu_{\max}^k}\exp\left(-\frac{kx}{\mu_{\max}} + L(k)\left(\frac{x}{\mu_{\max}-\varepsilon_2} - 1\right)\right) dx$$

When $L(k) = k$, the integrand behaves like $x^{k-1-L(k)}$ as $x \to 0^+$, which is not integrable and leads to divergence of $I_k$.

# E Supporting Lemmas

In this section, we present two key lemmas that are used to prove all the propositions in Appendix D, along with other auxiliary lemmas in Appendix E.3.

Lemma 19 addresses the case where all estimates are well-bounded. Specifically, it is used to bound the probability that all estimates of arm 1, from $t = 1$ to $t = T$, remain within a bounded range.

On the other hand, Lemma 20 deals with the case where at least one $\hat{\mu}_{t,1}$, for $t = 1$ to $t = T$, falls outside the interval. This lemma is used to bound the probability that at least one estimate of an arm, from $t = 1$ to $t = T$, lies outside a restricted interval.

Additionally, Lemma 25 provides a lower bound for the KL divergence. The other lemmas are considered folklore results, and their proofs will be provided as needed.

### E.1 All Estimates of the Optimal Arm Are Restricted to Limited Intervals

**Lemma 19.** *Suppose we have a series of real values $\boldsymbol{\alpha} = \{\alpha_k\}_{k=1}^{T}$ and $\alpha_k \leq \mu_{\max} - \varepsilon_2, \forall 1 \leq k \leq T$. Under Assumptions 1 and 3, when $0 < L(k) < k$ we have:*

$$
B_a^{2,1} = \mathbb{E}\left[\sum_{t=K+1}^{T} \mathbf{1}\left\{A_{t,a} \cap F_{t-1,a}^c \cap \mathcal{E}(\boldsymbol{\alpha})\right\}\right]
$$

$$
\leq \frac{1}{D_{\mathrm{KL}}\left(\mu_{\max} - \varepsilon_2, \mu_{\max}\right)} + \sum_{k=1}^{T} \frac{L(k)e^{-kD_{\mathrm{KL}}(\mu_{\max} - \varepsilon_2, \mu_{\max})}\left(1 - e^{-(k-L(k))D_{\mathrm{KL}}(\alpha_k, \mu_{\max} - \varepsilon_2)}\right)}{k - L(k)} \tag{20}
$$

*and when $L(k) = k$, we have:*

$$
\mathbb{E}\left[\sum_{t=K+1}^{T} \mathbf{1}\left\{A_{t,a} \cap F_{t-1,a}^c \cap \mathcal{E}(\boldsymbol{\alpha})\right\}\right] \leq \frac{1}{D_{\mathrm{KL}}\left(\mu_{\max} - \varepsilon_2, \mu_{\max}\right)} + \sum_{k=1}^{T} \frac{kD_{\mathrm{KL}}\left(\alpha_k, \mu_{\max} - \varepsilon_2\right)}{e^{-kD_{\mathrm{KL}}(\mu_{\max} - \varepsilon_2, \mu_{\max})}} \tag{21}
$$

*Proof.* Recall the notations $A_{t,a} = \{I_t = a\}$ and $F_{t-1,a}^c = \{\hat{\mu}_{t-1,\max} < \mu_{\max} - \varepsilon_2\}$.

**Step 1: Applying probability transferring lemma** Starting from the LHS of Equation (20),

$$
\mathbb{E}\left[\sum_{t=K+1}^{T} \mathbf{1}\left\{A_{t,a} \cap F_{t-1,a}^c \cap \mathcal{E}(\boldsymbol{\alpha})\right\}\right] \leq \mathbb{E}\left[\sum_{t=K+1}^{T} \mathbf{1}\left\{A_{t,a} \cap F_{t-1,a}^c \cap \left\{\hat{\mu}_{t-1,1} \geq \alpha_{N_{t-1,1}}\right\}\right\}\right]
$$

(Drop the event $\mathcal{E}(\boldsymbol{\alpha})$ to $\left\{\hat{\mu}_{t-1,1} \geq \alpha_{N_{t-1,1}}\right\}$ to make it easier to apply the probability transferring lemma.)

$$
= \mathbb{E}\left[\sum_{t=K+1}^{T} \mathbf{1}\left\{F_{t-1,a}^c \cap \left\{\hat{\mu}_{t-1,1} \geq \alpha_{N_{t-1,1}}\right\}\right\} \cdot \mathbb{E}\left[\mathbf{1}\left\{A_{t,a}\right\} \mid \mathcal{H}_{t-1}\right]\right] \quad \text{(Law of total expectation)}
$$

$$
\leq \mathbb{E}\left[\sum_{t=K+1}^{T} \mathbf{1}\left\{F_{t-1,a}^c \cap \left\{\hat{\mu}_{t-1,1} \geq \alpha_{N_{t-1,1}}\right\}\right\} \cdot \exp\left(L(N_{t-1,1}) \cdot D_{\mathrm{KL}}\left(\hat{\mu}_{t-1,1}, \hat{\mu}_{t-1,\max}\right)\right)\mathbb{E}\left[\mathbf{1}\left\{A_{t,1}\right\} \mid \mathcal{H}_{t-1}\right]\right]
$$

(By the probability transferring Lemma (Lemma 21))

$$
\leq \mathbb{E}\left[\sum_{t=K+1}^{T} \mathbf{1}\left\{A_{t,1} \cap F_{t-1,a}^c \cap \left\{\hat{\mu}_{t-1,1} \geq \alpha_{N_{t-1,1}}\right\}\right\} \cdot \exp\left(L(N_{t-1,1}) \cdot D_{\mathrm{KL}}\left(\hat{\mu}_{t-1,1}, \hat{\mu}_{t-1,\max}\right)\right)\right]
$$

(Law of total expectation)

$$
\leq \mathbb{E}\left[\sum_{k=2}^{T} \mathbf{1}\left\{\alpha_{k-1} \leq \hat{\mu}_{(k-1),1} \leq \mu_{\max} - \varepsilon_2\right\} \cdot \exp\left(L(k-1) \cdot D_{\mathrm{KL}}\left(\hat{\mu}_{\tau_1(k)-1,1}, \hat{\mu}_{\tau_1(k)-1,\max}\right)\right)\right]
$$

(Only when the first arm is pulled ($t = \tau_1(k)$ for some $k$) the indicator function is non-zero.)

$$
\leq \mathbb{E}\left[\sum_{k=1}^{T} \mathbf{1}\left\{\alpha_k \leq \hat{\mu}_{(k),1} \leq \mu_{\max} - \varepsilon_2\right\} \cdot \exp\left(L(k) \cdot D_{\mathrm{KL}}\left(\hat{\mu}_{(k),1}, \mu_{\max} - \varepsilon_2\right)\right)\right] \quad \text{(shift index $k$ by 1)}
$$

$$
\tag{22}
$$

**Step 2: Double integration trick** Continuing from Equation (22) we can do an integral calculation by using a double integration trick to simplify the integral. Let $f_k(x) = \exp\left(L(k) \cdot D_{\mathrm{KL}}\left(x, \mu_{\max} - \varepsilon_2\right)\right)$ and $p_k(\cdot)$ be the PDF of $\hat{\mu}_{(k),1}$. When $\hat{\mu}_{(k),1}$ has a discrete probability distribution, the double integration trick can still be applied by replacing the inner integral with summation over the support of $[\hat{\mu}_{(k),1}, \mu_{\max} - \varepsilon_2]$ and $[x, \mu_{\max} - \varepsilon_2]$. For simplicity, we only provide the proof when $\hat{\mu}_{(k),1}$ has a continuous probability distribution, and the proof for a discrete distribution is similar. By applying a double integration trick, Equation (22) becomes

$$\mathcal{B}_a^{2,1} \le \mathbb{E}\left[\sum_{k=1}^{T} \mathbf{1}\left\{\alpha_k \le \hat{\mu}_{(k),1} < \mu_{\max} - \varepsilon_2\right\} \cdot f_k(\hat{\mu}_{(k),1})\right] = \sum_{k=1}^{T} \int_{\alpha_k}^{\mu_{\max} - \varepsilon_2} f_k(x) p_k(x)\, dx$$

$$= \sum_{k=1}^{T} \int_{\alpha_k}^{\mu_{\max} - \varepsilon_2} p_k(x)\left(f_k(\mu_{\max} - \varepsilon_2) - \int_{x}^{\mu_{\max} - \varepsilon_2} f_k'(y)\, dy\right) dx$$

$$\left(f_k(x) = f_k(\mu_{\max} - \varepsilon_2) - \int_x^{\mu_{\max} - \varepsilon_2} f_k'(y)\, dy\right)$$

$$= \sum_{k=1}^{T} \int_{\alpha_k}^{\mu_{\max} - \varepsilon_2} p_k(x) f_k(\mu_{\max} - \varepsilon_2)\, dx + \sum_{k=1}^{T} \int_{\alpha_k}^{\mu_{\max} - \varepsilon_2} \int_{x}^{\mu_{\max} - \varepsilon_2} p_k(x)\left(-f_k'(y)\right) dy\, dx$$

$$= \underbrace{\sum_{k=1}^{T} \int_{\alpha_k}^{\mu_{\max} - \varepsilon_2} p_k(x)\, dx}_{A} + \underbrace{\sum_{k=1}^{T} \int_{\alpha_k}^{\mu_{\max} - \varepsilon_2} \int_{\alpha_k}^{y} p_k(x)\left(-f_k'(y)\right) dx\, dy}_{B}$$

$$(f_k(\mu_{\max} - \varepsilon_2) = 1 \text{ and exchange the order of integral})$$

**For $A$:**

$$A = \sum_{k=1}^{T} \int_{\alpha_k}^{\mu_{\max} - \varepsilon_2} p_k(x)\, dx \le \sum_{k=1}^{\infty} \int_{\alpha_k}^{\mu_{\max} - \varepsilon_2} p_k(x)\, dx \le \sum_{k=1}^{\infty} \exp\left(-k \cdot D_{\mathrm{KL}}\left(\mu_{\max} - \varepsilon_2, \mu_{\max}\right)\right)$$

$$\text{(By Lemma 26)}$$

$$= \frac{\exp\left(-D_{\mathrm{KL}}\left(\mu_{\max} - \varepsilon_2, \mu_{\max}\right)\right)}{1 - \exp\left(-D_{\mathrm{KL}}\left(\mu_{\max} - \varepsilon_2, \mu_{\max}\right)\right)} = \frac{1}{\exp\left(D_{\mathrm{KL}}\left(\mu_{\max} - \varepsilon_2, \mu_{\max}\right)\right) - 1} \qquad \text{(Geometric sum)}$$

$$\le \frac{1}{D_{\mathrm{KL}}\left(\mu_{\max} - \varepsilon_2, \mu_{\max}\right)} \qquad (e^x \ge x + 1 \text{ when } x \ge 0)$$

$$\tag{23}$$

**For $B$** Notice that the derivative $\frac{dD_{\mathrm{KL}}(y, \mu_{\max} - \varepsilon_2)}{dy}$ derived from $f_k'(y)$ is negative when $y \le \mu_{\max} - \varepsilon_2$ the term $B$ is still positive. When $0 < L(k) < k$,

$$B = \sum_{k=1}^{T} \int_{\alpha_k}^{\mu_{\max} - \varepsilon_2} \int_{\alpha_k}^{y} p_k(x)\left(-f_k'(y)\right) dx\, dy$$

$$= \sum_{k=1}^{T} \int_{\alpha_k}^{\mu_{\max} - \varepsilon_2} \mathbb{P}(\alpha_k \le \hat{\mu}_{(k),1} \le y) \cdot \left(-f_k(y)\right) L(k) \frac{dD_{\mathrm{KL}}\left(y, \mu_{\max} - \varepsilon_2\right)}{dy}\, dy$$

$$\text{(Calculate the derivative and inner integral)}$$

$$\le \sum_{k=1}^{T} \int_{\alpha_k}^{\mu_{\max} - \varepsilon_2} e^{-kD_{\mathrm{KL}}(y, \mu_{\max})} \cdot \left(-f_k(y)\right) L(k) \frac{dD_{\mathrm{KL}}\left(y, \mu_{\max} - \varepsilon_2\right)}{dy}\, dy \qquad \text{(Apply Lemma 26)}$$

$$= \sum_{k=1}^{T} \int_{\alpha_k}^{\mu_{\max} - \varepsilon_2} e^{-kD_{\mathrm{KL}}(y, \mu_{\max}) + L(k)D_{\mathrm{KL}}(y, \mu_{\max} - \varepsilon_2)} \cdot \left(-L(k)\right) \frac{dD_{\mathrm{KL}}\left(y, \mu_{\max} - \varepsilon_2\right)}{dy}\, dy$$

$$\le \sum_{k=1}^{T} \int_{\alpha_k}^{\mu_{\max} - \varepsilon_2} e^{-kD_{\mathrm{KL}}(\mu_{\max} - \varepsilon_2, \mu_{\max}) - (k - L(k))D_{\mathrm{KL}}(y, \mu_{\max} - \varepsilon_2)} \cdot \left(-L(k)\right) \frac{dD_{\mathrm{KL}}\left(y, \mu_{\max} - \varepsilon_2\right)}{dy}\, dy$$

$$\text{(Apply Lemma 23)}$$

$$= \sum_{k=1}^{T} \frac{L(k) e^{-kD_{\mathrm{KL}}(\mu_{\max} - \varepsilon_2, \mu_{\max})} e^{-(k - L(k))D_{\mathrm{KL}}(y, \mu_{\max} - \varepsilon_2)}}{k - L(k)} \Big|_{\alpha_k}^{\mu_{\max} - \varepsilon_2}$$

$$= \sum_{k=1}^{T} \frac{L(k) e^{-kD_{\mathrm{KL}}(\mu_{\max} - \varepsilon_2, \mu_{\max})} \left(1 - e^{-(k - L(k))D_{\mathrm{KL}}(\alpha_k, \mu_{\max} - \varepsilon_2)}\right)}{k - L(k)}$$

$$\tag{24}$$

where in the first inequality we apply Lemma 26 to bound $\mathbb{P}(\alpha_k \leq \hat{\mu}_{(k),1} \leq y)$. In the second inequality, we apply Bregman Divergence Identity (Lemma 23). Since $\alpha_k \leq y \leq \mu_{\max} - \varepsilon_2$, we have $D_{\mathrm{KL}}(y, \mu_{\max}) \geq D_{\mathrm{KL}}(y, \mu_{\max} - \varepsilon_2) + D_{\mathrm{KL}}(\mu_{\max} - \varepsilon_2, \mu_{\max})$.

When $L(k) = k$, we can reuse the above analysis until the last inequality,

$$
\begin{aligned}
B &\leq \sum_{k=1}^{T} \int_{\alpha_k}^{\mu_{\max} - \varepsilon_2} \exp\left(-k D_{\mathrm{KL}}(\mu_{\max} - \varepsilon_2, \mu_{\max})\right) \cdot (-k) \frac{\mathrm{d} D_{\mathrm{KL}}(y, \mu_{\max} - \varepsilon_2)}{\mathrm{d}y} \, \mathrm{d}y \\
&= \sum_{k=1}^{T} k \exp\left(-k D_{\mathrm{KL}}(\mu_{\max} - \varepsilon_2, \mu_{\max})\right) D_{\mathrm{KL}}(y, \mu_{\max} - \varepsilon_2) \big|_{\mu_{\max} - \varepsilon_2}^{\alpha_k} \\
&= \sum_{k=1}^{T} \frac{k D_{\mathrm{KL}}(\alpha_k, \mu_{\max} - \varepsilon_2)}{\exp\left(-k D_{\mathrm{KL}}(\mu_{\max} - \varepsilon_2, \mu_{\max})\right)}
\end{aligned}
\tag{25}
$$

Based on Equations (23) to (25), we obtain the final conclusion that when $0 < L(k) < k$,

$$
\mathbb{E}\left[ \sum_{t=K+1}^{T} \mathbf{1}\left\{ A_{t,a} \cap F_{t-1,a}^c \cap \mathcal{E}(\boldsymbol{\alpha}) \right\} \right]
$$
$$
\leq \frac{1}{D_{\mathrm{KL}}(\mu_{\max} - \varepsilon_2, \mu_{\max})} + \sum_{k=1}^{T} \frac{L(k) e^{-k D_{\mathrm{KL}}(\mu_{\max} - \varepsilon_2, \mu_{\max})} \left(1 - e^{-(k - L(k)) D_{\mathrm{KL}}(\alpha_k, \mu_{\max} - \varepsilon_2)}\right)}{k - L(k)}
$$

and when $L(k) = k$,

$$
\mathbb{E}\left[ \sum_{t=K+1}^{T} \mathbf{1}\left\{ A_{t,a} \cap F_{t-1,a}^c \cap \mathcal{E}(\boldsymbol{\alpha}) \right\} \right] \leq \frac{1}{D_{\mathrm{KL}}(\mu_{\max} - \varepsilon_2, \mu_{\max})} + \sum_{k=1}^{T} \frac{k D_{\mathrm{KL}}(\alpha_k, \mu_{\max} - \varepsilon_2)}{\exp\left(-k D_{\mathrm{KL}}(\mu_{\max} - \varepsilon_2, \mu_{\max})\right)}
$$

$\qquad\qquad\qquad\qquad\qquad\qquad\qquad\qquad\qquad\qquad\qquad\qquad\qquad\qquad\qquad\qquad$ $\square$

## E.2 Bounding the Deviation of Mean Estimation Exceeding the Threshold

We borrow Lemma 19 from Lemma 3.2 in Jin et al. (2023) and make a slight modification to the statement to better align with our requirements. We provide the full proof here, as the proof in Jin et al. (2023) relies on an assumption regarding the upper bound on the variance of the reward distribution, which may not hold in our setting.

**Lemma 20.** *Suppose we have a random variable $X$ following distribution $\nu$ with mean $\mu$ from an OPED family $\mathcal{F}_m$. Assume that Assumptions 1 and 3 hold. We have collected a sequence of sample $\{X_i\}_{i=1}^{T}$ draw i.i.d. from $\nu$. Denote $\sum_{i=1}^{s} X_i / s$ as $\hat{\mu}_s$.*

*We have the inequality,*

$$
\mathbb{P}\left( \exists 1 \leq s \leq T : \hat{\mu}_s \leq \mu - \varepsilon, D_{\mathrm{KL}}(\hat{\mu}_s, \mu - \varepsilon) \geq \frac{e \ln(T/s)}{s} \right) \leq \frac{5}{T D_{\mathrm{KL}}(\mu - \varepsilon, \mu)}
$$

*Proof.* Based on Lemma 23 and the monotonicity of the natural parameter with respect to mean parameter, under the condition $\hat{\mu}_s \leq \mu - \varepsilon$ and $\varepsilon \geq 0$, we have $D_{\mathrm{KL}}(\hat{\mu}_s, \mu - \varepsilon) \leq D_{\mathrm{KL}}(\hat{\mu}_s, \mu) - D_{\mathrm{KL}}(\mu - \varepsilon, \mu)$, and thus,

$$
\mathbb{P}\left( \exists s : 1 \leq s \leq T, \hat{\mu}_s \leq \mu - \varepsilon, D_{\mathrm{KL}}(\hat{\mu}_s, \mu - \varepsilon) \geq \frac{e \ln(T/s)}{s} \right)
$$
$$
\leq \mathbb{P}\left( \exists s : 1 \leq s \leq T, \hat{\mu}_s \leq \mu - \varepsilon, D_{\mathrm{KL}}(\hat{\mu}_s, \mu) - D_{\mathrm{KL}}(\mu - \varepsilon, \mu) \geq \frac{e \ln(T/s)}{s} \right)
$$

Then we apply the peeling device $\frac{T}{e^{n+1}} < s \leq \frac{T}{e^n}$ to give an upper bound to the above equation

$$\mathbb{P}\left(\exists s : 1 \leq s \leq T, \hat{\mu}_s \leq \mu - \varepsilon, D_{\text{KL}}\left(\hat{\mu}_s, \mu\right) - D_{\text{KL}}\left(\mu - \varepsilon, \mu\right) \geq \frac{e \ln(T/s)}{s}\right)$$

$$\leq \sum_{n=0}^{\infty} \mathbb{P}\left(\exists s : s \in \mathbb{N}^+ \bigcap (\frac{T}{e^{n+1}}, \frac{T}{e^n}], \hat{\mu}_s \leq \mu - \varepsilon, D_{\text{KL}}\left(\hat{\mu}_s, \mu\right) - D_{\text{KL}}\left(\mu - \varepsilon, \mu\right) \geq \frac{e \ln(T/s)}{s}\right)$$

$$\leq \sum_{n=0}^{\infty} \underbrace{\mathbb{P}\left(\exists s : s \in \mathbb{N}^+ \bigcap (\frac{T}{e^{n+1}}, \frac{T}{e^n}], \hat{\mu}_s \leq \mu - \varepsilon, D_{\text{KL}}\left(\hat{\mu}_s, \mu\right) - D_{\text{KL}}\left(\mu - \varepsilon, \mu\right) \geq \frac{ne^{n+1}}{T}\right)}_{:=a_n}$$

$$\left(\text{For each case, } s \leq \frac{T}{e^n} \Rightarrow \frac{e \ln(T/s)}{s} \geq \frac{ne^{n+1}}{T}\right) \tag{26}$$

Here we need to discuss several cases:

1. $n > \ln(T)$.

   In this case $n > \ln(T) \implies \frac{T}{e^n} < 1 \implies \mathbb{N}^+ \bigcap (\frac{T}{e^{n+1}}, \frac{T}{e^n}] = \emptyset$. Then we can bound $a_n$ by 0 since there is no valid choice of $s$.

2. $\ln(T) - 1 < n \leq \ln(T)$.

   In this case, the condition implies that $\frac{T}{e^n} \geq 1$ and $\frac{T}{e^{n+1}} < 1$ and the interval $(\frac{T}{e^{n+1}}, \frac{T}{e^n}]$ contains at most two integers 1 and 2.

3. $n \leq \ln(T) - 1$

   The above inequality implies that $\frac{T}{e^{n+1}} \geq 1$.

Then the summation of $n$ from 0 to $+\infty$ is equivalent to the sum from 0 to $\lfloor \ln(T) \rfloor$.

$$(26) = \sum_{n=0}^{\lfloor \ln(T) \rfloor} \mathbb{P}\left(\exists s : s \in \mathbb{N}^+ \bigcap (\frac{T}{e^{n+1}}, \frac{T}{e^n}], D_{\text{KL}}\left(\hat{\mu}_s, \mu\right) - D_{\text{KL}}\left(\mu - \varepsilon, \mu\right) \geq \frac{ne^{n+1}}{T}\right) + \sum_{n=\lfloor \ln(T) \rfloor + 1}^{\infty} 0$$

$$\leq \sum_{n=0}^{\lfloor \ln(T) \rfloor} \mathbb{P}\left(\exists s \geq \lceil \frac{T}{e^{n+1}} \rceil, D_{\text{KL}}\left(\hat{\mu}_s, \mu\right) - D_{\text{KL}}\left(\mu - \varepsilon, \mu\right) \geq \frac{ne^{n+1}}{T}\right)$$

$$\leq \sum_{n=0}^{\lfloor \ln(T) \rfloor} \exp\left(-\lceil \frac{T}{e^{n+1}} \rceil \cdot \left(\frac{ne^{n+1}}{T} + D_{\text{KL}}\left(\mu - \varepsilon, \mu\right)\right)\right) \qquad \text{(Maximal Inequality, Lemma 26)}$$

$$\leq \sum_{n=0}^{\infty} \exp\left(-n - \frac{T D_{\text{KL}}\left(\mu - \varepsilon, \mu\right)}{e^{n+1}}\right) = \sum_{n=0}^{\infty} \frac{1}{e^n} \exp\left(-\frac{T D_{\text{KL}}\left(\mu - \varepsilon, \mu\right)}{e^{n+1}}\right)$$

$$\leq \int_0^{\infty} \frac{1}{e^x} \exp\left(-\frac{T D_{\text{KL}}\left(\mu - \varepsilon, \mu\right)}{e^{x+1}}\right) \mathrm{d}x + \frac{1}{T D_{\text{KL}}\left(\mu - \varepsilon, \mu\right)}$$

$$\leq \frac{e}{T D_{\text{KL}}\left(\mu - \varepsilon, \mu\right)} \exp\left(-\frac{T D_{\text{KL}}\left(\mu - \varepsilon, \mu\right)}{e^{x+1}}\right) |_{x=0}^{x=\infty} + \frac{1}{T D_{\text{KL}}\left(\mu - \varepsilon, \mu\right)}$$

$$\text{(Integral and } e^x \geq x \text{ when } x > 0)$$

$$= \frac{e}{T D_{\text{KL}}\left(\mu - \varepsilon, \mu\right)} \left(1 - \exp\left(-\frac{T D_{\text{KL}}\left(\mu - \varepsilon, \mu\right)}{e}\right)\right) + \frac{1}{T D_{\text{KL}}\left(\mu - \varepsilon, \mu\right)} \qquad \text{(Algebra)}$$

$$\leq \frac{5}{T D_{\text{KL}}\left(\mu - \varepsilon, \mu\right)}$$

The first inequality relaxes the range of $s$ from $(\frac{T}{e^{n+1}}, \frac{T}{e^n}]$ to $(\frac{T}{e^{n+1}}, \infty]$. The second inequality uses Lemma 26 where for each $n$, we apply Lemma 26 once by setting $N = \lceil \frac{T}{e^{n+1}} \rceil$ and $y$ to be $\frac{ne^{n+1}}{T} + D_{\mathrm{KL}}(\mu - \varepsilon, \mu)$. In the third inequality, we remove the ceiling function. The fourth inequality uses $\sum_{x=a}^{b} f(x) \le \int_a^b f(x)\, \mathrm{d}x + \max_{x \in [a,b]} f(x)$ when $f(x)$ is unimodal. We let $f(x) = \frac{1}{e^x}\exp\left(-\frac{T}{e^{x+1}}D_{\mathrm{KL}}(\mu - \varepsilon, \mu)\right)$. $f(x)$ is unimodal since $g(z) = aze^{-bz}$ is unimodal when $a, b > 0$ and $e^{-x} \mapsto z$ is monotonic. For the last inequality, we relax $\left(1 - \exp\left(-\frac{TD_{\mathrm{KL}}(\mu - \varepsilon, \mu)}{e}\right)\right)$ to 1. $\qquad\square$

### E.3  Other Auxiliary Lemmas

### E.3.1  Probability Transferring

**Lemma 21** (Probability Transferring Lemma). *Suppose Algorithm 1 is run. Let $\mathcal{H}_{t-1}$ denote the $\sigma$-field derived from the historical path up to and including time $t-1$, which is represented as $\sigma\left(\{I_i, r_i\}_{i=1}^{t-1}\right)$ (where $I_i$ indicates the arm pulled at time round $i$ and $r_i$ is the corresponding reward). Then,*

$$\mathbb{P}(I_t = a | \mathcal{H}_{t-1}) \le \exp\left(L(N_{t-1,1})D_{\mathrm{KL}}(\hat{\mu}_{t-1,1}, \hat{\mu}_{t-1,\max})\right)\mathbb{P}(I_t = 1 \mid \mathcal{H}_{t-1}) \qquad (27)$$

*Proof.* To prove Equation (27), recall the algorithm setting, we have the following relationship

$$\mathbb{P}(I_t = a | \mathcal{H}_{t-1}) = \frac{\exp\left(-L(N_{t-1,a})D_{\mathrm{KL}}(\hat{\mu}_{t-1,a}, \hat{\mu}_{t-1,\max})\right)}{\exp\left(-L(N_{t-1,1})D_{\mathrm{KL}}(\hat{\mu}_{t-1,1}, \hat{\mu}_{t-1,\max})\right)} \cdot \mathbb{P}(I_t = 1 \mid \mathcal{H}_{t-1})$$

$$\le \frac{\mathbb{P}(I_t = 1 \mid \mathcal{H}_{t-1})}{\exp\left(-L(N_{t-1,1})D_{\mathrm{KL}}(\hat{\mu}_{t-1,1}, \hat{\mu}_{t-1,\max})\right)} = \exp\left(L(N_{t-1,1})D_{\mathrm{KL}}(\hat{\mu}_{t-1,1}, \hat{\mu}_{t-1,\max})\right)\mathbb{P}(I_t = 1 \mid \mathcal{H}_{t-1})$$

where the inequality is due to $D_{\mathrm{KL}}(\hat{\mu}_{t-1,a}, \hat{\mu}_{t-1,\max}) \ge 0 \Rightarrow \exp\left(-L(N_{t-1,a})D_{\mathrm{KL}}(\hat{\mu}_{t-1,a}, \hat{\mu}_{t-1,\max})\right) \le 1$. $\quad\square$

### E.3.2  Properties of KL Divergence in OPED Family

**Lemma 22.** *(Harremoës, 2017) Let $\mu$ and $\mu'$ be the mean values of two distributions in $\mathcal{F}_m$. The Kullback-Leibler divergence between them satisfies:*

$$D_{\mathrm{KL}}(\mu, \mu') = \int_{\mu}^{\mu'} \frac{x - \mu}{V(x)}\, \mathrm{d}x,$$

*recall that $V(x) := b''((b')^{-1}(x))$ is the variance of the distribution in $\mathcal{F}_m$ with mean parameter $x$.*

**Lemma 23** (Bregman Divergence Identity). *Suppose we have three distributions in $\mathcal{F}_m$ with model parameter $\theta_a, \theta_b$ and $\theta_c$, and their means are $\mu_a, \mu_b$ and $\mu_c$, respectively. Then we have the following relationship*

$$D_{\mathrm{KL}}(\mu_a, \mu_b) + D_{\mathrm{KL}}(\mu_b, \mu_c) = D_{\mathrm{KL}}(\mu_a, \mu_c) - (\mu_b - \mu_a)(\theta_c - \theta_b)$$

**Remark 24.** *When $\mu_a \le \mu_b \le \mu_c$, based on the Bregman Divergence Identity (Lemma 23) we have*

$$D_{\mathrm{KL}}(\mu_a, \mu_b) + D_{\mathrm{KL}}(\mu_b, \mu_c) \le D_{\mathrm{KL}}(\mu_a, \mu_c),$$

*since $(\mu_b - \mu_a)(\theta_c - \theta_b)$ is non-negative.*

*Proof.* According to Equation (3), there are

$$D_{\mathrm{KL}}(\mu_a, \mu_b) = b(\theta_b) - b(\theta_a) - \mu_a(\theta_b - \theta_a)$$

$$D_{\mathrm{KL}}(\mu_a, \mu_c) = b(\theta_c) - b(\theta_a) - \mu_a(\theta_c - \theta_a)$$

therefore,

$$
\begin{aligned}
&D_{\mathrm{KL}}\left(\mu_a, \mu_b\right) - D_{\mathrm{KL}}\left(\mu_a, \mu_c\right) \\
=\, & b(\theta_b) - b(\theta_a) - \mu_a\left(\theta_b - \theta_a\right) - b(\theta_c) + b(\theta_a) + \mu_a\left(\theta_c - \theta_a\right) \\
=\, & b(\theta_b) - b(\theta_c) - \mu_a\left(\theta_b - \theta_c\right) \\
=\, & -\left(b(\theta_c) - b(\theta_b) - \mu_b\left(\theta_c - \theta_b\right)\right) - \left(\mu_b - \mu_a\right)\left(\theta_c - \theta_b\right) \\
=\, & -D_{\mathrm{KL}}\left(\mu_b, \mu_c\right) - \left(\mu_b - \mu_a\right)\left(\theta_c - \theta_b\right)
\end{aligned}
$$

$\square$

**Lemma 25** (Lower Bound of KL). *Given two distributions $\nu$ and $\nu'$ from $\mathcal{F}_m$ with means $\mu, \mu'$, respectively. Denote $\Delta := |\mu - \mu'|$. We have:*

1. *If $\mathcal{F}_m$ satisfies Assumption 4 with Lipschitzness constant $C_L$, we have*

$$
D_{\mathrm{KL}}\left(\mu, \mu'\right) \geq \frac{1}{2}\left(\frac{\Delta^2}{V(\mu) + C_L\Delta} \vee \frac{\Delta^2}{V(\mu') + C_L\Delta}\right)
$$

2. *If $\mathcal{F}_m$ satisfies Assumption 2, then*

$$
D_{\mathrm{KL}}\left(\mu, \mu'\right) \geq \frac{\Delta^2}{2\bar{V}}
$$

*Proof.*     1. $\mathcal{F}_m$ satisfies Assumption 4 with Lipschitzness constant $C_L$, then based on the integral form of $D_{\mathrm{KL}}\left(\mu, \mu'\right)$ in Lemma 22 we have

$$
\begin{aligned}
D_{\mathrm{KL}}\left(\mu, \mu'\right) &= \int_\mu^{\mu'} \frac{x - \mu}{V(x)}\,\mathrm{d}x \\
&\geq \int_\mu^{\mu'} \frac{x - \mu}{V(\mu) + C_L\Delta}\,\mathrm{d}x \vee \int_\mu^{\mu'} \frac{x - \mu}{V(\mu') + C_L\Delta}\,\mathrm{d}x \\
&= \frac{1}{2}\left(\frac{\Delta^2}{V(\mu) + C_L\Delta} \vee \frac{\Delta^2}{V(\mu') + C_L\Delta}\right)
\end{aligned}
$$

2. Under Assumption 2, for all $x \in [\mu, \mu']$, $V(x) \leq \bar{V}$, then

$$
D_{\mathrm{KL}}\left(\mu, \mu'\right) = \int_\mu^{\mu'} \frac{x - \mu}{V(x)}\,\mathrm{d}x \geq \int_\mu^{\mu'} \frac{x - \mu}{\bar{V}}\,\mathrm{d}x = \frac{\Delta^2}{2\bar{V}}
$$

$\square$

### E.3.3   Chernoff Bound for Exponential Family

**Lemma 26** (Chernoff Bound for Exponential Family). *(Ménard & Garivier, 2017) Given a natural number $N$ in $\mathbb{N}^+$, and a sequence of R.V.s $\{X_i\}_{i=1}^N$ drawn from a one parameter exponential distribution $\nu$ with model parameter $\theta$ and mean $\mu$. Let $\hat{\mu}_n := \frac{1}{n}\sum_{i=1}^n X_i, n \in \mathbb{N}^+$, which is the empirical mean of the first $n$ samples.*

*Then, for $y \geq 0$*

$$
\mathbb{P}(\exists n \geq N, D_{\mathrm{KL}}\left(\hat{\mu}_n, \mu\right) \geq y, \hat{\mu}_n < \mu) \leq \exp(-Ny) \tag{28}
$$

$$
\mathbb{P}(\exists n \geq N, D_{\mathrm{KL}}\left(\hat{\mu}_n, \mu\right) \geq y, \hat{\mu}_n > \mu) \leq \exp\left(-Ny\right) \tag{29}
$$

*Consequently, the following inequalities are also true:*

$$
\mathbb{P}(\hat{\mu}_N < \mu - \varepsilon) \leq \exp\left(-N \cdot D_{\mathrm{KL}}\left(\mu - \varepsilon, \mu\right)\right) \tag{30}
$$

$$
\mathbb{P}(\hat{\mu}_N > \mu + \varepsilon) \leq \exp\left(-N \cdot D_{\mathrm{KL}}\left(\mu + \varepsilon, \mu\right)\right) \tag{31}
$$

### E.3.4 Bounding the Sum of a Series of Geometric-log

**Lemma 27.** *Suppose that $T \in \mathbb{N}^+$ and $a > 1/T$ is a positive real number, we have the following:*

$$\sum_{k=1}^{T} \exp(-ka) \ln(T/k) \le \frac{5 \ln(Ta \vee e)}{a}$$

*Proof.* Here we consider two cases:

- $a \ge 1$

- $1/T < a < 1$

**Case 1:** $a \ge 1$  In this case, we note that $\ln(T/k) \le \ln(T) \le \ln(Ta)$ for all $k \ge 1$ and bound the sum using a geometric series.

$$\sum_{k=1}^{T} \exp(-ka) \ln(T/k) \le \sum_{k=1}^{T} \exp(-ka) \ln(Ta) \le \ln(Ta) \sum_{k=1}^{\infty} \exp(-ka)$$

$$= \ln(Ta) \frac{\exp(-a)}{1 - \exp(-a)} \le \frac{\ln(Ta)}{a} \le \frac{5 \ln(Ta \vee e)}{a}$$

**Case 2:** $1/T < a < 1$  In this case, we split the sum over $k$ into two ranges, one is $k \le \lceil \frac{1}{a} \rceil$ and another is $k > \lceil \frac{1}{a} \rceil$. For the sum in the first range, we can bound it by:

$$\sum_{k=1}^{\lceil \frac{1}{a} \rceil} \exp(-ka) \ln(T/k) \le \left\lceil \frac{1}{a} \right\rceil \ln(eT / \left\lceil \frac{1}{a} \right\rceil) \le \left\lceil \frac{1}{a} \right\rceil \ln(eTa) = \left\lceil \frac{1}{a} \right\rceil (\ln(Ta) + 1)$$

$$\le \left\lceil \frac{1}{a} \right\rceil 2 \ln(Ta \vee e) \le \frac{4 \ln(Ta \vee e)}{a}$$

For the first inequality, we bound $\exp(-ka)$ by 1 since $a > 0$. Then we use a well-known inequality $m! \ge (m/e)^m$ to bound the summation:

$$\sum_{k=1}^{\lceil 1/a \rceil} \ln(T/k) = \sum_{k=1}^{\lceil 1/a \rceil} \ln(T) - \ln(k) = \lceil 1/a \rceil \ln(T) - \ln(\lceil 1/a \rceil!)$$

$$\le \lceil 1/a \rceil \ln(T) - \lceil 1/a \rceil \ln(\lceil 1/a \rceil / e)$$

$$= \lceil 1/a \rceil \ln(eT / \lceil 1/a \rceil)$$

For the sum in the second range $k > \lceil \frac{1}{a} \rceil$, we can relax the $\ln(T/k)$ to $\ln(Ta)$ and proceed to bound it:

$$\sum_{k=\lceil \frac{1}{a} \rceil+1}^{T} \exp(-ka) \ln(T/k) \le \sum_{k=\lceil \frac{1}{a} \rceil+1}^{T} \exp(-ka) \ln(Ta) \le \frac{\ln(Ta)}{a}$$

Overall, we can bound the sum by combining the above two ranges,

$$\sum_{k=1}^{T} \exp(-ka) \ln(T/k) \le \frac{4 \ln(Ta \vee e)}{a} + \frac{\ln(Ta)}{a} \le \frac{5 \ln(Ta \vee e)}{a}$$

$\square$

### E.3.5 Integral Inequality

Below, we include the proof of a folklore lemma used in Jin et al. (2022); we include its proof here for completeness, as we cannot find a proof in the literature.

**Lemma 28.** *Given a nonnegative integrable function $f(x)$ which is unimodal in the range $[a, b]$, $a < b$ and $a, b \in \mathbb{N}^+$. We have the following inequality*

$$\sum_{i=a}^{b} f(i) \le \int_a^b f(x)\,dx + \max_{x \in [a,b]} f(x)$$

*Proof.* If $f(x)$ is increasing on $[c, c+1]$, we have the inequality $f(c) \le \int_c^{c+1} f(x)\,dx$. If $f(x)$ is decreasing on $[c, c+1]$, we have the inequality $f(c+1) \le \int_c^{c+1} f(x)\,dx$. Since the function $f(x)$ is unimodal in the range $[a, b]$, we can consider that there are four cases.

- If $f(x)$ is always increasing on $(a, b)$.

$$\sum_{i=a}^{b} f(i) = \sum_{i=a}^{b-1} \int_i^{i+1} f(i)\,dx + f(b)$$
$$\le \int_a^b f(x)\,dx + f(b) = \int_a^b f(x)\,dx + \max_{x \in [a,b]} f(x)$$

- If $f(x)$ is always decreasing on $(a, b)$.

$$\sum_{i=a}^{b} f(i) = \sum_{i=a+1}^{b} \int_{i-1}^{i} f(i)\,dx + f(a)$$
$$\le \int_a^b f(x)\,dx + f(a) = \int_a^b f(x)\,dx + \max_{x \in [a,b]} f(x)$$

- There exists a $c \in (a, b)$, $f(x)$ such that it is increasing on $[a, c]$ and is decreasing on $[c, b]$. When $c \in \mathbb{N}^+$, we have

$$\sum_{i=a}^{b} f(i) = \sum_{i=a}^{c-1} f(i) + \sum_{i=c+1}^{b} f(i) + f(c) = \sum_{i=a}^{c-1} \int_i^{i+1} f(i)\,dx + \sum_{i=c+1}^{b} \int_{i-1}^{i} f(i)\,dx + f(c)$$
$$\le \sum_{i=a}^{c-1} \int_i^{i+1} f(x)\,dx + \sum_{i=c+1}^{b} \int_{i-1}^{i} f(x)\,dx + f(c)$$
$$= \int_a^c f(x)\,dx + \int_c^b f(x)\,dx + f(c)$$
$$= \int_a^b f(x)\,dx + \max_{x \in [a,b]} f(x)$$

  When $c \in \mathbb{N}^+$, we can split the sum into $\sum_{i=a}^{\lfloor c-1 \rfloor}$, $\sum_{i=\lfloor c \rfloor}$ and $\sum_{i=\lceil c \rceil}$ and prove the Lemma.

- We omit the case where there exists a $c \in (a, b)$, $f(x)$ is decreasing on $[a, c]$ and is increasing on $[c, b]$, since the proof is very similar to the third case.

$\square$

## F    Experiments

### F.1    Regret comparison

We compare the regret performance of our algorithm, Exp-KL-MS, with that of several state-of-the-art methods, including kl-UCB (Cappé et al., 2013), ExpTS (Jin et al., 2022), and kl-UCB++ (Ménard & Garivier, 2017). The reward environments are adopted from Kaufmann et al. (2012). In the original experimental setup, two Bernoulli environments are considered: one with mean vector $[0.20, 0.25]$ and another with mean vector $[0.80, 0.90]$. In our experiments, we retain these mean vectors but replace the Bernoulli reward distributions with Gaussian, Gamma, and Poisson distributions. For each environment, we set the time horizon to $T = 10^4$ and the number of arms to $K = 2$. Each algorithm is evaluated over 64 independent trials.

We report the average pseudo reward of all algorithms as a function of time in Figure 3, and the average pseudo regret in Figure 4. Here, pseudo-reward refers to the noiseless reward, equivalently, the expected reward of the selected arm, while pseudo regret denotes the noiseless regret, that is, the difference between the expected reward of the optimal arm and that of the selected arm. The results indicate that Exp-KL-MS achieves performance comparable to that of the competing algorithms across both environments.

### F.2    Different $L(k)$ functions

We present theoretical regret analysis in Appendix A about the choice of $L(k)$ function affecting the regret analysis of General-Exp-KL-MS. In this section, we include a numerical experiment with General-Exp-KL-MS with different $L(k)$ functions to demonstrate that General-Exp-KL-MS is robust to the choice of $L$. We use the same environments as in Appendix F.1 and choose five different $L(k)$ functions: $L(k) = k/3, k/2, k-1, 2k, 3k$. We set the time horizon to be $T = 10^4$ and the number of arms to be $K = 2$. Each algorithm is run for 50 independent trials. We present the average pseudo reward against time steps in Figure 5 and the average pseudo regret in Figure 6. We see that General-Exp-KL-MS is robust to the choice of $L(k)$ functions and $L(k) = k - 1$ is more stable than other choices.

### F.3    Minimax regret's $K$-$T$ scaling

We now examine how the regret of Exp-KL-MS depends on the number of arms $K$ and the horizon $T$ in the worst case. Following the construction used to establish the $\sqrt{KT}$ lower bound for stochastic bandits (Auer et al., 2002; Audibert et al., 2009a; Lattimore & Szepesvári, 2020), we work with single-optimal-arm instances, where $\mu = (\mu_1, \mu_1 - \Delta, \mu_1 - \Delta, \ldots, \mu_1 - \Delta) \in \mathbb{R}^K$, with the minimax-tight gap $\Delta = \sqrt{K/T}$; we vary $K \in \{2, 4, 8, 16, 32\}$ and $T \in \{10^3, 3 \cdot 10^3, 10^4, 3 \cdot 10^4, 10^5\}$ (with the reward family and the choice of $\mu_1$ fixed per row, see Table 2), and then compute the regret of Exp-KL-MS and variants of General-Exp-KL-MS to verify that the upper bound of $R_T \lesssim \sqrt{KT \log K}$ in Corollary 2 is tight up to logarithmic factors. We restrict attention to two algorithms: Exp-KL-MS and the General-Exp-KL-MS variant with $L(k) = k/2$ analyzed in Appendix A. In each instance we repeat the experiment for 50 independent trials with independent random seeds, and report the average regret at the end of the horizon.

**Discussion.**    Figures 7 to 9 include the full $(K, T)$ grid as three heatmaps to visualize different regrets. Each heatmap is partitioned into 2 Exp-KL-MS-family algorithm column-blocks (separated by thick black lines) and, within each algorithm block, 5 reward-family row-blocks (separated by thin lines); within each row-block the cells span the five $K$ values (columns) and the five $T$ values (rows) at $\Delta = \sqrt{K/T}$. The unnormalized heatmap (Figure 7) shows the raw regret varying over both $K$ and $T$; the $\sqrt{KT \log K}$ rescaling (Figure 8) flattens the values to a near-constant range across the entire grid, consistent with the prediction of Corollary 2; the $\sqrt{KT \log T}$ rescaling (Figure 9) does not collapse equally well, confirming that the $\log K$ factor is the tighter one.

### F.4    Adaptive variance ratio

We now examine how the regret of Exp-KL-MS depends on the variance $V(\mu_{\max})$ of the optimal arm. Following the variance-aware bandit literature (Audibert et al., 2009b; Qin et al., 2023), we work with

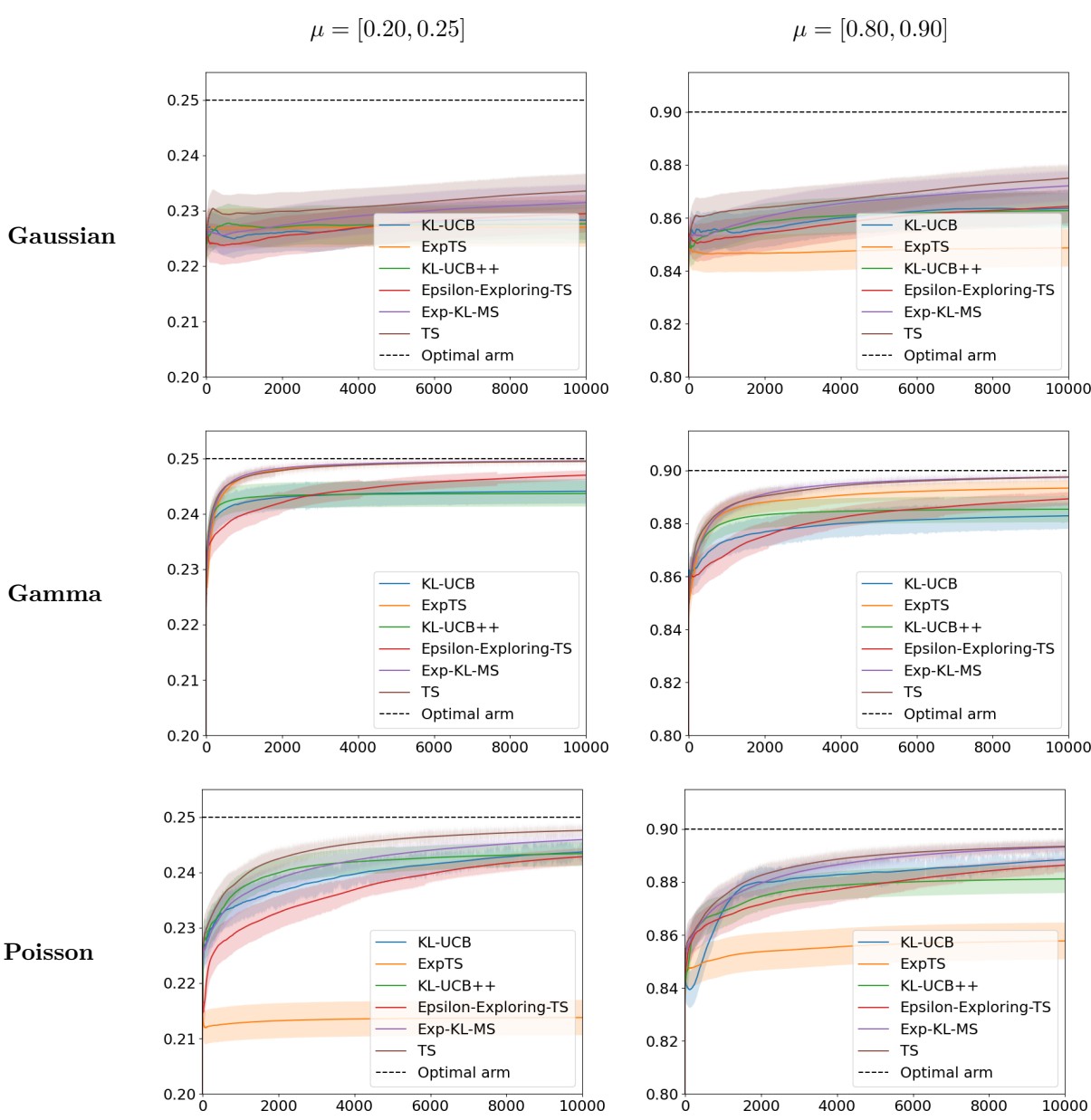

Figure 3: Average cumulative reward across different bandit algorithms designed for OPED reward distributions. "Optimal arm" refers to the expected reward of the optimal arm. x-axis is the time horizon and y-axis is the average cumulative pseudo reward.

single-optimal-arm instances, where $\mu = (\mu_1, \mu_1 - \Delta, \mu_1 - \Delta, \ldots, \mu_1 - \Delta) \in \mathbb{R}^K$, with the variance-tight gap $\Delta = \sqrt{V(\mu_1) K/T}$; we vary $V(\mu_1)$ with $(K, T)$ held fixed at $(32, 10^5)$ in each type of distribution, and then compute the regret of EXP-KL-MS and variants of GENERAL-EXP-KL-MS to verify that the upper bound of $R_T \lesssim \sqrt{V(\mu_{\max}) KT \log K}$ in Corollary 5 is tight up to logarithmic factors – a rate strictly sharper than the minimax rate of Corollary 2 whenever $V(\mu_{\max})$ is small. We use the same two algorithms as in Appendix F.3: EXP-KL-MS and the GENERAL-EXP-KL-MS variant with $L(k) = k/2$, and the four reward families of Table 2 (Bernoulli, Exponential, Gamma, Poisson). The Gaussian family is excluded because varying $V(\mu_1)$ there is implemented by directly setting $\sigma$, so the algorithm effectively knows $V(\mu_1)$ in advance. In each instance we repeat the experiment for 50 independent trials with independent random seeds, and report the average regret at the end of the horizon.

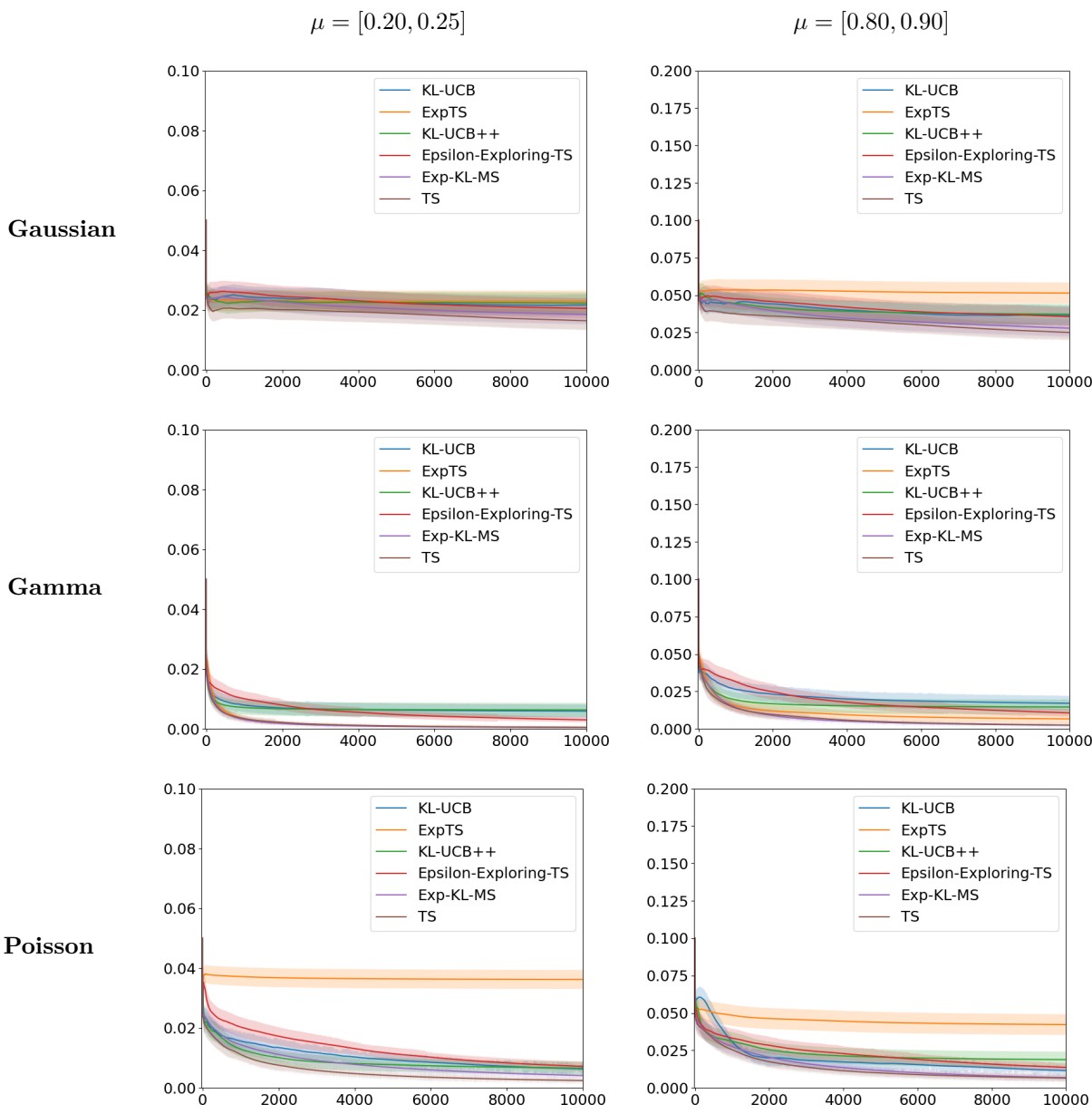

Figure 4: Average cumulative regret across different bandit algorithms designed for OPED reward distributions. The x-axis is the time horizon and the y-axis is the average cumulative pseudo regret.

**Discussion.** Figure 10 presents the full $V(\mu_1)$ changing as two heatmap panels placed side-by-side. The left panel reports the variance-adaptive rescaling $R_T/\sqrt{V(\mu_1)\,KT\log K}$, and the right panel reports the minimax rescaling $R_T/\sqrt{KT\log K}$ (which drops the $V(\mu_1)$ factor). Within each panel, rows index $V(\mu_1)$ from small to large and are partitioned into 2 EXP-KL-MS-family algorithm row-blocks (separated by thick black lines); columns index the four reward families (separated by thin lines). The variance-adaptive rescaling flattens the values to a near-constant range across the entire panel, consistent with the prediction of Corollary 5; in contrast, the minimax rescaling scales roughly as $\sqrt{V(\mu_1)}$ and becomes nearly zero for small $V(\mu_1)$ and grows to several hundred for the largest $V(\mu_1)$. The heatmap demonstrates that the minimax rate $\sqrt{KT\log K}$ is genuinely loose in the low-variance regime. The Bernoulli column covers a narrower $V$ range because $V(\mu) = \mu(1-\mu)$ is bounded by $1/4$.

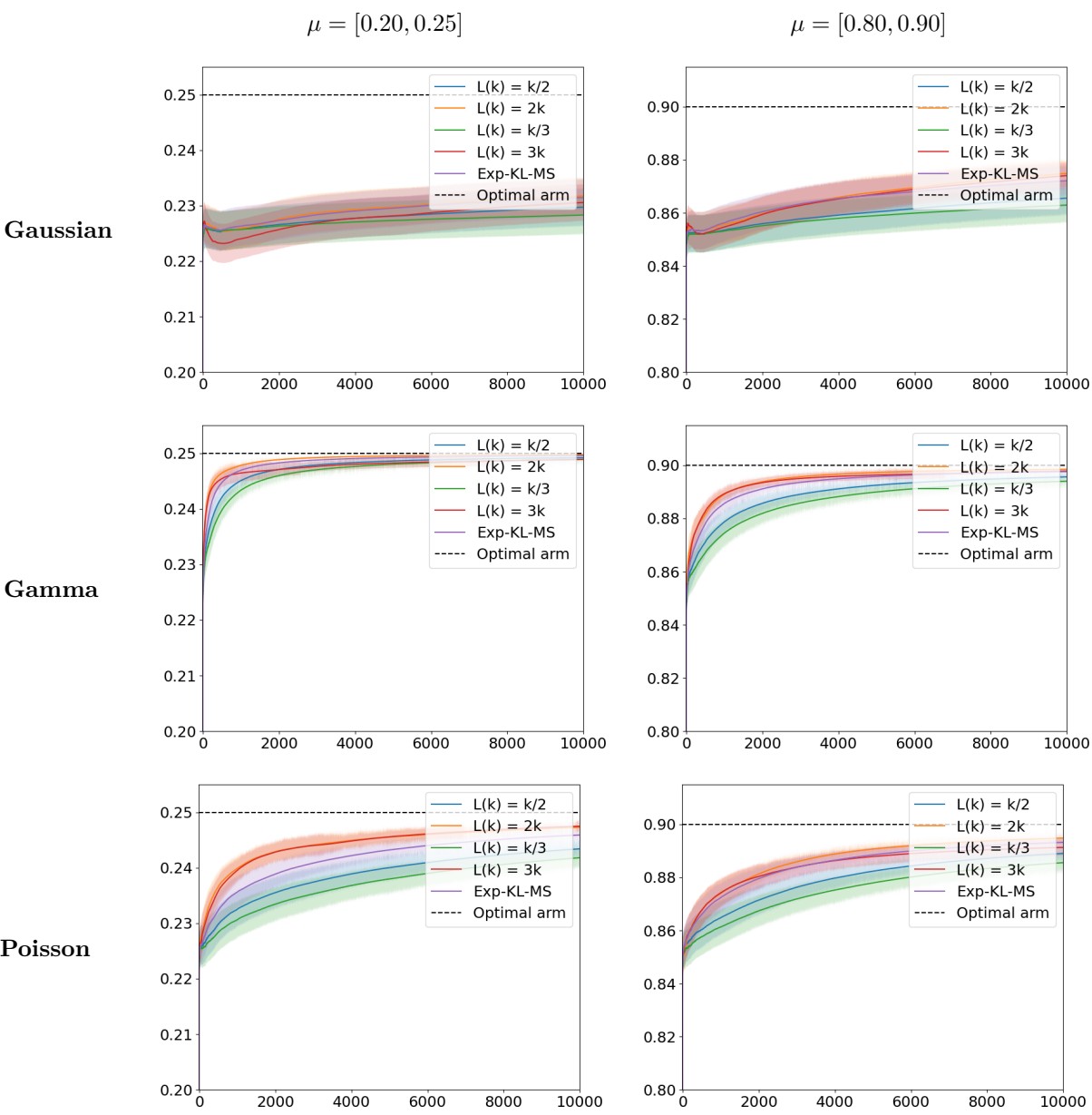

Figure 5: Average cumulative reward across different choices of $L(k)$ in General-Exp-KL-MS. "Optimal arm" refers to the expected reward of the optimal arm. The x-axis is the time step and the y-axis is the average cumulative pseudo reward.

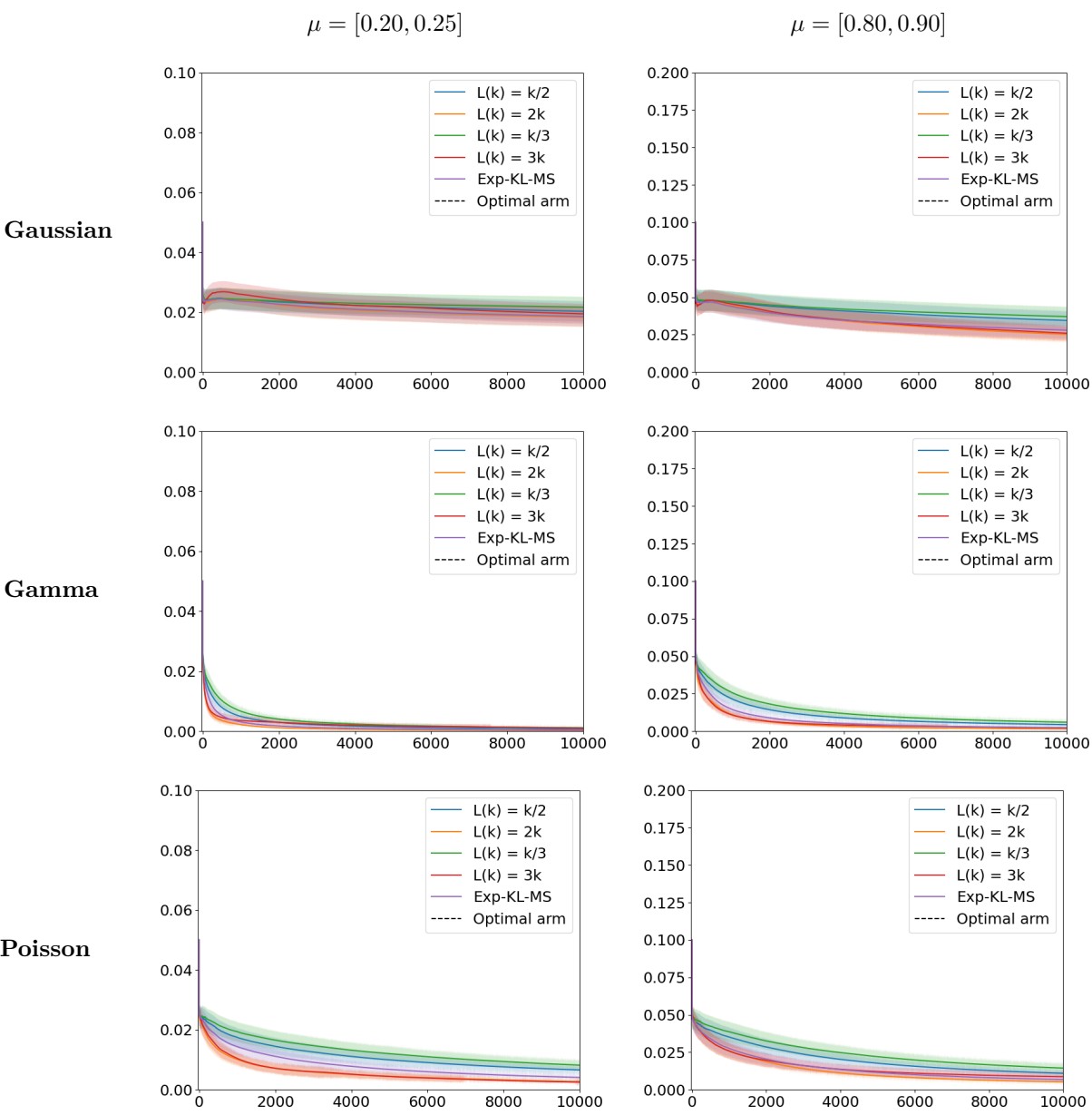

Figure 6: Average cumulative regret across different choices of $L(k)$ in Sᴍᴀʟʟ-Exᴘ-KL-MS. The x-axis is the time step, and y-axis is the average cumulative pseudo regret.

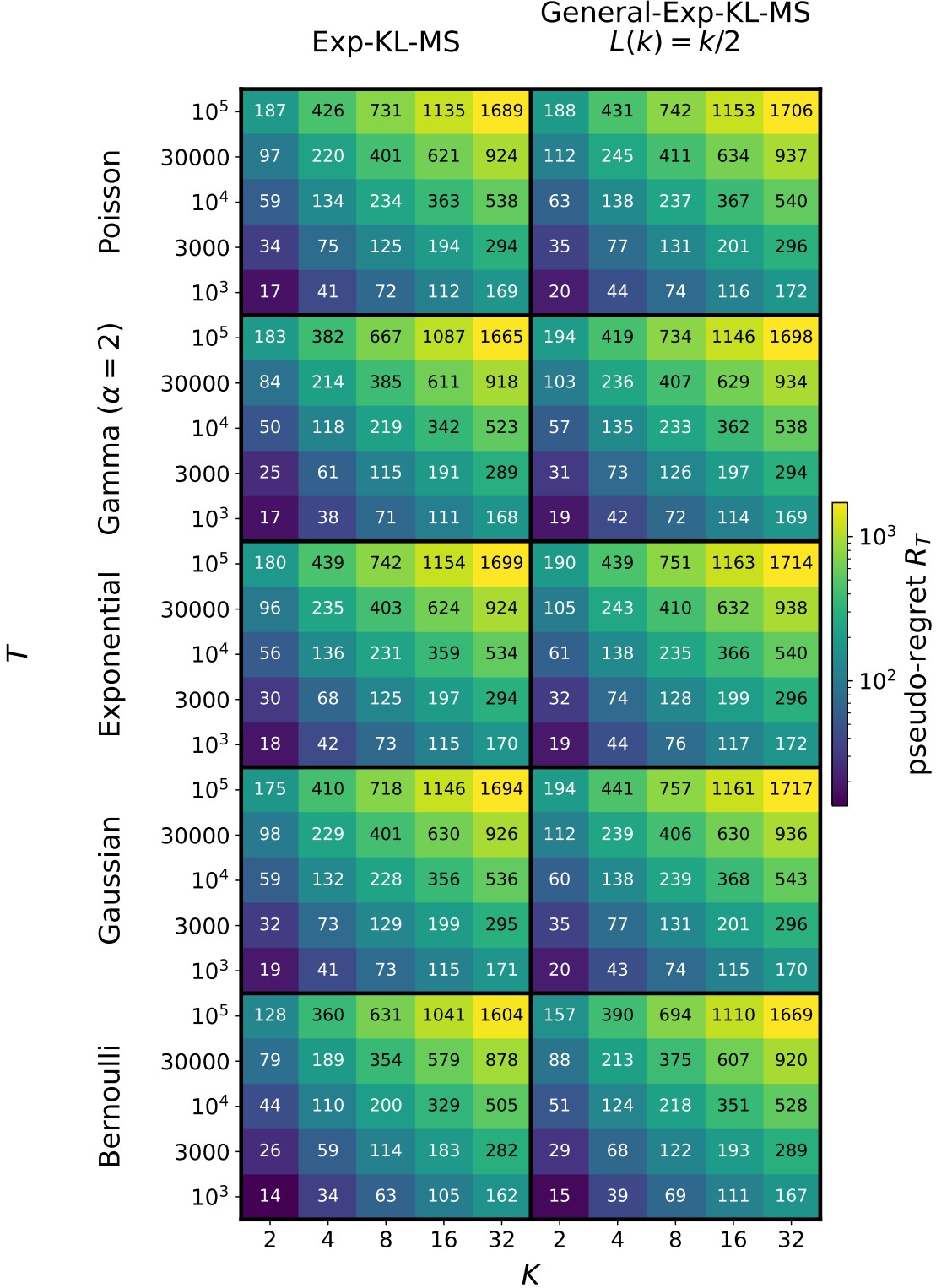

Figure 7: $(K, T)$ heatmap, unnormalized. Single $25 \times 10$ heatmap with the 5 reward-family row-blocks and the 2 EXP-KL-MS-family algorithm column-blocks delimited by solid black lines. Within each block, rows are $T \in \{10^3, 3 \cdot 10^3, 10^4, 3 \cdot 10^4, 10^5\}$ (smallest $T$ at the bottom) and columns are $K \in \{2, 4, 8, 16, 32\}$; cell values are the mean pseudo-regret $R_T$ over 50 runs at $\Delta = \sqrt{K/T}$. A single shared (log) color scale is used across the whole figure.

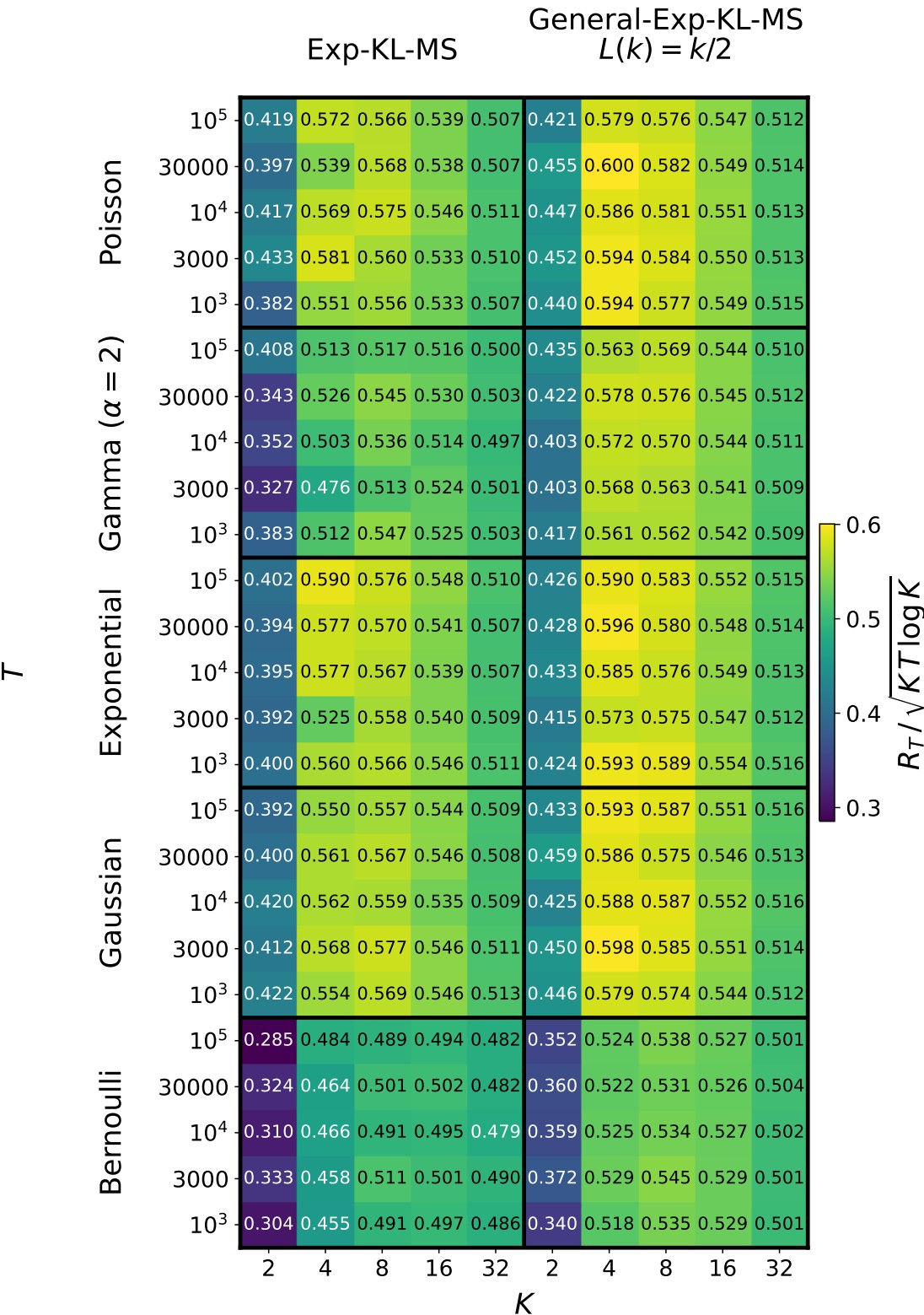

Figure 8: $(K, T)$ heatmap, normalized by $\sqrt{KT \log K}$. Same layout as Figure 7; the predicted minimax rate of Corollary 2 would collapse every cell to a constant, and the values indeed concentrate in a narrow range across the entire grid.

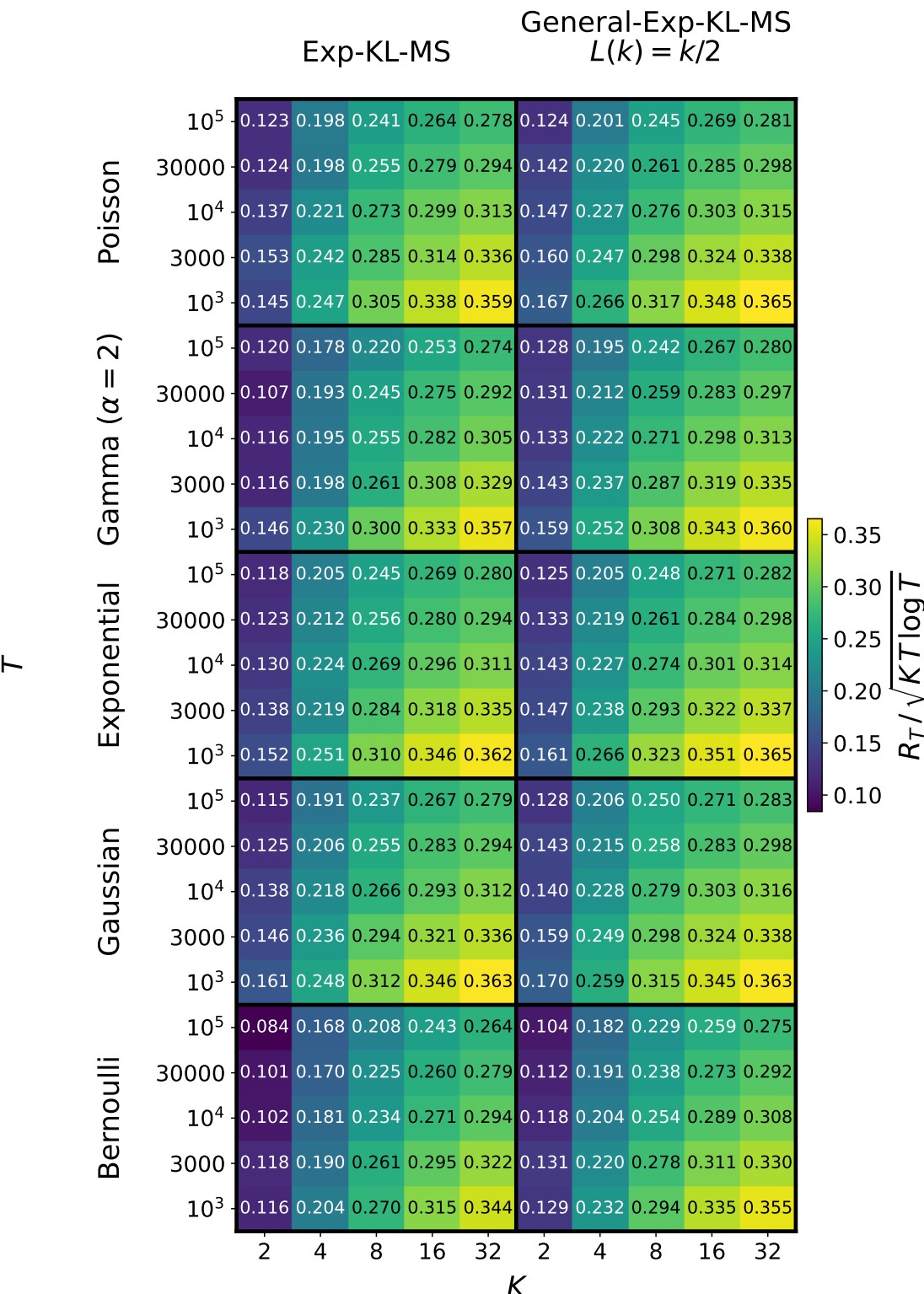

Figure 9: $(K, T)$ heatmap, normalized by $\sqrt{KT \log T}$. Same layout as Figure 7. The collapse does not happen as well as under the $\sqrt{KT \log K}$ rescaling, consistent with the latter being the tight rate.

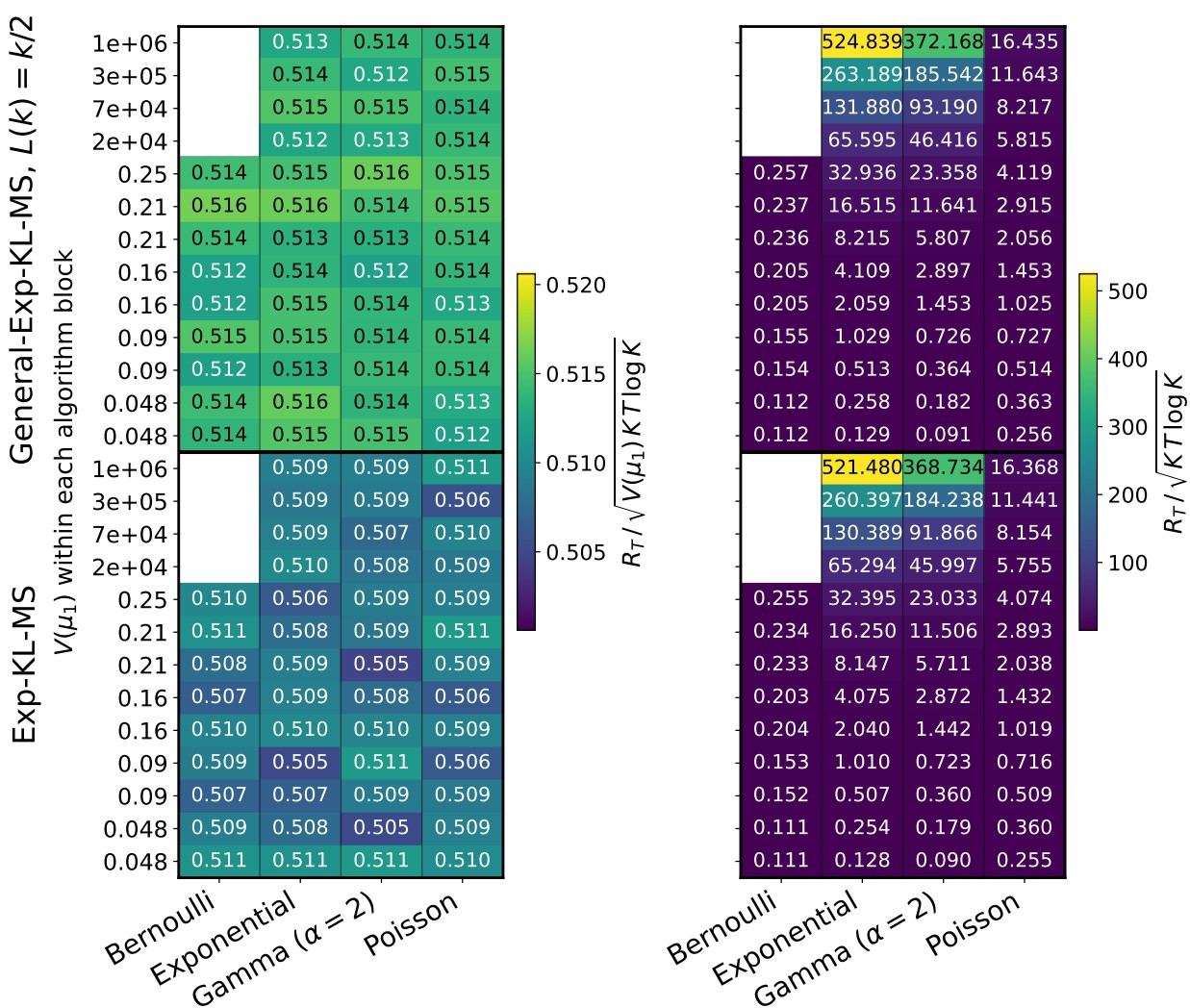

Figure 10: $V$ heatmap at fixed $(K, T) = (32, 10^5)$. Two panels side-by-side, left: variance-adaptive rescaling $R_T/\sqrt{V(\mu_1) \, KT \log K}$ (collapses to a near-constant); right: minimax rescaling $R_T/\sqrt{KT \log K}$ (grows roughly as $\sqrt{V(\mu_1)}$, confirming that the minimax rate is loose when $V(\mu_1)$ is small). Rows index $V(\mu_1)$ and are split into 2 EXP-KL-MS-family algorithm row-blocks (Exp-KL-MS at the bottom, $L(k) = k/2$ on top); columns index the four reward families.

