# OpenReview forum: "Achieving Adaptivity and Optimality for Multi-armed Bandits using Exponential-Kullback Leibler Maillard Sampling"
_TMLR — Accepted by TMLR_

### Review · Reviewer_Drfm · 2026-03-27

**Summary Of Contributions:**

This paper studies the K-armed bandit problem under OPED rewards. Its main contribution is to unify four types of guarantees (i.e., asymptotic optimality, a $\sqrt{\log K}$ minimax ratio, Sub-UCB, and an adaptive variance ratio) within a single Maillard-sampling-style algorithm. This is a substantial theoretical advance. The key technical contributions are mainly threefold:

1. The KL-Maillard sampling form itself in the OPED setting. The authors design the sampling probability of each arm as an exponential penalty based on the sample size multiplied by the KL distance to the empirically optimal distribution. This induces $N_{T, a} \approx \frac{\ln T}{D(\nu_a, \nu^{\ast})}$, which exactly matches the Lai-Robbins-type asymptotically optimal sampling ratio.
2. The change of the inverse temperature from $L(k)=k$ to $L(k)=k-1$. This seems like a very small modification, but it allows the analysis to control the part of the regret caused by the optimal arm performing very poorly in the short term, namely the hardest $\mathcal{B}_a^2$ term.
3. The upper-bound analysis for the hardest event $\mathcal{B}_a^2$ after the regret decomposition has a very elegant proof design.

However, the experimental part is relatively underdeveloped.

**Strengths**

1. The motivation is clear and the novelty is strong. The paper studies a very valuable unification problem: simultaneously achieving multiple bandit optimality/adaptivity criteria under general OPED assumptions. This framing is meaningful.
2. The theoretical work is very solid, and the technical route contains several interesting ideas.

**Weaknesses**

1. The experiments are clearly weak and do not match the headline claims.
2. Some key claims currently feel more conjectural or heuristic than fully established. For example, the authors state that for $L(k)=k$ they “did not succeed in showing” asymptotic optimality, and further conjecture that this may be a fundamental limitation. At the same time, they also state that they believe the $\sqrt{\ln K}$ minimax ratio proved in this paper is tight. In the current manuscript, both types of statements still need stronger support.

**Audience:**

Yes

**Audience Explanation:**

The problem is clearly of interest to the bandit / online learning / theoretical ML community. The paper targets a meaningful unification question under general OPED rewards, and the claimed combination of guarantees is stronger than what is usually obtained simultaneously in nearby lines of work.

**Broader Impact Concerns:**

No.

**Claims And Evidence:**

Yes

**Claims Explanation:**

The theoretical contribution is very solid. Although I found a few minor errors, I believe they can be fixed locally.

**Requested Changes:**

1. I found several derivations that seem problematic. I would like the authors to clarify the following:

    - In the proof of Proposition 18, the authors decompose $\mathcal{B}\_{a}^{2}$ into
        $$
        \mathcal{B}\_{a}^{2, 1} = \mathbb{E} \left[\sum\_{t=K+1}^{T} \mathbf{1} \{A\_{t, a} \cap \mathcal{E}(\alpha)\}  \right]
        \quad \text{and} \quad
        \mathcal{B}\_{a}^{2, 2} = \mathbb{E}\left[\sum\_{t=K+1}^{T} \mathbf{1}\{\mathcal{E}(\alpha)^{c}\} \right],
        $$
        and then use Lemma 19 and Lemma 20, respectively, to bound them. However, Lemma 19 actually bounds $\mathbb{E} \left[\sum\_{t=K+1}^{T} \mathbf{1}\{A\_{t, a} \cap F\_{t-1, a}^{c} \cap \mathcal{E}(\alpha)\} \right]$. Moreover, Lemma 20 is a bound for a lower-deviation event, whereas by definition $\mathcal{E}(\alpha)^c$ contains both lower and upper deviations, so it does not seem that Lemma 20 can be applied directly to bound it.

    - In Proposition 17 / Lemma 19, after taking $\alpha_k = R\_{\min}$, the authors claim that "$\mathcal{E}(\alpha)$ will not impose any additional constraints on $\hat{\mu}\_{(k), 1}$ except $F\_{t-1, a}^{c}$." However, according to the earlier definition, $\mathcal{E}(\alpha)$ also requires all $\hat{\mu}^{(k), 1} \leq \mu\_{\max} - \varepsilon\_{2}$, which is clearly much stronger than $F\_{t-1, a}^{c}$ at a single time step. Therefore,

        $$
        \mathcal{B}\_{a}^{2}
        = \mathbb{E} \left[\sum\_{t=K+1}^{T} \mathbf{1}\{A\_{t, a} \cap F\_{t-1, a}^{c}\} \right]
        = \mathbb{E} \left[\sum\_{t=K+1}^{T} \mathbf{1}\{A\_{t, a} \cap F\_{t-1, a}^{c} \cap \mathcal{E}(\alpha)\} \right]
        $$

        may not hold.

    However, I believe these two issues can be repaired locally, and at least they do not seem to affect the main conclusion at the level of order.

2. This paper makes very strong theoretical claims, but the experiments only provide a minimal sanity check. I understand that this is common in this type of bandit theory paper, but I still suggest that the authors strengthen the experimental section. At a minimum, it should include:

    - $K$-$T$ scaling experiments (simultaneously testing the minimax ratio, Sub-UCB, and asymptotic optimality). The most important conclusions of this paper are all about how the regret scales with $K$ and $T$, but the current experiments fix $K=2$ and $T=10^{4}$, which essentially only gives a small benchmark. The authors should at least scan a larger grid of $K$ and $T$, and report

        $$
        R\_T,\quad \frac{R\_T}{\sqrt{KT \log K}},\quad \frac{R\_T}{\sqrt{KT \log T}}.
        $$

    - Corollary 5 (adaptive variance ratio) states that under Assumption 4,

        $$
        R\_T = O\left( \sqrt{V(\mu\_{\max}) KT \ln K} + K \ln T\right),
        $$

        which is also one of the main innovations of this paper, but no corresponding experiment is provided. The authors could fix $K$, $T$, and the gap structure, and vary only $\mu\_{\max}$ so that $V(\mu\_{\max})$ changes systematically, then report the relationship between $R\_T$ and $V(\mu\_{\max})$.

3. Is $L(k)=k-1$ theoretically necessary, or is it only necessary within the current proof framework? Since this is a key design choice in the paper, can the authors provide a stronger mechanistic explanation?

---

> ### Author Response · Authors · 2026-05-22
> **Response to Reviewer Drfm**
>
> We thank Reviewer Drfm for the careful read of the proofs and for the constructive suggestions on experiments and algorithm design. We address each point below.
>
> **1. Two proof issues in the proofs of Propositions 17 and 18.**
>
> We thank the reviewer for pointing out these two issues and apologize for our errors. Both are addressed by an adjustment of the case split in the proof of Proposition 17 and Lemma 19 (Appendices D.3 and E). The root cause is that our old definition $\\mathcal{E}\_k(\alpha\_k) = \{\alpha\_k \leq \hat{\mu}\_{(k),1} \leq \mu\_{\max} - \varepsilon\_{2,a}\}$ conflated the lower-bound constraint (from $\boldsymbol{\alpha}$) with the upper-bound constraint (which should come from $F^c_{t-1,a}$). We redefine the clean event as lower-bound only, $\mathcal{E}\_k(\alpha\_k) := \\{\alpha\_k \leq \hat{\mu}\_{(k),1}\\}$, and let the upper bound $\hat{\mu}\_{(k),1} \leq \mu\_{\max} - \varepsilon\_2$ be supplied by $F^c_{t-1,a}$ inside the probability-transferring step of Lemma 19. With this single change, issue (i) is resolved because the $F^c_{t-1,a}$ contribution to ${B}^{2,1}_{a}$ is now explicit, and issue (ii) is resolved because $\\mathcal{E}(\boldsymbol{\alpha})^c = \\{\exists k: \hat{\mu}\_{(k),1} < \alpha\_{k}\\}$ now matches the LHS event of Lemma 20 verbatim. The asymptotic rates and all corollaries are unaffected.
>
> **2. Strengthening the experimental section.**
>
> We added two new experiments (Appendices F.3 and F.4) across five OPED reward families (Bernoulli, Gaussian, Exponential, Gamma, Poisson) that test the predictions of Corollaries 5 and 8 directly:
>
> - **Minimax $K$-$T$ scaling** (Appendix F.3): we use a single-optimal-arm instance at the minimax-tight gap $\Delta = \sqrt{K/T}$, with $K$ and $T$ varied across a $5 \times 5$ grid. The $1/\sqrt{KT \log K}$ rescaling of $R_T$ flattens across the entire grid, confirming the minimax upper bound of Corollary 5 is tight up to logarithmic factors.
> - **Adaptive variance ratio** (Appendix F.4): same instance at the variance-tight gap $\Delta = \sqrt{V(\mu_1)\,K/T}$, with $(K, T)$ fixed and $V(\mu_1)$ varied (via $\mu_1$ for non-Gaussian families and via $\sigma$ for Gaussian). The $1/\sqrt{V(\mu_1)\,KT \log K}$ rescaling flattens the curve, confirming the variance-adaptive bound of Corollary 5.
>
> **3. Is $L(k) = k-1$ theoretically necessary?**
>
> $L(k) = k - \delta$ for $\delta > 0$ is necessary using our current proof framework but, to our knowledge, may not be fundamental to the algorithm. After the probability-transferring step, asymptotic optimality requires bounding $\mathcal{B}^2_a$ by $T$-independent term; our framework does this via a sum of integrals $I_k$ whose integrand contains a factor $x^{k-1-L(k)}$ near $0$. When $L(k) = k$, this factor becomes $x^{-1}$, which is not integrable on unbounded-support OPED families (e.g., exponential rewards), forcing the strict inequality $L(k) < k$. Appendix D.4 of the revised manuscript gives the full derivation with the exponential example.
>
> ---
>
> **References**
>
> - Jourdan, M., Degenne, R., Baudry, D., de Heide, R., & Kaufmann, E. (2022). Top two algorithms revisited. *Advances in Neural Information Processing Systems*, 35.
> - Lattimore, T. (2018). Refining the confidence level for optimistic bandit strategies. *The Journal of Machine Learning Research*, 19.
> - Qin, H., Jun, K.-S., & Zhang, C. (2023). Kullback-Leibler Maillard sampling for multi-armed bandits with bounded rewards. *Advances in Neural Information Processing Systems*, 36.

---

> > ### Comment · Reviewer_Drfm · 2026-05-25
> >
> > I have read the authors’ response and the revised manuscript. The authors have satisfactorily addressed my main concerns. In particular, the clarification and correction of the proof around the clean event and the $B_a^2$ decomposition resolve the issues I raised about Propositions 17 and 18. The added experiments on $K$-$T$ scaling and the adaptive variance ratio also substantially strengthen the empirical support for the main theoretical claims. Finally, the explanation of the role of $L(k)=k-1$ is now much clearer, especially in distinguishing what is required by the current proof framework from what may or may not be fundamental to the algorithm.
> >
> > Overall, I am satisfied with the revision and do not have further questions for the authors.

---

### Review · Reviewer_JHBV · 2026-04-04

**Summary Of Contributions:**

This paper studies stochastic multi-armed bandits with rewards from one-parameter exponential families, and proposes a sampling-based algorithm, Exp-KL-MS, with arm probabilities proportional to an exponential of an empirical KL gap. The main claim is that, with the choice $L(k)=k-1$, the algorithm simultaneously achieves asymptotic optimality, a $\sqrt{\ln K}$ minimax ratio, Sub-UCB, and an adaptive variance-ratio guarantee. A strength of the paper is that it extends Maillard-sampling-style ideas beyond the bounded/Bernoulli setting and gives a reasonably detailed proof sketch for why the $L(k)=k-1$ modification matters. My main weaknesses are about positioning rather than technical correctness: I found the novelty relative to Qin et al. (2023) not fully clear, the use of instance-adaptive somewhat confusing, and the motivation for Sub-UCB not developed enough.

**Additional Comments:**

Overall, I think the paper is technically solid and could be accepted after revision.

**Audience:**

Yes

**Audience Explanation:**

Researchers in the bandit community may find this topic interesting.

**Broader Impact Concerns:**

I do not have major broader-impact concerns.

**Claims And Evidence:**

Yes

**Claims Explanation:**

At a high level, the paper does provide the proof sketch, and a coherent story connecting the main regret bound to the four advertised properties.

**Requested Changes:**

First, I think the paper should clarify much more carefully what is meant by “instance-adaptive.” Right now Table 1 marks kl-UCB++ as having a positive variance-ratio entry, while ExpTS+ is marked “No.” My understanding from Section 3 is that the favorable kl-UCB++ entry comes from the refined Bernoulli analysis in Qin et al. (2023), whereas ExpTS / ExpTS+ are discussed under analyses that use a maximum-variance assumption and therefore do not yield adaptive-variance guarantees. If that is the intended comparison, the paper should say this explicitly. As written, the table mixes guarantees proved in somewhat different settings, and that makes the comparison feel harder to parse than it should be.

Second, I would strongly encourage the authors to rephrase the adaptive-variance discussion. The bound involving $\sqrt{V(\mu_{\max})KT}$ is not the same kind of statement as a worst-case guarantee of order $\sqrt{KT}$. The paper itself classifies adaptive variance ratio as a partially instance-independent notion, and even explains that it can be much smaller than $\sqrt{KT}$ on favorable instances such as Bernoulli rewards with $\mu_{\max}$ near 0 or 1. Because of this, wording like “variance-adaptive worst-case regret bound” feels potentially misleading. I think this should be clarified in the main text.

Third, the novelty relative to Qin et al. (2023) should be stated more directly. Qin et al. already gave a sampling-based KL-MS result with minimax-type, Sub-UCB, and adaptive-variance guarantees in the bounded/Bernoulli setting. That is still interesting, but the current writeup sometimes makes the contribution sound broader than that. I would like the authors to sharpen this comparison and explain exactly where the new technical difficulty is. This is important for acceptance.

Fourth, I think the motivation for Sub-UCB should be strengthened. The paper cites the standard point from Lattimore (2018), namely that asymptotic optimality and minimax optimality do not fully characterize finite-time behavior, and it notes that algorithms like MOSS and kl-UCB++ can fail Sub-UCB. But this remains a bit abstract in the current version. It would help a lot to include one concrete instance or figure showing an algorithm that is asymptotically and minimax optimal but behaves poorly in finite time, so that the need for Sub-UCB becomes more tangible.

Finally, I would like a slightly more explicit future-work discussion around whether one can get a sampling-based algorithm that is asymptotically optimal, constant-minimax optimal, and Sub-UCB all at once in this setting. The paper already notes that for Exp-KL-MS the minimax ratio remains $\sqrt{\ln K}$, and that achieving constant minimax ratio together with Sub-UCB is open even here. Since this seems closely related to the main limitation of the proposed approach, it would be helpful to discuss it more openly.

---

> ### Author Response · Authors · 2026-05-22
> **Response to Reviewer JHBV**
>
> We thank Reviewer JHBV for the careful read and for the thoughtful suggestions on positioning. We address each point below.
>
> **1. Clarifying "instance-adaptive" in Table 1.**
>
> All entries in Table 1 refer to the general OPED reward setting (Assumption 1), and we now state this explicitly in the revised caption. Concretely, we (i) re-mark kl-UCB++'s Variance Ratio entry as "No$^\star$", since its published analysis does not yield an adaptive variance ratio in the general OPED setting (the $^*$, defined in the caption, points to the Bernoulli-setting result of Qin et al. (2023)); (ii) add a caption note explaining that ExpTS/ExpTS$^+$ rely on the maximum-variance assumption, so it is unclear whether their bounds satisfy the adaptive variance ratio guarantee; and (iii) align the related-work discussion in Section 3 by adding the "Bernoulli reward setting" qualifier to the corresponding sentence about Qin et al. (2023).
>
> **2. Renaming "adaptive-variance worst-case regret bound".**
>
> We agree that "variance-adaptive worst-case regret bound" is misleading: the bound $\sqrt{V(\mu_{\max}) KT}$ is partially instance-dependent (it depends on the instance-specific quantity $V(\mu_{\max})$), not a worst-case bound of order $\sqrt{KT}$. In the revised manuscript we have dropped the "worst-case" qualifier throughout.
>
> **3. Novelty relative to Qin et al. (2023).**
>
> We have added a short remark to the "Our Techniques" paragraph of Section 1 pointing to the full technical explanation in Appendix D.4, placed immediately after Propositions 17 and 18 so that the condition mismatch between the two propositions is visible at the point of use. The appendix shows, via an exponentially-distributed reward example, that the integral in the asymptotic-optimality bound diverges precisely at $L(k) = k$ (it behaves like $\int_0^1 x^{-1} dx$); this integrability gap, absent in the Bernoulli analysis of Qin et al. (2023), forces our framework to stay at $L(k) < k$ and is the reason we believe asymptotic optimality for $L(k)=k$ is no longer easy to show.
>
> **4. Strengthening Sub-UCB motivation with a concrete example.**
>
> We added a discussion in Section 3 (related work) recalling the MOSS failure instance of Lattimore (2018): a $K$-armed Gaussian bandit with gaps $\Delta_2 = 1/K$, $\Delta_a = 1$ for $a > 2$, and horizon $T = K^3$. Although MOSS achieves asymptotically and minimax optimal, MOSS incurs $\Omega(\sqrt{KT}) = \Omega(K^2)$ regret while the simpler UCB incurs only $O(K \ln K)$. This makes the necessity of the Sub-UCB criterion concrete at the point where it is introduced.
>
> **5. On removing the $\sqrt{\ln K}$ factor in the worst-case bound.**
>
> We have expanded the discussion on achieving a constant minimax ratio and Sub-UCB simultaneously in Section 5, p. 9, "On achieving constant minimax ratio and sub-UCB simultaneously." Our updated text copied here for your convenience:
>
> > "Specifically, the data-dependent confidence-level adjustment of ADA-UCB (Lattimore, 2018) constructs its confidence width at arm $a$ based on a data-dependent quantity $H_a(t-1)$ that couples across arms, and no other route to a constant minimax ratio with Sub-UCB is currently known. Translating this to our Maillard-sampling framework, the term that enters $\log(\cdot)$ in a UCB index corresponds to a multiplicative factor in front of $\exp(\cdot)$ in our sampling probability. We therefore speculate that multiplying $1/H$ (with $H$ as defined in Lattimore (2018)) in front of the $\exp(\cdot)$ in our sampling rule may enable us to remove the $\sqrt{\ln K}$ factor while preserving Sub-UCB. The corresponding analysis, however, is substantially more involved, and we leave it as future work."
>
> ---
>
> **References**
>
> - Lattimore, T. (2018). Refining the confidence level for optimistic bandit strategies. *The Journal of Machine Learning Research*, 19.
> - Qin, H., Jun, K.-S., & Zhang, C. (2023). Kullback-Leibler Maillard sampling for multi-armed bandits with bounded rewards. *Advances in Neural Information Processing Systems*, 36.

---

> > ### Comment · Reviewer_JHBV · 2026-06-01
> >
> > I am satisfied with the rebuttal and do not have further questions.

---

### Review · Reviewer_pbWk · 2026-05-10

**Summary Of Contributions:**

The paper studies four important criteria for quantifying the performance of multi-armed bandit algorithms---asymptotic optimality, minimax optimality, sub-UCB criterion, and adaptive variance---and proposes an algorithm (Generalised-Exp-KL-MS) for the stochastic multi-armed bandit setting that meets the aforementioned criteria simutaneously. Notably, the proposed algorithm is based on a simple, yet effective idea of Maillard sampling, and is profoundly easy to implement. Several popular algorithms available in the existing literature such as KL-MS, MS, MED, etc., arise as special cases of the proposed algorithm. Numerical results indicate that the proposed algorithm outperforms existing baselines while achieving all of the aforementioned performance criteria.

The paper is well-written and easy to follow.

**Audience:**

Yes

**Audience Explanation:**

Asymptotic optimality, minimax optimality, sub-UCB criterion, and adaptive variance are useful performance measures that several papers in the recent past have begun looking into. The first two criteria, in particular, have been well investigated since the time of Lai and Robbins's seminal paper on multi-armed bandit algorithms. I am therefore certain that the result of the current paper will be of broad use to the community of researchers working in multi-armed bandits, or more generally in reinforcement learning. Given the profound simplicity of the sampling algorithm, it would be interesting to see if a similar algorithm could be used in the setting of pure exploration in multi-armed bandits. In the latter setting, designing an efficient sampling rule that is provably asymptotically optimal has remained one of the biggest challenges. MS-style algorithms, if successful for pure exploration problems, could pave the way for efficient implementations in practical scenarios, and the current paper provides a natural link to explore the connection between MS-style sampling schemes and pure exploration.

**Broader Impact Concerns:**

The work is theoretical with not much of concerns on impact to report at this point.

**Claims And Evidence:**

Yes

**Claims Explanation:**

The main result of the paper in Theorem 1 is stated clearly. Although I did not go through the detailed proof of the theorem (provided in the Appendix), I did parse the proof outline provided in Section 6, and it seems to check out fine. In my opinion, the key ingenuity in the proof is to come up with the events $\mathcal{A}\_{t,a}, \mathcal{G}\_{t,a}, \mathcal{B}\_{t,a}^{1}, \mathcal{B}\_{t,a}^2$, and the authors have done a good job in describing these events clearly in Section 6 without the reader having to refer the appendix for the description of the aforementioned events (which form the backbone of the proof). The key claims of the paper are contained in Corollary 2-5, and they look good to me. The simplicity of the arm probabilities--particularly the choice of $L(\cdot)$--stood out for me.

**Requested Changes:**

The authors use the phrases "In our main algorithm, Generalised-Exp-KL-MS" and "our main algorithm Exp-KL-MS" interchangeably. I understand that the latter is a special case of the former with L(k)=k-1, but it may be better to avoid the usage of both of the aforementioned phrases in the same paragraph to the best possible extent.

---

> ### Author Response · Authors · 2026-05-22
> **Response to Reviewer pbWk**
>
> We thank Reviewer pbWk for the positive assessment and for highlighting the simplicity of the arm-sampling rule and the cleanness of the case decomposition.
>
> **Requested change: "General-Exp-KL-MS" vs. "Exp-KL-MS" used interchangeably.**
>
> We apologize for the confusion. In the revised Section 4, we make the distinction explicit: General-Exp-KL-MS refers to our main "algorithmic framework" (Algorithm 1), while Exp-KL-MS refers to our main "algorithm", the specific instance of General-Exp-KL-MS with $L(k) = k-1$. We use the two terms consistently throughout the revised manuscript.
>
> **On the suggestion to explore an MS-style algorithm for pure exploration.**
>
> We agree that such a design is plausible. We have added a paragraph to Section 7 acknowledging this as a follow-up direction and sketching a top-two-sampling-inspired (Jourdan et al., 2022) algorithm that preserves a closed-form sampling distribution.
>
> ---
>
> **References**
>
> - Jourdan, M., Degenne, R., Baudry, D., de Heide, R., & Kaufmann, E. (2022). Top two algorithms revisited. *Advances in Neural Information Processing Systems*, 35.

---

> ### Comment · Reviewer_pbWk · 2026-06-09
> **No further questions**
>
> I am satisfied with the authors' responses. I have no further questions to the authors at this point.

---

### Author Response · Authors · 2026-05-22
**Shared response to reviews**

We thank all reviewers for their careful reading and constructive feedback. We have revised the manuscript accordingly, and we address each point below. We highlight new content in the revised manuscript in $\textcolor{blue}{\text{blue}}$ color and deleted content in $\textcolor{red}{\text{red}}$ color.

---

### Decision · Action_Editor_Sfmy · 2026-06-15

**Recommendation:** Accept as is

**Audience:**

Yes

**Audience Explanation:**

Bandits is a topic of interest to the ML community.

**Claims And Evidence:**

Yes

**Claims Explanation:**

The claims are correct (as far as I and the reviewers can tell) and the work is of interest to the ML community.